# Matrix-associated extracellular vesicles modulate human smooth muscle cell adhesion and directionality by presenting collagen VI

Alexander N Kapustin[1]*, Sofia Serena Tsakali[1], Meredith Whitehead[1], George Chennell[2], Meng-Ying Wu[1,3], Chris Molenaar[1], Anton Kutikhin[4], Yimeng Chen[1], Sadia Ahmad[1], Leo Bogdanov[4], Maxim Sinitsky[5], Kseniya Rubina[6], Aled Clayton[7], Frederik J Verweij[8], Dirk Michiel Pegtel[9], Simona Zingaro[10], Arseniy Lobov[11], Bozhana Zainullina[12], Dylan Owen[13], Maddy Parsons[10], Richard E Cheney[14], Derek T Warren[15], Martin James Humphries[16], Thomas Iskratsch[17], Mark Holt[9], Catherine M Shanahan[1]

[1]School of Cardiovascular and Metabolic Medicine and Sciences, James Black Centre, King's College London, London, United Kingdom; [2]Wohl Cellular Imaging Centre, King's College London, London, United Kingdom; [3]Department of Urology, The Third Affiliated Hospital of Soochow University, Suzhou, China; [4]Laboratory for Molecular, Translational and Digital Medicine, Research Institute for Complex Issues of Cardiovascular Diseases, Kemerovo, Russian Federation; [5]Laboratory of Genome Medicine, Research Institute for Complex Issues of Cardiovascular Diseases, Kemerovo, Russian Federation; [6]Laboratory of Morphogenesis and Tissue Reparation, Faculty of Medicine, Lomonosov Moscow State University, Moscow, Russian Federation; [7]Tissue Microenvironment Research Group, Division of Cancer & Genetics, School of Medicine Cardiff University, Cardiff, United Kingdom; [8]Division of Cell Biology, Neurobiology & Biophysics, Utrecht University, Utrecht, Netherlands; [9]Department of Pathology, Amsterdam UMC, Location Vrije Universiteit Amsterdam, Cancer Center Amsterdam, Amsterdam, Netherlands; [10]Randall Centre for Cell and Molecular Biophysics, School of Basic and Medical Biosciences, King's College London, New Hunt's House, London, United Kingdom; [11]Laboratory of Regenerative Biomedicine, Institute of Cytology of the Russian Academy of Sciences, Moscow, Russian Federation; [12]Centre for Molecular and Cell Technologies, Research Park, St. Petersburg State University, St Petersburg, Russian Federation; [13]Institute of Immunology and Immunotherapy, School of Mathematics and Centre of Membrane Proteins and Receptors (COMPARE), University of Birmingham, Birmingham, United Kingdom; [14]Department of Cell Biology and Physiology, School of Medicine, University of North Carolina at Chapel Hill, Chapel Hill, United States; [15]School of Pharmacy, University of East Anglia, Norwich Research Park, Norwich, United Kingdom; [16]Wellcome Centre for Cell-Matrix Research, Faculty of Biology, Medicine & Health, Manchester Academic Health Science Centre, University of Manchester, Manchester, United Kingdom; [17]School of Engineering and Materials Science, Faculty of Science and Engineering, Queen Mary University of London, London, United Kingdom

*For correspondence:
Alexander.kapustin@kcl.ac.uk

## eLife Assessment

This paper explores the role of extracellular vesicles in providing extracellular matrix signals for migration of vascular smooth muscle cells. The evidence, based on cell culture experiments and supporting imaging of human samples, is mostly **convincing**. The paper will be **valuable** for researchers investigating cell migration during vessel repair and atherogenesis.

**Abstract** The extracellular matrix (ECM) supports blood vessel architecture and functionality and undergoes active remodelling during vascular repair and atherogenesis. Vascular smooth muscle cells (VSMCs) are essential for vessel repair and, via their secretome, can invade from the vessel media into the intima to mediate ECM remodelling. Accumulation of fibronectin (FN) is a hallmark of early vascular repair and atherosclerosis. Here, we show that FN stimulates human VSMCs to secrete small extracellular vesicles (sEVs) by activating the β1 integrin/FAK/Src pathway as well as Arp2/3-dependent branching of the actin cytoskeleton. We found that sEVs are trapped by the ECM in vitro and colocalise with FN in symptomatic atherosclerotic plaques in vivo. Functionally, ECM-trapped sEVs induced the formation of focal adhesions (FA) with enhanced pulling forces at the cellular periphery preventing cellular spreading and adhesion. Proteomic and GO pathway analysis revealed that VSMC-derived sEVs display a cell adhesion signature and are specifically enriched with collagen VI on the sEV surface. In vitro assays identified collagen VI as playing a key role in cell adhesion and invasion directionality. Taken together, our data suggests that the accumulation of FN is a key early event in vessel repair acting to promote secretion of collagen VI enriched sEVs by VSMCs. These sEVs stimulate directional invasion, most likely by triggering peripheral focal adhesion formation and actomyosin contraction to exert sufficient traction force to enable VSMC movement within the complex vascular ECM network.

## Introduction

The healthy arterial vasculature is dominated by a highly organised medial extracellular matrix (ECM) containing organised layers of contractile vascular smooth muscle cells (VSMCs). Vascular pathologies such as atherosclerosis are associated with dramatic remodelling of the ECM ultimately leading to plaque rupture and myocardial infarction or stroke (*Libby et al., 2011*). Progenitor medial VSMCs are essential for vessel repair and must invade through the ECM to form the protective intimal fibrous cap in the plaque. This process of VSMC invasion actively contributes to ECM remodelling (*Chappell et al., 2016*; *Misra et al., 2018*; *Durham et al., 2018*). The accumulation of liver-derived fibronectin (FN) in the vasculature is an early biomarker of atherosclerotic plaque formation and conditional FN knockout in the liver blocked VSMC invasion and fibrous cap formation in the ApoE mouse model. This suggests that FN is an essential signal for VSMC recruitment and invasion during vascular repair, yet its exact role remains unknown (*Rohwedder et al., 2012*; *Langley et al., 2017*; *Glukhova et al., 1989*).

FN has been shown to play key signalling roles by modulating cellular spreading, adhesion, invasion, differentiation, and viability during both developmental and pathological processes (*Pankov and Yamada, 2002*). Cell adhesion to FN is primarily mediated by α5β1 integrins and proteoglycans acting together to activate small GTP-binding proteins, Cdc42, Rac1, and Rho which in turn induce branched actin cytoskeletal rearrangements to form cellular membrane protrusions, filopodia, and lamellipodia, which are attached to the ECM via transient peripheral focal complexes (*Pankov and Yamada, 2002*; *Hocking et al., 1998*; *Bass et al., 2007*; *Nobes and Hall, 1995*). In turn, maturation of focal complexes into focal adhesions (FAs) anchors cytoplasmic actin stress fibres to the ECM and actin polymerisation and actomyosin-mediated contractility generate the traction forces required for cell body locomotion (*Sheetz et al., 1998*; *Ridley, 2001*).

Exosome-like small extracellular vesicles (sEVs) are novel, cell-matrix crosstalk entities that decorate the ECM and form 'migration paths' for tumour cells by enhancing cell adhesion, motility, and directionality (*Sung et al., 2015*; *Sung and Weaver, 2017*; *Sung et al., 2020*; *Koumangoye et al., 2011*). Mechanistically, sEVs are secreted at the cellular leading edge and promote FA formation by presenting FN (*Sung et al., 2015*; *Sung and Weaver, 2017*; *Hoshino et al., 2013*; *Clayton et al.,*

*2004*). In addition, sEVs can stimulate motility of immune cells and *Dictyostelium discoideum* by presenting cytokines and/or generating chemoattractants (*Brown et al., 2018*; *Majumdar et al., 2016*; *Kriebel et al., 2018*). We recently showed that FN is also enriched in VSMC-derived sEVs, but the exact role of sEVs in VSMC migration and invasion within the ECM meshwork environment of the vasculature remains unexplored (*Kapustin et al., 2015*).

Here, we report that FN in the ECM induces sEV secretion by activating β1 integrin/FAK/Src and Arp2/3-dependent actin cytoskeleton assembly. In turn, sEVs are trapped to ECM, and these ECM-associated sEVs regulate VSMC invasion directionality in a 3D model. Mechanistically, ECM-associated sEVs induced earlier formation of focal adhesions (FA) with enhanced pulling forces at the cell periphery, thus switching the leading edge from protrusion activity into contractile mode to enable cell body propulsion. Importantly, we found that the sEV cargo, collagen VI, is indispensable for triggering FA formation and directed cell invasion. We hypothesise that sEV-triggered peripheral FA formation anchors the cellular leading edge and activates cell contraction to exert sufficient force to allow VSMC invasion into the complex ECM fibre meshwork. This novel mechanism opens a unique therapeutic opportunity to target specifically VSMC invasion activity during vascular repair and restenosis.

## Results

### ECM components involved in vessel injury repair stimulate sEV release via b1 integrins

FN is detectable in the vasculature as early as the fatty streak stage and preceding major atherogenic alterations such as neointima formation and it is also a novel marker for symptomatic carotid plaques (*Rohwedder et al., 2012*; *Langley et al., 2017*; *Glukhova et al., 1989*). Given its presence during early stages of vessel injury, we hypothesised that FN may modulate sEV secretion to enable vessel repair. FN is secreted as a monomeric protein forming fibrils upon cellular binding, so we compared the effects of monomeric FN with the effects of FN immobilised on tissue culture plates to mimic FN fibrils (*Pankov and Yamada, 2002*). Plating VSMCs on immobilised FN increased the release of CD63+/CD81+sEVs 3.5±0.6 fold whilst addition of soluble FN had no effect on EV secretion as detected by CD63-bead capture assay (*Figure 1A*). Fibrillar collagen I but not a non-fibrillar laminin also stimulated secretion of CD63+/CD81 + sEVs by VSMCs to the same extent as FN (*Figure 1—figure supplement 1A*).

We next tested if the native VSMC-derived 3D matrix that contains FN and collagen could modulate sEV secretion (*Beacham et al., 2007*). VSMCs were induced to produce ECM, cells were removed and fresh VSMCs were plated onto these 3D matrices. VSMCs acquired an elongated shape (*Figure 1B and C*), which is typical for mesenchymal cells in a 3D environment (*Beacham et al., 2007*) and increased secretion of CD63+/CD81+ sEVs compared to VSMCs plated onto the non-coated plastic (*Figure 1D*). We observed no changes in the size distribution of sEVs secreted by VSMCs plated either on plastic or FN matrix as detected using Nanoparticle Tracking Analysis (NTA; *Figure 1E*). To further characterise the EV populations released, western blotting confirmed that sEVs isolated in both conditions were loaded with similar levels of FN and the sEV-specific markers CD63 and Syntenin-1 and lacked the EV-specific marker, α-actinin-4 (*Figure 1F*; *Kowal et al., 2016*). In addition, treatment of the cells plated onto the matrix with sphingomyelin phosphodiesterase 3 inhibitor (3-O-Methyl-Sphingomyelin, 3-OMS) revealed that CD63+/CD81+ sEV secretion in response to collagen and FN was regulated by sphingomyelin phosphodiesterase 3 (*Figure 1—figure supplement 1B*) which regulates sEV generation in multivesicular bodies (*Trajkovic et al., 2008*). Hence, FN triggers secretion of CD63+/CD81+ sEVs, most likely with a late endosomal origin.

Both collagen I and FN are ECM ligands that bind and transduce intracellular signalling via β1 integrin. Therefore, we explored the effect of β1 integrin activating (12G10) or inhibiting antibodies (4B4) on FN-stimulated secretion of CD63+/CD81+ sEVs. Activation of β1 integrin using 12G10 antibody in VSMCs enhanced the effect of FN on sEV secretion (*Figure 1G*). Inhibition of β1 integrin using the 4B4 antibody blocked CD63+/CD81+ sEV secretion by VSMCs plated on FN (*Figure 1G*). Next, we tested the role of the β1 integrin downstream signalling mediators, FAK and Src (*Klinghoffer et al., 1999*). Blocking these pathways with small inhibitors (FAM14 and PP2, respectively) reduced sEV secretion

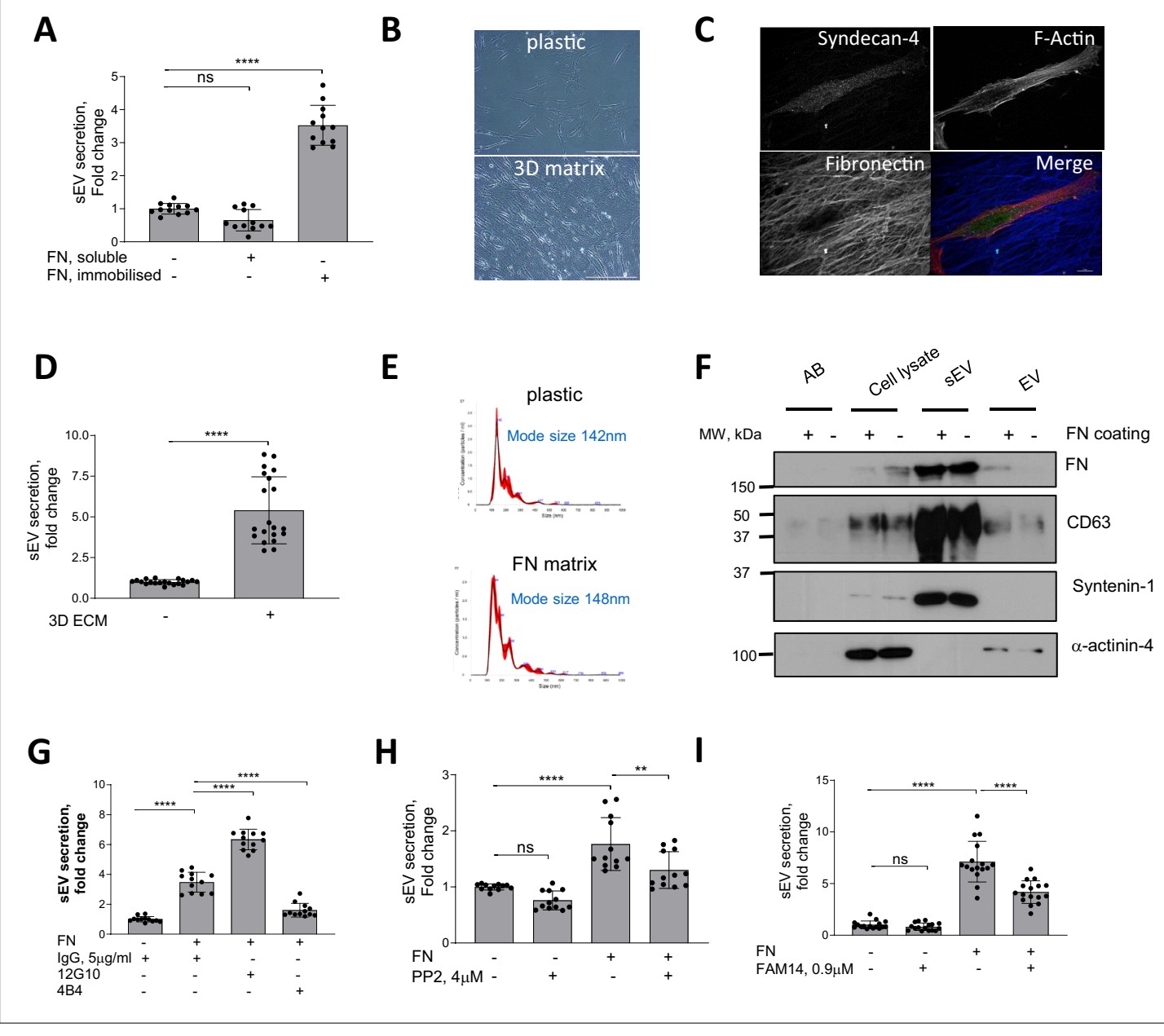

**Figure 1.** FN matrix stimulates sEV secretion by VSMCs in vitro. (**A**) Immobilised FN but not a soluble FN promotes sEV secretion. Cells were cultured for 24 hr and sEVs in conditioned media were measured by CD63-beads assay. N=3 biological replicates, n=4 technical replicates, ANOVA, ****p<0.0001 (**B**) Micrograph showing VSMC plated onto plastic or 3D matrix. Olympus CKX41 inverted microscope equipped with a 20 x objective and QImaging QIcam Fast 1394 camera. Size bar, 100 µm (estimated using manufacturer specifications for microscope and camera field of view). (**C**) VSMCs were plated on the 3D matrix for 24 hr, fixed and labelled for Syndecan-4 (green), F-actin (phalloidin, red), and fibronectin (blue). Size bar, 10 µm. (**D**) 3D matrix promotes sEV secretion. Cells were cultured for 24 hr and conditioned media was collected and sEV secretion was measured by CD63-beads assay. N=5, biological replicates, n=4 technical replicates, t-test (**E**) FN matrix does not affect sEV mode size. VSMCs were plated on non-coated or FN-coated flasks and incubated for 24 hr. Isolated sEVs were analysed by Nanoparticle Tracking Analysis. Representative from N=3 biological replicates. (**F**) sEV and EV markers distribution is not altered by FN matrix. Cells were plated on non-coated or FN-coated flasks and apoptotic bodies (AB, 1.2 K pellet), extracellular vesicles (EV, 10 K pellet) and small extracellular vesicles (sEVs, 100 K pellet) were isolated by differential ultracentrifugation and analysed by western blotting. Representative image from N=3 biological replicates. (**G**) FN induces secretion of sEVs by activating β1 integrin. VSMCs were plated on non-coated or FN-coated plates in the absence or presence of integrin activating (12G10) or inhibiting (4B4) antibodies for 24 hr and conditioned media was analysed by CD63-bead assay. N=3 biological replicates, n=4 technical replicates, ANOVA, ****p<0.0001 (**H**) Src is required for the sEV secretion. Cells were plated and sEV secretion was measured as in 2 A. N=3, biological replicates, n=4, technical replicates, ANOVA, **p<0.0001. (**I**) Inhibition of FAK blocks FN-induced sEV secretion. Cells were plated and sEV secretion was measured as in 1 A. N=3 biological replicates, n=4 technical replicates, ANOVA.

*Figure 1 continued on next page*

Figure 1 continued

The online version of this article includes the following figure supplement(s) for figure 1:

**Figure supplement 1.** FN matrix stimulates sEV secretion by VSMCs.

by cells plated on FN but not on plastic (*Figure 1H and I*). Taken together, these data suggest that FN matrices stimulate secretion of CD63+/CD81+ sEVs via the β1 integrin signalling pathway.

## FN-induced sEV secretion is modulated by Arp2/3 and formin-dependent actin cytoskeleton remodelling

ECM-induced focal FAs initiate dynamic remodelling of the actin cytoskeleton, a process regulated by the Arp2/3 complex and formins, which have recently been implicated in exosome secretion in lymphocytes and tumour cells (*Sinha et al., 2016*; *Ruiz-Navarro et al., 2024*; *Campellone et al., 2023*). So next, we tested the contribution of Arp2/3 and formins by using the small molecule inhibitors, CK666 and SMIFH2, respectively (*Nolen et al., 2009*; *Rizvi et al., 2009*). CK666 reduced sEV secretion both in VSMCs plated on plastic and FN matrix, while inhibition of formins using SMIFH2 was only effective on the FN matrix (*Figure 2A and B*), indicating that formins may be involved in FN-dependent sEV secretion.

As regulators of branched actin assembly, the Arp2/3 complex and cortactin are thought to contribute to sEV secretion in tumour cells by mediating MVB intracellular transport and plasma membrane docking (*Sinha et al., 2016*; *Taunton et al., 2000*). Therefore, we overexpressed the Arp2/3 subunit, ARPC2-GFP and the F-actin marker, F-tractin-RFP in VSMCs and performed live-cell imaging. As expected, Arp2/3 and F-actin bundles formed a distinct lamellipodia scaffold in the cellular cortex (*Figure 2—figure supplement 1A* and *Video 1*). Unexpectedly, we also observed numerous Arp2/3/F-actin positive spots moving through the VSMC cytoplasm that resembled previously described endosome actin tails observed in Xenopus eggs (*Taunton et al., 2000*) and parasite infected cells where actin comet tails propel parasites via filopodia to neighbouring cells (*Abella et al., 2016*; *Thomas et al., 2010*; *Figure 2—figure supplement 1A*, arrow, and *Video 1*). Analysis of the intracellular distribution of F-actin and CD63-positive endosomes in VSMCs showed CD63-MVB propulsion by the F-actin tail in live cells (*Figure 2C* and *Video 2*). Although these were rare events, possibly due to the short observational time in a single microscopic plane (*Figure 2C*), we observed numerous F-actin spots in fixed VSMCs that were positive both for F-actin and cortactin, indicating that these are branched-actin tails (*Figure 2D*). Moreover, cortactin/F-actin spots colocalised with CD63+ endosomes and addition of the SMPD3 inhibitor, 3-OMS, induced the appearance of enlarged doughnut-like cortactin/F-actin/CD63 complexes resembling invadopodia-like structures similar to those observed in tumour cells (*Figure 2D*, arrows; *Hoshino et al., 2013*). To quantify CD63 overlap with the actin tail-like structures, we extracted round-shaped actin structures and calculated the thresholded Manders colocalisation coefficient (*Figure 2—figure supplement 1B*). We observed overlap between F-actin tails and CD63 as well as close proximity of these markers in fixed VSMCs (*Figure 2—figure supplement 1B*). Approximately 50% of the F-actin tails were associated with ~13% of all endosomes (tM1=0.44 ± 0.23 and tM2=0.13 ± 0.06, respectively, N=3). Addition of 3-OMS enhanced this overlap further (tM1=0.75 ± 0.18 and tM2=0.25 ± 0.09) suggesting that Arp2/3-driven branched F-actin tails are involved in CD63 +MVB intracellular transport in VSMCs.

Formins and the Arp2/3 complex play a crucial role in the formation of filopodia, a cellular protrusion required for sensing the extracellular environment and cell-ECM interactions (*Mellor, 2010*). To test whether MVBs can be delivered to filopodia, we stained VSMCs for Myosin-10 (Myo10; *Bohil et al., 2006*). We observed no difference between total filopodia number per cell on plastic or FN matrices (n=18 ± 8 and n=14 ± 3, respectively); however, the presence of endogenous CD63+ MVBs along the Myo10-positive filopodia was observed in both conditions (*Figure 2E*, arrows). Filopodia have been implicated in sEV capture and delivery to endocytosis 'hot-spots' (*Heusermann et al., 2016*), so next we examined the directionality of CD63+ MVB movement in filopodia by overexpressing Myo10-GFP and CD63-RFP in live VSMCs. Importantly, we observed anterograde MVB transport toward the filopodia tip (*Figure 2F* and *Video 3*) indicative of MVB secretion.

We also attempted to visualise sEV release in filopodia using CD63-pHluorin, where fluorescence is only observed upon the fusion of MVBs with the plasma membrane (*Verweij et al., 2018*). Using

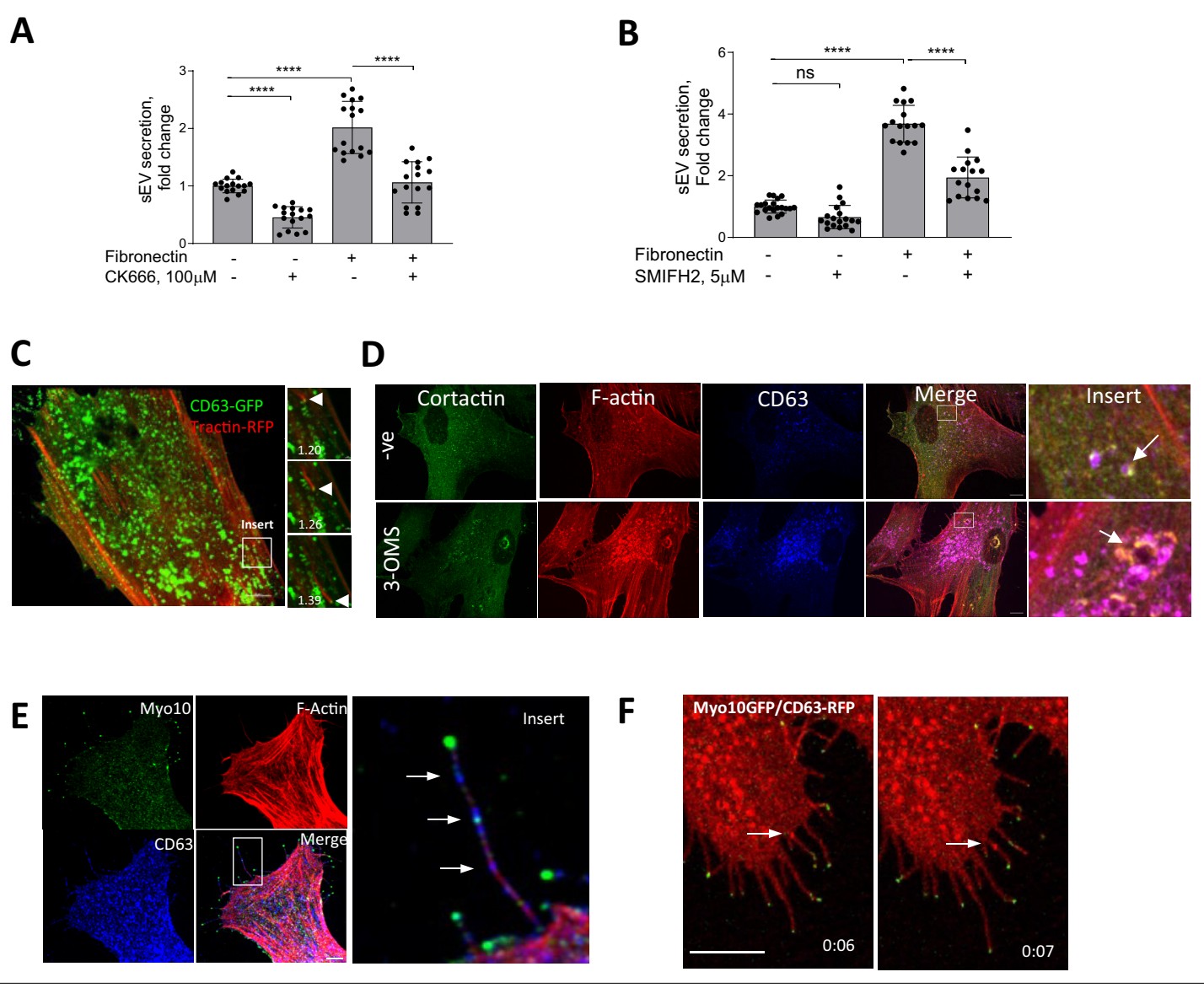

**Figure 2.** sEV secretion is regulated by Arp2/3 and formin-dependent actin cytoskeleton remodelling. (**A**) Arp2/3 inhibition with CK666 reduces sEV secretion in VSMC. Cells were plated onto non-coated or FN-coated plates for 24 hr and conditioned media was analysed by CD63-bead assay. N=4 biological replicates, n=4 technical replicates, ANOVA. (**B**) Formin inhibitor, SMIFH2 blocking filopodia formation reduces sEV secretion from FN-plated cells only. Cells were plated onto non-coated or FN-coated plates for 24 hr and conditioned media was analysed by CD63-bead assay. N=4–5 biological replicates, n=2–4 technical replicates, ANOVA. (**C**) Still images of a time-lapse showing that transported MVBs are attached to F-actin tails. VSMCs were co-transfected with CD63-GFP and F-tractin-RFP and time-lapse was captured using confocal spinning-disk microscopy. Arrow head – position of CD63 MVB across time. Time, min. Size bar, 10 μm. (**D**) VSMC were plated for 24 hr in the absence (top, -ve) or presence of sEV secretion inhibitor (3-OMS, 10 μM). Cells were fixed and stained for cortactin, F-actin and CD63. Note an overlap between CD63 endosomes and branched actin proteins (F-actin and cortactin), which is enhanced by 3-OMS treatment (arrow). Size bar, 10 μm. (**E**) CD63 MVBs (arrows) are detected in filopodia-like structures. VSMCs were plated on FN-coated plates for 24 hr and cells were stained for Myo10 (green), CD63 (blue), F-actin (phalloidin, red). Size bar, 10 μm. (**F**) Still images of a time-lapse showing that MVBs are transported to the filopodia tip in the live VSMC. Cells were co-transfected by CD63-RFP and Myo10-GFP, cultured for 24 hr and time-lapse video was captured using confocal spinning disk microscopy. Snapshots were taken at T=6 s and T=7 s after start of the video. Size bar, 10 μm.

The online version of this article includes the following figure supplement(s) for figure 2:

**Figure supplement 1.** Actin cytoskeleton involvement in CD63+ endosome transport and secretion.

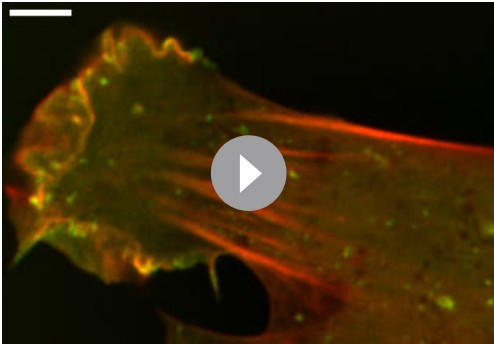

**Video 1.** Time-lapse imaging of cytosolic Arp2/3 complex (ARPC2–GFP) and F-actin (F-tractin–RFP) distribution in vascular smooth muscle cells at 24 hr post-transfection. Scale bar, 5 μm.
https://elifesciences.org/articles/90375/figures#video1

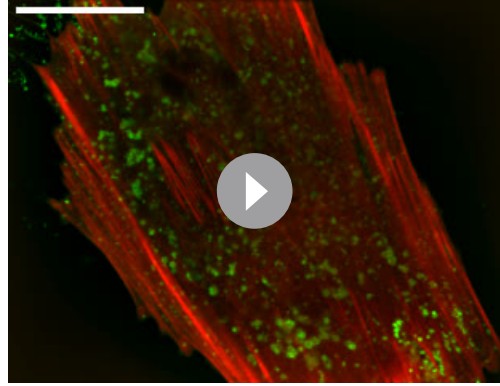

**Video 2.** Time-lapse imaging of cytosolic CD63 (CD63–GFP) and F-actin (F-tractin–RFP) in vascular smooth muscle cells at 24 hr post-transfection. Scale bar, 5 μm.
https://elifesciences.org/articles/90375/figures#video2

total internal reflection fluorescence microscopy (TIRF), we observed the typical 'burst'-like appearance of sEV secretion at the cell-ECM interface in full agreement with an earlier report showing MVB recruitment to invadopodia-like structures in tumor cells (*Hoshino et al., 2013*; *Figure 2—figure supplement 1C* and *Video 4*). Although we also observed an intense CD63-pHluorin staining along filopodia-like structures, we were not able to detect typical 'burst'-like events to confirm sEV secretion in filopodia (*Figure 2—figure supplement 1C* and *Video 4*).

Taken together, these data show that sEV secretion by VSMCs is regulated via Arp2/3 and formin-dependent actin cytoskeleton remodelling. Branched actin tails can potentially contribute to MVB intracellular transport and secretion at the VSMC-ECM interface. Interestingly, CD63+MVBs can be observed in filopodia-like structures, suggesting that sEV secretion can also occur spatially via cellular protrusion-like filopodia, but more studies are needed to confirm this hypothesis.

## sEVs are trapped in the ECM in vitro and in atherosclerotic plaque

EV trapping in the ECM is a prominent feature of the blood vessel media and intima, and EVs become more abundant in the ECM with ageing and disease (*Hutcheson et al., 2016*; *Whitehead et al., 2023*). Therefore, we next set out to determine if sEVs that are secreted toward the ECM can be trapped in the native matrix in vitro. VSMCs were plated on plastic or FN for 24 hr and sEVs were visualised by CD63 immunofluorescence. We observed CD63 puncta in proximity to protrusions at the cell periphery (*Figure 3A*). These

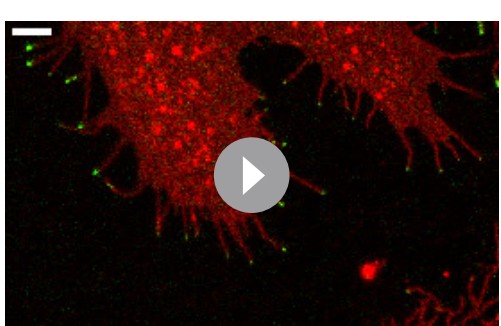

**Video 3.** Time-lapse imaging of CD63 (CD63–RFP) and Myosin-10 (Myo10–GFP) in vascular smooth muscle cells at 24 hr post-transfection. Scale bar, 5 μm.
https://elifesciences.org/articles/90375/figures#video3

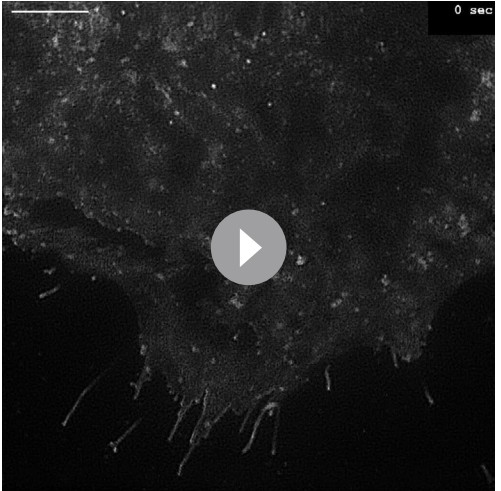

**Video 4.** Time-lapse imaging of CD63 (CD63–pHluorin) in vascular smooth muscle cells at 24 hr post-transfection. Scale bar, 5 μm.
https://elifesciences.org/articles/90375/figures#video4

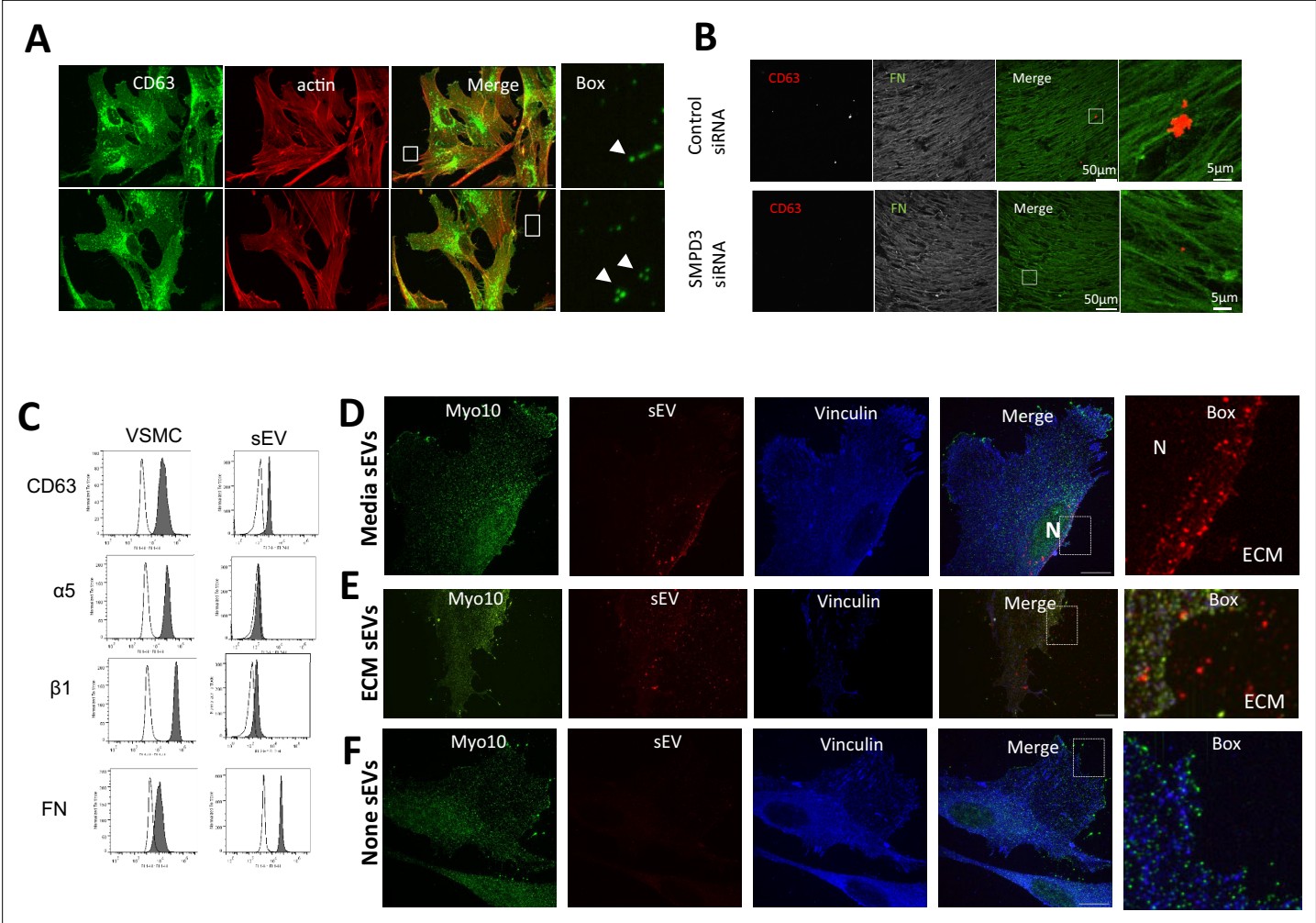

**Figure 3.** Endogenous and exogenous sEVs are trapped by ECM in vitro. (**A**) VSMC were plated onto non-coated (top) or FN-coated (bottom) plastic for 24 hr and stained for the membrane sEV marker, CD63 and F-actin. Note the accumulation of CD63 puncta in close proximity to filopodia-like projections. Size bar, 10 μm. (**B**) VSMCs were plated onto gelatin-coated plates and treated with control or SMPD3 siRNA for 72 hr. 3D matrices were generated and stained for CD63 and FN. Images were acquired using Nikon AX inverted confocal microscope. Note the decrease in CD63-positive sEVs associated with the FN fibrils. (**C**) FN is presented on the surface of the VSMC-derived sEV along with α5β1 integrin. VSMC sEVs were immobilised on the 4 μm beads. sEV-beads and VSMCs were stained with the antibodies (filled graphs) in non-permeabilised conditions and analysed by flow cytometry. (**D**) VSMC were plated on FN-coated dishes and Alexa568-labelled sEV were added to the cell media for 3 hr. Cells were fixed and stained for filopodia marker Myo10 (green) and vinculin (blue). Note perinuclear localisation of internalised sEVs. N, nucleus. ECM, extracellular matrix. Size bar, 10 μm. Representative image from N=3 biological replicates. (**E**) VSMC were plated on FN-coated dishes pre-coated with Alexa568-labelled sEV and incubated for 24 hr. Cell staining as in **D**. Note even distribution of sEVs across the matrix and cell area. Size bar, 10 μm. (**F**) VSMC were plated on the FN-coated dishes in the absence of Alexa568-labelled sEV and incubated for 24 hr. Cell staining as in **D**. Note the absence of signal in sEV channel. Size bar, 10 μm.

The online version of this article includes the following figure supplement(s) for figure 3:

**Figure supplement 1.** SMPD3-dependent sEVs are trapped in ECM.

CD63+ sEVs were observed both on the plastic and FN-coated plates, suggesting that sEVs can bind to the ECM secreted by VSMCs over the 24 hr incubation. In agreement with our previous data, we also observed CD63+ sEV trapping in the native ECM produced by VSMCs over a 7-day period and their reduction/absence from the matrix when VSMCs were treated with SMPD3 siRNA (*Figure 3B*; *Whitehead et al., 2023*).

To quantify ECM-trapped sEVs, we applied a modified protocol for the sequential extraction of extracellular proteins using salt buffer (0.5 M NaCl) to release sEVs which are loosely attached to ECM via ionic interactions, followed by 4 M guanidine HCl buffer (GuHCl) treatment to solubilise strongly bound sEVs (*Figure 3—figure supplement 1*; *Didangelos et al., 2010*). We quantified total

sEV and characterised the sEV tetraspanin profile in conditioned media, and the 0.5 M NaCl and GuHCl fractions using ExoView. The total particle count showed that EVs are both loosely bound and strongly trapped within the ECM. sEV tetraspanin profiling showed differences between these 3 EV populations. While there was close similarity between the conditioned media and the 0.5 M NaCl fraction with high abundance of CD63+/CD81+ sEVs as well as CD63+/CD81+/CD9+ in both fractions (*Figure 3—figure supplement 1*). In contrast, the GuHCl fraction was particularly enriched with CD63+ and CD63+/CD81+ sEVs with very low abundance of CD9+ EVs (*Figure 3—figure supplement 1*). The abundance of CD63+/CD81+ sEVs was confirmed independently by a CD63+ bead capture assay in the media and loosely bound fractions (*Figure 3—figure supplement 1*).

We previously found that the serum protein prothrombin binds to the sEV surface both in the media and MVB lumen, showing it is recycled in sEVs and catalyses thrombogenesis being on the sEV surface (*Kapustin et al., 2017*). So we investigated whether FN can also be associated with sEV surface where it can be directly involved in sEV-cell cross-talk (*Kapustin et al., 2017*). We treated serum-deprived primary human aortic VSMCs with FN-Alexa568 and found that it was endocytosed and subsequently delivered to early and late endosomes together with fetuin A, another abundant serum protein that is a recycled sEV cargo and elevated in plaques (*Figure 3—figure supplement 1C and D*). CD63 visualisation with a different fluorophore (Alexa488) confirmed FN colocalisation with CD63+MVBs (*Figure 3—figure supplement 1E*). Next, we stained non-serum deprived VMC cultured in normal growth media supplemented with serum (M199 supplemented with 2.5% sEV-free FBS) with an anti-FN antibody and observed colocalisation of CD63 and serum-derived FN. Co-localisation was reduced, likely due to competitive bulk protein uptake by non-deprived cells (*Figure 3—figure supplement 1F*). Notably, when we compared FN distribution in sparsely growing VSMCs versus confluent cells, we found that FN intracellular spots, as well as colocalisation with CD63, completely disappeared in the confluent state (*Figure 3—figure supplement 1F and G*). This correlated with nearly complete loss of CD63+/CD81+sEV secretion by the confluent cells, indicating that confluence abrogates intracellular FN trafficking as well as sEV secretion by VSMCs (*Figure 3—figure supplement 1H*). Finally, FN could be co-purified with sEVs from VSMC conditioned media (*Figure 3—figure supplement 1I*) and detected on the surface of sEVs by flow cytometry confirming its loading and secretion via sEVs (*Figure 3C*).

To understand how VSMCs interact with sEVs in the ECM versus in the media, we isolated and fluorescently labelled sEVs and added them either to the ECM or the media and tracked sEV distribution. Super-resolution microscopy (iSIM) revealed that the addition of sEVs to the cell media resulted in fast uptake by VSMCs and delivery to the nuclear periphery (*Figure 3D*, *Figure 3—figure supplement 1J*). This was particularly obvious in 3D projections (*Figure 3—figure supplement 1J*). In contrast, purified sEVs added to FN-coated plates resulting in their even distribution across the ECM were not internalised even 24 hr after cell addition with no perinuclear localisation of ECM-associated sEVs observed (*Figure 3E*, *Figure 3—figure supplement 1K*). No red fluorescence was observed in the absence of added sEVs (*Figure 3F*).

To spatially map the accumulation of sEV markers in the atherosclerotic plaque in vivo, we collected 12 carotid atherosclerotic plaques and adjacent intact vascular segments excised during carotid endarterectomy. Unbiased proteomics analysis confirmed considerable differences between these 2 sample groups, revealing 213 plaque-specific and 111 intact arterial-specific proteins (*Figure 4A and B*, *Supplementary file 1*; *Supplementary file 2*; *Supplementary file 3*; *Supplementary file 4*; *Supplementary file 5*). Differential expression analysis identified 46 proteins significantly overexpressed (fold change ≥2 and FDR-adjusted p-value ≤0.05) in plaques and 13 proteins significantly upregulated in the intact arterial segments (*Figure 4C*, *Supplementary file 3*). Among the top proteins differentially expressed in plaques were catalytic lipid-associated bioscavenger paraoxonase 1, atherogenic apolipoprotein B-100, HDL-associated apolipoprotein D, iron-associated protein haptoglobin, and inflammation-related matrix metalloproteinase-9 (*Supplementary file 3*). These proteins have previously been implicated in lipid metabolism alterations as well as inflammation in the plaque, hence indicating the advanced atherosclerotic plaque signature of the analysed samples. Comparison of the differentially expressed proteins also revealed an accumulation in atherosclerotic regions of fetuin-A (Alpha-2-HS-glycoprotein, P02765, 2.7-fold increase, p=0.003869659), an abundant sEV cargo protein which is recycled by VSMCs (*Kapustin et al., 2015*). Likewise, the level of another sEV cargo, Apolipoprotein B-100 (P04114) was also significantly elevated (2.8-fold increase, p=0.000124446)

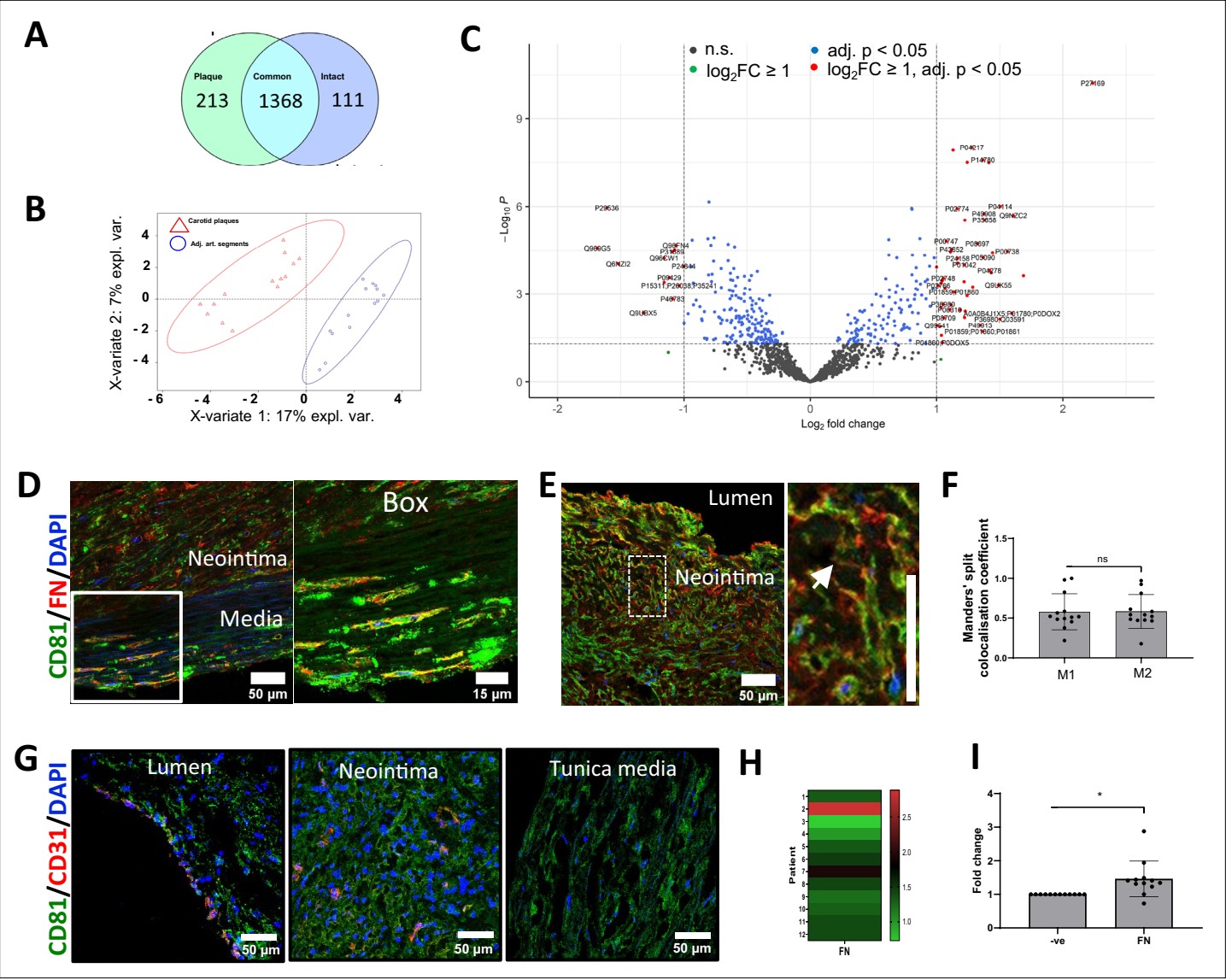

**Figure 4.** Fibronectin deposition in the atherosclerotic plaque is spatially associated with sEV markers. (**A**) Proteomic profiling of pairwise collected carotid atherosclerotic plaques and adjacent intact arterial segments. A Venn diagram shows the number of plaque-specific (n=213) and intact-specific (n=111) proteins as well as the number of proteins which are common for both vascular regions (1368). (**B**) Partial least-squares discriminant analysis indicates clear classification pattern between the carotid plaques (indicated by red triangles) and adjacent intact arterial segments (indicated by blue circles). N=14 patients. (**C**) Volcano plot illustrates that 46 proteins are significantly overexpressed in plaques whilst 13 proteins are significantly upregulated in adjacent intact arterial segments. (**D, E**) Atherosclerotic plaques were co-stained for fibronectin (FN) and sEV marker, CD81. Cell nuclei were counterstained with DAPI. Main figure: x200 magnification, size bar, 50 µm. Box: x400 magnification, size bar, 15 µm. Note an accumulation of FN in the neointima. (**F**) Spatial distribution of FN and CD81 in the neointima. Note high overlap between FN and CD81 in the extracellular matrix. x200 magnification, size bar, 50 µm. Manders' split colocalisation coefficient for the overlap of FN with CD81 (M1) and CD81 with FN (M2). Neointima region as in **E**. (**G**) Co-localisation of endothelial cells (stained for CD31/PECAM1, red colour) and sEVs (stained for CD81, green colour). Nuclei are counterstained with 4',6-diamidino-2-phenylindole (DAPI, blue colour). Magnification: ×200, scale bar: 50 µm. Note endothelialised (i.e. those covered with CD31-positive cells) segments at the luminal side and abundant CD81 expression both in the neointima and in the tunica media. Endothelial cells (CD31, red colour) are largely co-localised with CD81-positive extracellular vesicles (green colour). Note CD31-positive capillaries in the neointima. (**H, I**) Quantification of FN content in atherosclerotic plaques. Samples were analysed by western blot and band intensity was quantified in ImageJ. Fold change was calculated as the ratio of band intensity in the atherosclerotic plaque to band intensity in the adjacent intact arterial segments normalised to GAPDH. Note that FN content is elevated in atherosclerotic plaques relative to the adjacent intact arterial segments. Paired t-test.

The online version of this article includes the following figure supplement(s) for figure 4:

**Figure supplement 1.** Fibronectin deposition in the atherosclerotic plaque is spatially associated with sEV marker CD81.

in atherosclerotic plaque (*Supplementary file 2*; *Kapustin et al., 2015*). Notably, a negative regulator of sEV secretion, Coronin 1C (Q9ULV4) was downregulated in the plaque (0.6-fold decrease, p=0.000805372, *Supplementary file 2*) consistent with previous studies that suggested that sEV secretion is upregulated during plaque progression (*Hutcheson et al., 2016*; *Tagliatela et al., 2020*).

To test if FN associates with sEV markers in atherosclerosis, we investigated the spatial association of FN with sEV markers using the sEV-specific marker CD81. Staining of atherosclerotic plaques with haematoxylin and eosin revealed well-defined regions with the neointima as well as tunica media layers formed by phenotypically transitioned or contractile VSMCs, respectively (*Figure 4—figure supplement 1*). Masson's trichrome staining of atherosclerotic plaques showed abundant haemorrhages in the neointima and sporadic haemorrhages in the tunica media (*Figure 4—figure supplement 1*). Staining of atherosclerotic plaques with orcein indicated weak connective tissue staining in the atheroma with a confluent extracellular lipid core, and strong specific staining at the tunica media containing elastic fibres which correlated well with the intact elastin fibrils in the tunica media (*Figure 4—figure supplement 1*). Using this clear morphological demarcation, we found that FN accumulated both in the neointima and the tunica media where it was significantly colocalised with the sEV marker, CD81 (*Figure 4D, E and F*). Notably, CD81 and FN colocalisation was particularly prominent in cell-free, matrix-rich plaque regions (*Figure 4E and F*). Interestingly, colocalisation analysis of CD81 and FN distribution showed that there was a trend of higher colocalisation levels of CD81 with FN in the neointima as compared to tunica media, which was not significant (Thresholded Mander's split colocalisation coefficient for CD81 colocalisation with FN 0.58±0.21 and 0.49±0.14, respectively, N=13, p=0.052, paired T test). An enhanced expression of CD81 by endothelial cells in early atheroma has been previously reported so to study the contribution of CD81+ sEVs derived from endothelial cells, we investigated the localisation of CD31 and CD81 (*Rohlena et al., 2009*). In agreement with a previous study, we found that the majority of CD31 colocalises with CD81 (Thresholded Mander's split colocalisation coefficient 0.54±0.11, N=6) indicating that endothelial cells express CD81 (*Figure 4G*; *Rohlena et al., 2009*). However, only a minor fraction of total CD81 colocalised with CD31 (Thresholded Mander's split colocalisation coefficient 0.24±0.06, N=6) confirming that the majority of CD81 in the neointima is originating from the most abundant VSMCs. Finally, western blot analysis confirmed FN accumulation in the plaque region (*Figure 4H and I*, *Figure 4—figure supplement 1E*). To understand the origin of plaque FN, we performed RT-PCR. FN expression could neither be detected in plaques nor in intact vessels, although it was abundant in the liver (*data not shown*), suggesting that the circulation is a key source of FN in atherosclerotic plaques.

## sEVs stimulate VSMC directional invasion and induce early peripheral focal adhesion assembly with an enhanced pulling force

FN as a cargo in sEVs promotes FA formation in tumour cells and increases cell speed (*Sung et al., 2015*; *Sung and Weaver, 2017*). As we found that FN is loaded into VSMC-derived sEVs, we hypothesised that ECM-entrapped sEVs can enhance cell migration by increasing cell adhesion and FA formation in the context of a FN-rich ECM. Therefore, we tested the effect of sEV deposition onto the FN matrix on VSMC migration in 2D and 3D models. We found that FN coating promoted VSMC velocity and inhibition of bulk sEV secretion with 3-OMS reduced VSMC speed in a 2D single-cell migration model (*Figure 5A and B*) in agreement with previous studies using tumour cells (*Sung et al., 2015*; *Sung and Weaver, 2017*). However, the addition of sEVs to the ECM had no effect on VSMC speed at baseline but rescued cell speed and distance in the presence of the sEV secretion inhibitor, 3-OMS, suggesting the EVs are not primarily regulating cell speed (*Figure 5A and B*).

To assess the effect of sEVs on cell directionality, we exploited a 3D model developed by *Sung and Weaver, 2017* and examined VSMC invasion using a μ-Slide Chemotaxis assay where cells were embedded into FN-enriched 3D Matrigel matrices in the absence or presence of exogenously-added sEVs (*Figure 5C*). We developed a script to automatically measure cell invasion parameters (track length, cell speed, straightness, accumulated distance, lateral and vertical displacement and parallel forward motion index [FMI]; https://github.com/Alex-biochem/eLife-VOR-RA-2023-90375, copy archived at *Kapustin, 2025*). The addition of sEVs had no effect on cell speed, accumulated distance, or straightness (*Figure 5D, E and F*). However, VSMCs invading in the presence of embedded sEVs migrated more aligned to the FBS gradient compared to cells plated in the absence of sEVs. To test whether this was a feature of all EV populations, we also performed the experiment in the presence of

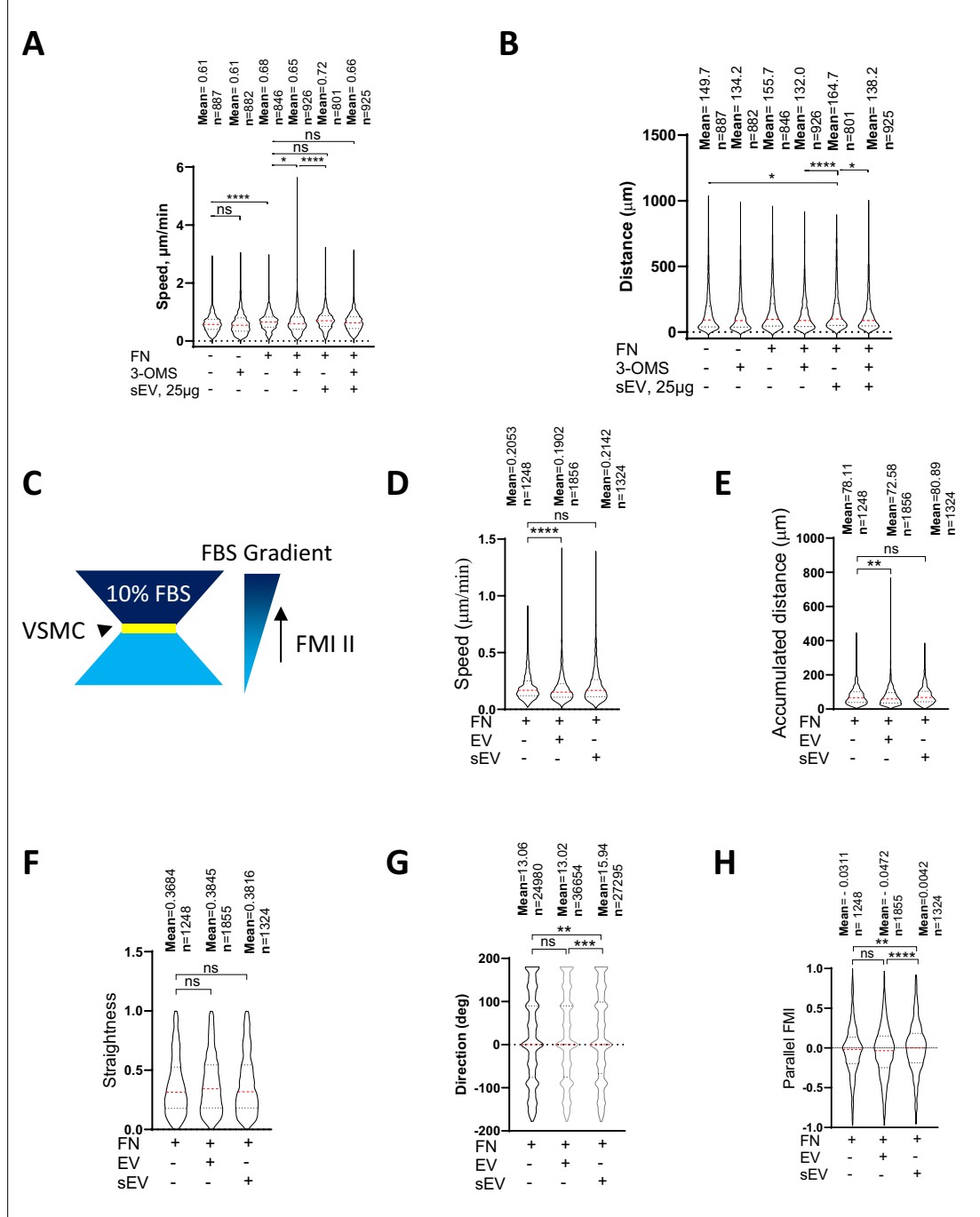

**Figure 5.** sEV induces directional VSMC invasion. (**A, B**) VSMC migration in 2D assay. VSMC were plated onto FN in the absence or presence of SMPD3 inhibitor (3-OMS) and/or sEV (25 µg). Cells were tracked for 8 hr. Kruskal-Wallis with Dunn's multiple comparison test, N=2–3 biological replicates, n indicates the total number of tracked cells per condition, **, p<0.01, ****, p<0.0001. (**C**) Chemotaxis µ-slide diagram. Yellow, cell chamber, blue chemoattractant-free medium chamber, dark blue – chemoattractant medium chamber. (**D-H**) sEV promote directional VSMC invasion. Cells were seeded to the FN-enriched Matrigel matrix in µ-Slide Chemotaxis assay and stained with Draq5. Cell tracking was conducted by OperaPhenix microscope for 12 hr and cell invasion parameters were quantified using Columbus. Kruskal-Wallis with Dunn's multiple comparison test, N=4, biological replicates, n indicates the total number of tracked cells per condition, **, p<0.01, ***, p<0.001, ****, p<0.0001.

larger EVs pelleted at the 10,000x *g* centrifugation step and observed no effect suggesting this mechanism was specific to sEVs (*Figure 5G and H*). Hence, ECM-associated sEVs have modest influence on VSMC speed but influence VSMC invasion directionality.

We wondered whether VSMC migration and invasion were affected by the FN cargo in sEVs that was shown previously to promote tumour cell migration by enhancing FA assembly and cellular adhesion to FN (*Sung et al., 2015*). Therefore, we measured VSMC adhesion using an adhesion assay where sEVs were added to FN-coated plates and cells were plated onto these matrices for 1 hr and firmly attached cells counted. Surprisingly, we observed a marked reduction of VSMC adhesion to FN in the presence of sEVs (*Figure 6A*), suggesting that ECM-associated sEV can impair VSMC adhesion. Mesenchymal cell attachment to the matrix is mediated by transient focal complexes present in the cellular protrusion as well as centripetal FAs that link the cellular cytoskeleton to the matrix and mediate the pulling force to move the cell body (*Sheetz et al., 1998*; *Ridley, 2001*). Consistently, we observed focal complexes in cellular protrusions as well as more centripetal FAs which were associated with mature actin stress fibres in VSMCs plated onto FN matrix (*Figure 6—figure supplement 1A*). Therefore, we used a live cell spreading assay to monitor cellular adhesion over time using ACEA's xCELLigence. As expected, FN alone stimulated VSMC spreading and adhesion (*Figure 6B and C*). Addition of sEVs did not impact cellular adhesion over the first ~15 min but then the adhesion was stalled in the presence of sEVs (*Figure 6B and C*), suggesting a defect in further cellular spreading. Again, to test whether this was specific for the sEV subset, we also tested the effect of larger EVs pelleted at 10,000 x *g* and found that these EVs had no effect on VSMC adhesion and spreading onto the FN matrix (*Figure 6B and C*). To interrogate the stalling event induced by sEVs in VSMCs, we plated cells and counted the number of FAs, average size as well as distance from the cell periphery after 30 min using TIRF microscopy. Once again, plating VSMCs onto FN increased the cell size, number of FAs as well as the number of centripetal FAs as compared to cells spreading over non-coated plastic (*Figure 6E–I*). Importantly, the addition of sEVs to FN significantly reduced the number of FAs, and the newly formed FAs were formed in close proximity to the cell periphery (*Figure 6E and F*). Interestingly, the average FA size was similar between the various conditions (*Figure 6I*).

Cellular traction force is generated by the FA gradient upon FA turnover - formation at the leading edge and disassembly at the tail (*Sheetz et al., 1998*; *Ridley, 2001*; *Burridge, 2005*). To test if sEVs can influence FA turnover in migrating VSMCs, we examined FA turnover in the presence of sEVs. Adhesion sites were visualised by paxillin-RFP reporter expression and the FA turnover index was calculated by counting FA overlap over time using a previously developed algorithm to track individual FAs (*Figure 6—figure supplement 1B*; *Holt et al., 2008*). Interestingly, the FA turnover index remains the same in the presence of sEVs, indicating that FA stability was not altered (*Figure 6J*). These data correlate well with no changes observed in FA average size across various conditions (*Figure 6I*) and indicate that sEVs are not influencing FA assembly or 'life-cycle' directly. Rho-dependent activation of actomyosin contractility stabilises FAs by mechanical forces halting fibroblast spreading onto FN (*Chrzanowska-Wodnicka and Burridge, 1996*; *Arthur and Burridge, 2001*). We hypothesised that sEVs trigger the appearance of premature, peripheral FAs by activating Rho-dependent cellular contractility. To test whether sEVs can modulate mature FA contractility, we measured individual traction force which is generated by mature adhesion sites in the absence or presence of sEVs (*Figure 6—figure supplement 1C*). VSMCs transfected with paxillin-RFP were plated on FN and sEV-covered PDMS pillars and pillar displacements were calculated as previously described (*Tabdanov et al., 2015*; *Schoen et al., 2010*; *Ghibaudo et al., 2008*). Importantly, the traction force was increased in the presence of sEVs (*Figure 6K*). Altogether, these data suggest that ECM-associated sEVs can trigger the formation of FAs with enhanced pulling force at the cell periphery.

## The sEV cargo Collagen VI regulates focal adhesion formation in VSMCs and cell invasion

To identify the components triggering peripheral FA formation, we compared the proteomic composition of sEVs with the larger EV fraction, which had no effect on FA formation. We identified 257 proteins in sEVs and 168 proteins in EVs with 142 proteins common between both datasets (*Figure 7A*, *Supplementary files 6-8*, *Figure 7—figure supplement 1*). Functional enrichment analysis revealed that the top 5 clusters were related to cell migration - ECM organisation, movement of cell or subcellular component, cell adhesion, leukocyte migration, and cell-cell adhesion (*Figure 7B*,

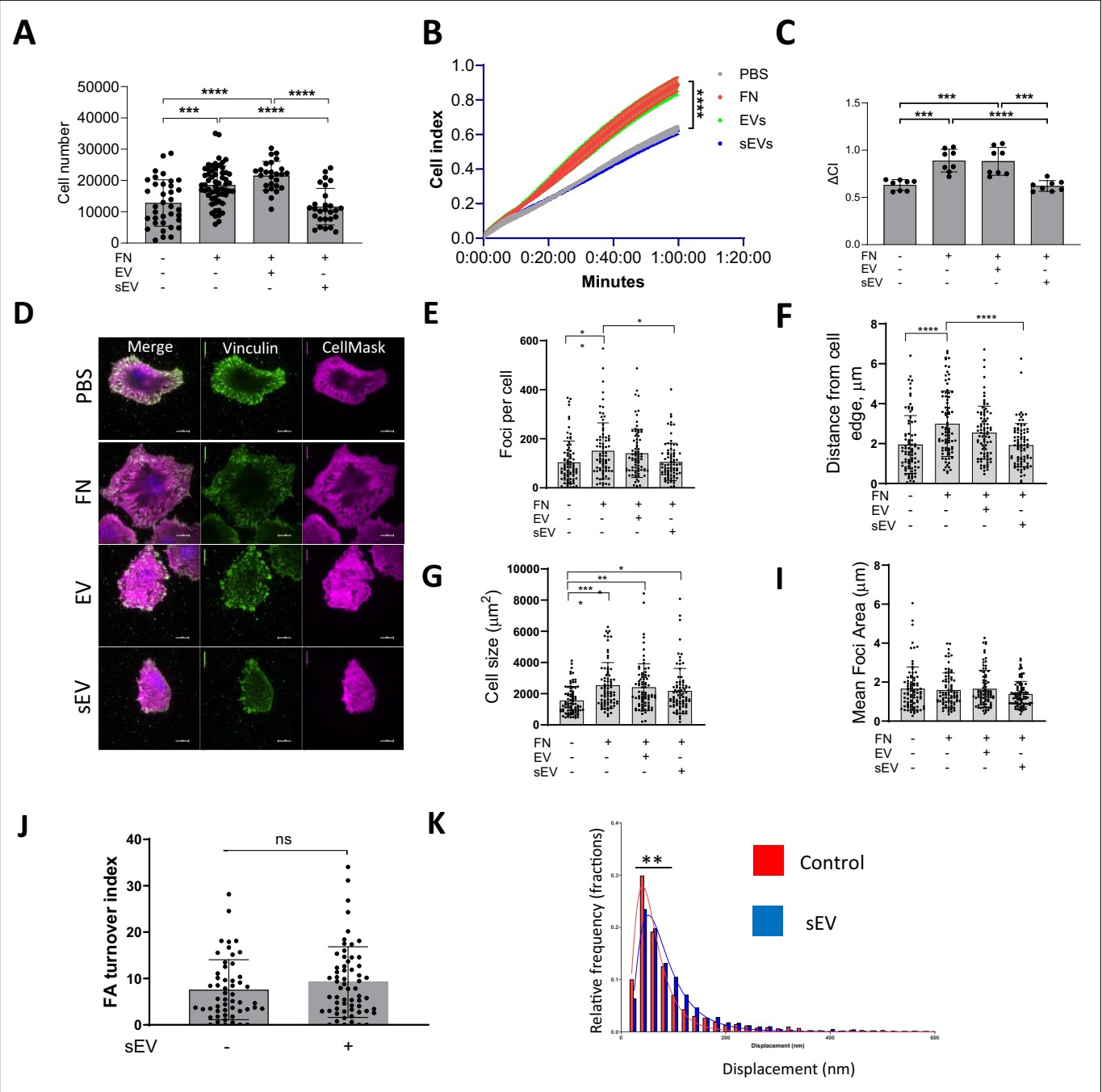

**Figure 6.** sEVs induce formation of peripheral FAs. (**A**) VSMC were plated on FN matrix for 30 min and adhered cells were counted by crystal violet staining. N=6 biological replicates, n=4–10 technical replicates, ANOVA, ***p<0.001, ****, p<0.0001 (**B, C**) VSMC spreading onto FN was tracked by using ACEA's xCELLigence Real-Time Cell Analysis. Note that the FN matrix promoted VSMC adhesion, but addition of sEVs inhibited cell spreading. N=3 biological replicates, n=2–3 technical replicates, ANOVA. (**D**) VSMCs were spread onto FN for 30 min and cells were fixed and stained with CellMask (magenta) and vinculin (green). Size bar, 10 μm. (**E, F, G, I**) Quantification of FA number, distance from plasma membrane, cell size and mean FA size per cell, respectively. FA were stained as in 5D and quantified. Representative data from N=3 biological replicates, n=80–82 total number of tracked cells per condition, ANOVA, *p<0.05, **p<0.01. (**J**) Focal adhesion turnover is not affected by sEVs. VSMC were transfected with Paxillin-RFP and plated on the FN in the absence or presence of immobilised sEVs. Images were captured for 30 min using confocal spinning disk microscopy and FA turnover was quantified using extracted images analysis, N=4, total n=53–61 cells per condition, Unpaired T-test. (**K**) sEV induces formation of strong-pulling FAs. VSMC transfected with Paxillin-RFP were plated on the PDMS pillars which were covered with FN and sEVs and pillar displacements were quantified. **p<0.01, Unpaired t-test, Data represent 2170–2322 measurements per condition pooled from two independent biological replicates.

The online version of this article includes the following figure supplement(s) for figure 6:

**Figure supplement 1.** sEVs regulate VSMC motility and invasion.

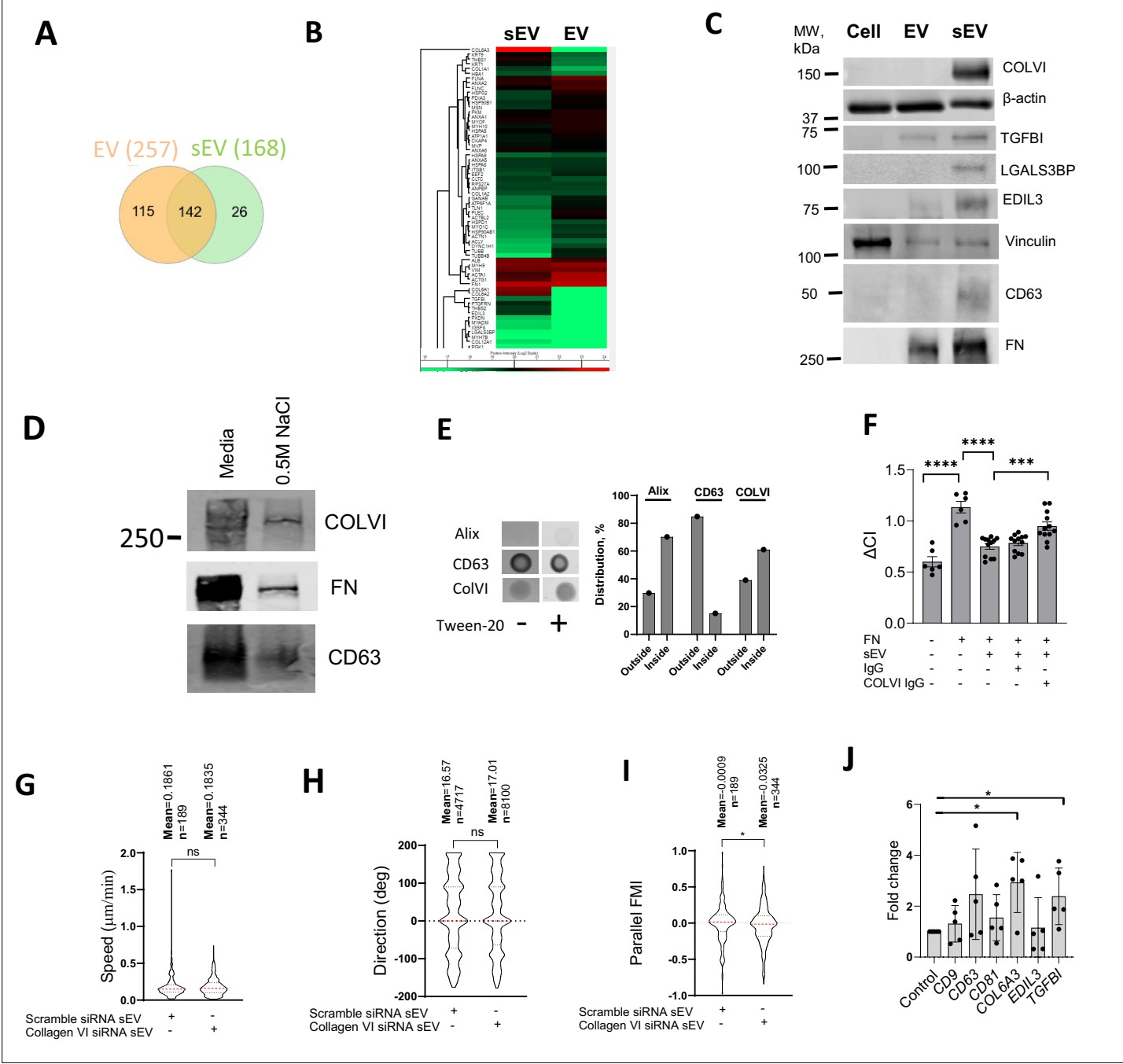

**Figure 7.** sEVs block focal adhesion formation by presenting collagen VI. (**A**) Proteomic analysis of VSMC-derived sEVs and EVs. Venn diagram. N=3 biological replicates. (**B**) Protein enrichment in the EV and sEV proteome. Heat Map. N=3 biological replicates. (**C**) Western blot validation of sEV cargos. EVs and sEVs were isolated from VSMC conditioned media by differential ultracentrifugation and analysed by western blotting. Representative image from N=3 biological replicates. (**D**) Collagen VI was presented in sEVs from conditioned media and in ECM-associated sEVs. sEVs were isolated from conditioned media and 0.5 M NaCl ECM fraction by differential ultracentrifugation and analysed by western blotting. (**E**) Collagen VI is presented on the surface of sEVs. sEVs were analysed by dot-blot in the non-permeabilised or permeabilised (PBS-0.2%-tween20) conditions. Staining intensity was quantified for non-permeabilised (outside) and permeabilised (inside) conditions. Representative from N=2 biological replicates. (**F**) VSMC adhesion is regulated by collagen VI loaded on sEV. FN matrices were incubated with sEV and anti-collagen VI antibody (COLVI IgG, 50 µg/ml) or control IgG (50 µg/ml). Cell adhesion was tracked by using ACEA's xCELLigence Real-Time Cell Analysis. ANOVA, N=3 biological replicates, ***, p<0.001, ****, p<0.0001. (**G, H, I**) sEVs promote directional VSMC invasion. VSMCs were treated with control siRNA (Scramble) or collagen VI-specific siRNA pools for 24 hr and were seeded to the FN-enriched Matrigel matrix in a µ-Slide Chemotaxis assay and stained with Draq5. Cell tracking was conducted by OperaPhenix microscope for 12 hr and cell invasion parameters were quantified using Columbus, n indicates the total number of tracked cells per condition,

*Figure 7 continued on next page*

*Figure 7 continued*

Kolmogorov-Smirnov test, *, p<0.05. (**J**) Real-time PCR analysis of expression of CD9, CD63, CD81, COL6A3, EDIL3, and TGFBI in atherosclerotic plaque. *, p<0.05, Paired t-test, N=5 patients.

The online version of this article includes the following figure supplement(s) for figure 7:

**Figure supplement 1.** Proteomic analysis of VSMC-derived sEVs and EV.

*Figure 7—figure supplement 1B*, *Supplementary file 8*). The cell adhesion cluster included collagen VI (chains COL6A1, COL6A2, and COL6A3), extracellular matrix glycoproteins (FN and thrombospondin, THBS2, THBS1) as well as EGF-like repeat and discoidin I-like domain-containing protein (EDIL3) and transforming growth factor-beta-induced protein ig-h3 (TGFBI; *Figure 7B*, *Figure 7—figure supplement 1B*, *Supplementary file 8*). Notably, these sEV proteins are predominantly involved in cell-matrix interactions and cellular adhesion. The lectin galactoside-binding soluble 3 binding protein (LGALS3BP), which regulates cell adhesion and motility, was also presented exclusively in sEVs (*Supplementary file 6*). These cell adhesion modulating proteins, including Collagen VI, TGFBI, EDIL, and LGALS3BP, were either uniquely or highly enriched in sEVs compared to larger EVs as detected by western blotting (*Figure 7C*).

Collagen VI was the most abundant protein in VSMC-derived sEVs (*Figure 7B*, *Supplementary file 7*) and was previously implicated in the interaction with the proteoglycan NG2 (*Tillet et al., 2002*) and suppression of cell spreading on FN (*Nishida et al., 2018*). To confirm the presence of collagen VI in ECM-associated sEVs, we analysed sEVs extracted from the 3D matrix using 0.5 M NaCl treatment and showed that both collagen VI and FN are present (*Figure 7D*). Next, we analysed the distribution of collagen VI using dot-blot. Alix staining was bright only upon permeabilisation of sEV, indicating that it is preferentially a luminal protein (*Figure 7E*). On the contrary, CD63 staining was similar in both conditions, showing that it is a surface protein (*Figure 7E*). Interestingly, collagen VI staining revealed that 40% of the protein is located on the outside surface with 60% in the sEV lumen (*Figure 7E*). To test the role of collagen VI in sEV-induced changes in VSMC spreading, we incubated FN-deposited sEVs with an anti-collagen VI antibody to block its interaction with the proteoglycan NG2 (*Sardone et al., 2016*). Addition of the anti-collagen VI antibody restored VSMC spreading on the FN matrix as compared to non-specific IgG treatment (*Figure 7F*). Moreover, sEVs isolated from VSMCs after collagen VI knockdown using siRNA had no effect in the 3D invasion model on cell speed (*Figure 7G*) or direction (*Figure 7H*) but reduced cell alignment to the gradient as compared to VSMCs treated with a scrambled siRNA control (*Figure 7I*). Finally, we compared the expression of collagen VI, EDIL3, and TGFBI between the plaque and control aorta tissues (*Supplementary file 1*). Importantly, collagen VI and TGFBI expression were significantly elevated in the plaque (*Figure 7J*), consistent with increased deposition of sEVs during disease progression. Altogether, these data indicate that sEVs can induce early formation of mature peripheral FAs by presenting collagen VI, thus acting to modulate cell directionality upon the invasion (Graphical Abstract).

## Discussion

VSMC migration and invasion to the site of vascular injury is crucial for vessel repair as well as the pathogenic development of atherosclerotic plaque; however, the mechanisms of effective invasion through the complex vascular ECM meshwork have been poorly studied. Here, we show that FN, a novel marker of unstable plaques (*Langley et al., 2017*), enables VSMC migration and invasion by stimulating secretion of collagen VI-loaded sEVs which decorate the ECM and stimulate FA maturation. Notably, our data are consistent and extend previous reports showing that EV secretion by tumour cells enhances nascent adhesion formation, hence facilitating tumour cell migration and invasion (*Sung et al., 2015*; *Sung and Weaver, 2017*; *Sung et al., 2020*). In particular, we found that sEVs induce formation of FAs with an enhanced pulling force at the cell periphery, thus favouring actomyosin-mediated contractility which is essential for cell body propulsion and locomotion.

Mesenchymal cell migration begins with protrusive activity at the leading edge followed by the forward movement of the cell body (*Ridley, 2001*; *Arthur and Burridge, 2001*; *Pankova et al., 2010*). Cell directionality is guided via the integrin β1 'probe' on the tip of cellular protrusions, filopodia, and lamellipodia (*Galbraith et al., 2007*). Cellular protrusions are attached to the ECM via focal complexes and extended by the physical force generated by the branched network of actin filaments

beneath the plasma membrane (*Pollard and Borisy, 2003*). Focal complexes either disassemble or mature into the elongated centripetally located FAs (*Chrzanowska-Wodnicka and Burridge, 1996*). In turn, these mature FAs anchor the ECM to actin stress fibres and the traction force generated by actomyosin-mediated contractility pulls the FAs rearward and the cell body forward (*Sheetz et al., 1998*; *Ridley, 2001*). Here, we report that β1 integrin activation triggers sEV release followed by sEV entrapment by the ECM. Curiously, we observed CD63+ MVB transport toward the filopodia tips as well as inhibition of sEV-secretion with filopodia formation inhibitors, suggesting that sEV secretion can be directly linked to filopodia, but further studies are needed to define the contribution of this pathway to the overall sEV secretion by cells. To gain insight into how sEVs can stimulate cell invasiveness, we explored VSMC protrusive activity using live cell spreading assays and TIRF imaging. FN stimulated VSMC spreading by increasing the number of FAs, which were formed centripetally from the cell plasma membrane. FN-associated sEV dramatically ceased VSMC spreading by inducing early formation of mature FAs at the cell periphery. Cell spreading and adhesion are orchestrated by the Rho family of small GTPases, Cdc42, Rac1, and Rho, with Cdc42 and Rac1 modulating filopodia and lamellipodia and focal complex formation and Rho controlling FA maturation and actomyosin-mediated contractility (*Nobes and Hall, 1995*; *Ridley and Hall, 1992*; *Price et al., 1998*). Interestingly, in fibroblasts, Rho is specifically degraded in cellular protrusions and cellular spreading on FN is accompanied by transient Rho suppression during the fast cell spreading phase (*Arthur and Burridge, 2001*; *Ren et al., 1999*; *Wang et al., 2003*). This phase is followed by Rho activation leading to formation of actin stress fibres, tension increase on the focal complexes and their maturation into elongated centripetally located FAs (*Chrzanowska-Wodnicka and Burridge, 1996*; *Arthur and Burridge, 2001*; *Ren et al., 1999*; *Rottner et al., 1999*). Excessive Rho activation leads to premature FA assembly and stress fibre formation, resulting in reduced cellular protrusions, inhibition of cellular spreading and motility on the FN matrix (*Arthur and Burridge, 2001*; *Ren et al., 1999*; *Roberts et al., 2003*; *Nobes and Hall, 1999*; *Cox et al., 2001*). Altogether, these data indicate that Rho activity, in mesenchymal cells with extensive FA networks such as fibroblasts or VSMCs, can slow down cell migration by immobilising cells (*Ridley, 2001*). However, we found that sEVs were not influencing mature FA stability. On the contrary, sEV-induced FAs were spatially restricted to the cell periphery near cellular protrusions and were characterised by an enhanced pulling force activity. Interestingly, fibroblast polarisation is driven by Smurf1-dependent RhoA ubiquitinylation and localised degradation in cellular protrusions (*Wang et al., 2003*). A recent study indicated that Rho-dependent contractility is essential for cell invasion in a 3D model and it is tempting to speculate that local Rho activity can enable VSMC invasion by stabilising FAs at the leading edge and stimulating actomyosin-mediated contractility and cell body movement (*Matera et al., 2021*). On the other hand, Rho–ROCK activity is critical for the protease-independent rounded motility of tumour cells and cell types with few adhesion contacts (ameboid) when Rho-dependent contractile forces generate hydrostatic pressure forming multiple membrane blebs to invade the ECM (*Sahai and Marshall, 2003*; *Sanz-Moreno et al., 2008*). In fact, we observed that an extensive secretion of sEVs effectively ceased protrusion activity; also, VSMCs acquired a rounded morphology when 'hovering' over the FN matrix decorated with sEVs (data not shown). Hence, it will be interesting in future studies to investigate whether sEVs can stimulate Rho activity by presenting adhesion modulators—particularly collagen VI—on their surface, thereby guiding cell directionality during invasion.

Collagen VI is a nonfibrillar collagen that assembles into beaded microfilaments upon secretion and it plays both structural and signalling roles (*Okada et al., 2007*). Interestingly, type VI collagen deposition by interstitial fibroblasts is increased in the infarcted myocardium and collagen VI knockout in mice improves cardiac function, structure and remodelling after myocardial infarction via unknown pathways (*Bryant et al., 2009*; *Luther et al., 2012*). Immunohistochemical analysis showed that in the healthy vasculature, collagen VI is detected in the endothelial basement membrane in the intima and it also forms fibrillar structures between smooth muscle cells in the media. In the fibrous plaque, collagen VI is diffusely distributed throughout both the fibrous cap and atheroma (*Katsuda et al., 1992*). Curiously, treatment of ApoE-/- mice with antibodies to collagen VI reduced atherosclerosis, but its exact role has remained unknown (*Liu et al., 2022*). We and others detected Collagen VI in sEVs, but its functional significance remained unknown (*Kapustin et al., 2015*; *He et al., 2015*). Here, we showed that collagen VI is essential for early FA formation at the VSMC periphery as well as

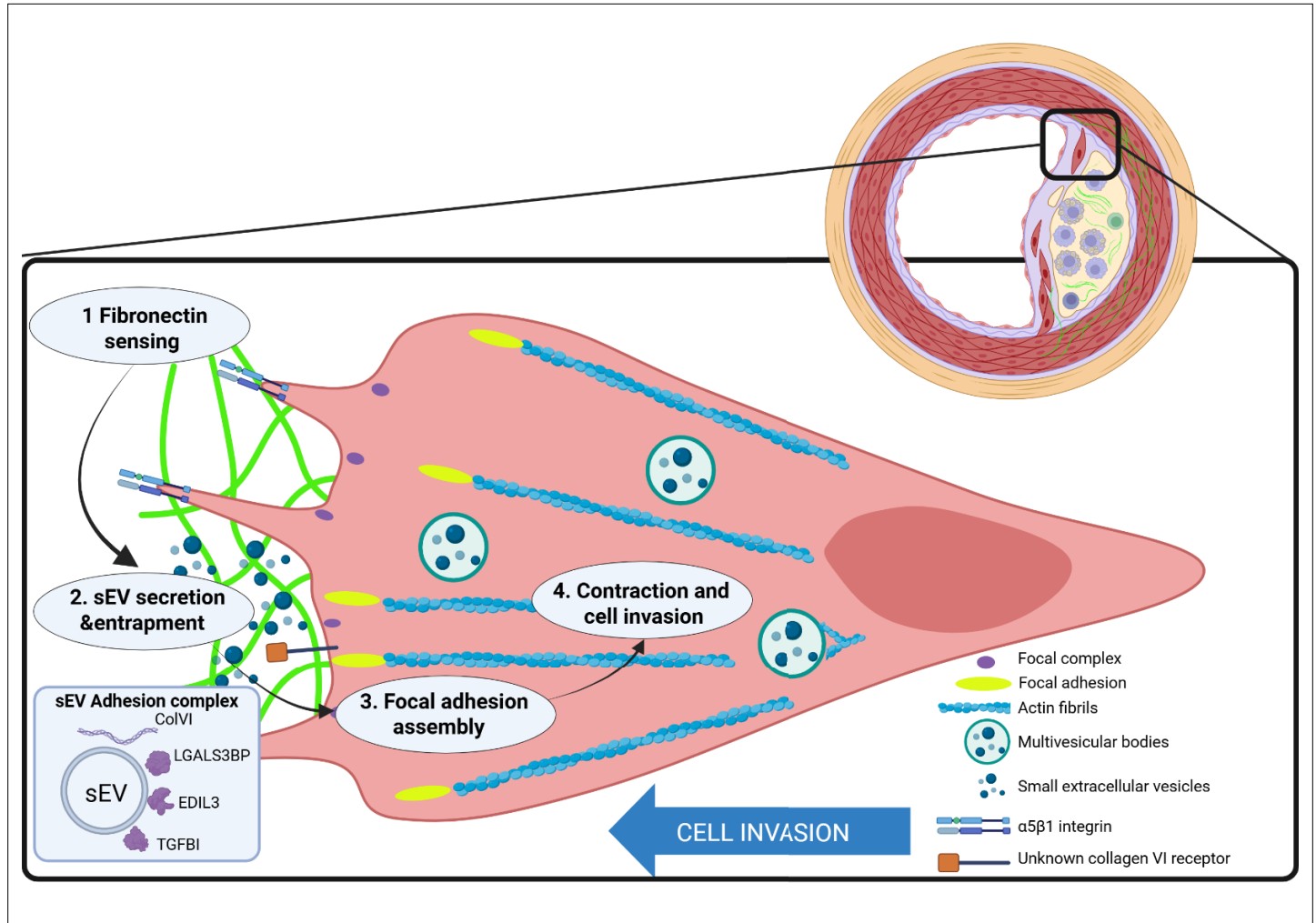

**Figure 8.** Vascular smooth muscle cells sense fibronectin via β1 integrin and secrete small extracellular vesicles loaded with collagen VI. These extracellular vesicles are entrapped in the extracellular matrix and induce formation of peripheral focal adhesions presenting adhesion complex ECM proteins including collagen VI, LGALS3BP, EDIL3, and TGFBI. Focal adhesions anchor the extracellular matrix to actin fibrils in the cell. Contraction of the actin fibrils generates the mechanical force for directional cell invasion through the matrix. This figure was created using BioRender.com.

VSMC invasion in a 3D model. We propose that collagen VI modulates FA formation either by activating cellular signalling or acting as a structural component changing local ECM stiffness (*Sheetz et al., 1998*). Identified collagen VI receptors include α3β1 integrin, the cell surface proteoglycan chondroitin sulfate proteoglycan-4 (CSPG4; also known as NG2), and the anthrax toxin receptors 1 and 2 (*Doane et al., 1998*; *Nishiyama and Stallcup, 1993*; *Nanda et al., 2004*; *Burgi et al., 2017*). Interestingly, a novel, recently identified Collagen VI signalling receptor (CMG2/ANTXR2) mediates localised Rho activation upon collagen VI binding (*Burgi et al., 2017*; *Bürgi et al., 2020*). NG2 also activates the Rho/ROCK pathway leading to effective amoeboid invasiveness of tumour cells which is characterised by excessive blebbing and enhanced actomyosin contractility (*Sahai and Marshall, 2003*; *Paňková et al., 2012*). Hence, binding of collagen VI to these receptors can potentially locally activate Rho and trigger FA formation and actomyosin contractility, thus increasing VSMC motility and directional invasiveness. Of note, it was also reported that NG2 expression is elevated in VSMCs in the atherosclerotic plaque and NG2 knockout prevents plaque formation (*Tillet et al., 2002*; *She et al., 2016*). Curiously, endotrophin, a collagen VI α3 chain-derived profibrotic fragment has recently been associated with a higher risk of arterial stiffness and cardiovascular and all-cause death in patients with type 1 diabetes and atherosclerosis, respectively (*Holm Nielsen et al., 2021*; *Frimodt-Møller et al., 2019*).

In addition to collagen VI, the unique adhesion cluster in VSMC-derived sEVS also includes EGF-like repeat and discoidin I-like domain-containing protein (EDIL3), transforming growth factor-beta-induced protein ig-h3 (TGFBI) and the lectin galactoside-binding soluble 3 binding protein (LGALS3BP), and these proteins are also directly implicated in activation of integrin signalling and cellular invasiveness (*Lee et al., 2016*; *Costanza et al., 2019*; *Stampolidis et al., 2015*). Although we found that collagen VI plays the key role in sEV-induced early formation of FAs in VSMCs, it is tempting to speculate that the high sEV efficacy in stimulating FA formation is driven by cooperative action of this unique adhesion complex on the sEVs surface and targeting this novel sEV-dependent mechanism of VSMC invasion may open up new therapeutic opportunities to modulate atherosclerotic plaque development or even to prevent undesired VSMC motility in restenosis.

We also identified a positive feedback loop and showed that FN stimulates sEV secretion by VSMCs. sEV secretion is a tightly regulated process which is governed by activation of signalling pathways including activation of G-protein-coupled receptors (*Verweij et al., 2018*) and alterations in cytosolic calcium (*Savina et al., 2003*), small GTPases Rab7, Rab27a, Rab27b, and Rab35 (*Jaé et al., 2015*; *Savina et al., 2002*), vesicular trafficking scaffold proteins including syntenin (*Baietti et al., 2012*), sortillin (*Goettsch et al., 2016*), and CD63 (*van Niel et al., 2011*) or accumulation of ceramide (*Trajkovic et al., 2008*). We found that the fibrillar ECM proteins, FN and collagen I, induce sEV secretion by activating the β1 integrin/FAK/Src and Arp2/3-dependent pathways. Our in vivo FN staining data are in good agreement with previous studies showing accumulation of FN in late-stage plaques (*Langley et al., 2017*; *Glukhova et al., 1989*). Moreover, we observed close co-distribution of FN and CD81 in the plaque, suggesting that accumulating FN matrices can stimulate sEV secretion in vivo. Notably, expression of the major FN receptor, α5β1 integrin is re-activated upon VSMC de-differentiation following vascular injury (*Pickering et al., 2000*). It is thought the α5β1 integrin mediates FN matrix assembly, whilst β3 integrins are important for cell-matrix interactions (*Pickering et al., 2000*; *Slepian et al., 1998*). Our data sheds new light on the functional role of α5β1 as an ECM sensor inducing production of collagen VI-loaded sEVs which in turn enhance cell invasion in the complex ECM meshwork. Although our small inhibitors and integrin modulating antibody data clearly indicate that β1 activation triggers sEV secretion via activation of actin assembly, we cannot fully rule out that FN may also be modulating growth factor activity which in turn contributes to sEV secretion by VSMCs (*Kapustin et al., 2015*). Excessive collagen and elastin matrix breakdown in atheroma has been tightly linked to acute coronary events; hence, it will be interesting to study the possible link between sEV secretion and plaque stability, as sEV-dependent invasion is also likely to influence the necessary ECM degradation induced by invading cells (*Libby and Aikawa, 2002*).

In summary, cooperative activation of integrin signalling and F-actin cytoskeleton pathways results in the secretion of sEVs which associate with the ECM and play a signalling role by controlling FA formation and cell-ECM crosstalk (*Figure 8*). Further studies are needed to test these mechanisms across various cell types and ECM matrices.

# Materials and methods

## Key resources table

| Reagent type (species) or resource | Designation | Source or reference | Identifiers | Additional information |
|---|---|---|---|---|
| Antibody | CD9 (rabbit monoclonal, SA35-08 clone) | Novus Biologicals | NBP2-67310 RRID:AB_3353585 | Tissue staining: 1:100 IF: 1:100 |
| Antibody | CD63 (mouse monoclonal, H5C6 clone) | BD Pharmingen | 556019 RRID:AB_396297 | IF: 1:500 WB: 1:1000 Dot-blot: 1:500 |
| Antibody | CD81 (mouse monoclonal, M38 clone) | Novus Biologicals | NBP1-44861 RRID:AB_10008097 | Tissue staining: 1:100 |
| Antibody | CD81 (mouse monoclonal, clone JS-81, PE-conjugated) | BD Pharmingen | 555676 RRID:AB_396029 | FlowCyt: 1:50 |
| Antibody | Syntenin-1 (rabbit monoclonal, clone EPR8102) | Abcam | ab133267 RRID:AB_11160262 | WB: 1:500 |
| Antibody | Syndecan-4 (rabbit polyclonal) | Abcam | ab24511 RRID:AB_448112 | IF: ≈1:500 WB: 1:1000 |

*Continued on next page*

*Continued*

| Reagent type (species) or resource | Designation | Source or reference | Identifiers | Additional information |
|---|---|---|---|---|
| Antibody | EEA1 (mouse monoclonal, clone 14/EEA1) | BD Pharmingen | 610456 RRID:AB_397829 | IF: 1:500 |
| Antibody | α-Actinin-4 (rabbit monoclonal, clone EPR2533(2)) | Abcam | ab108198 RRID:AB_10858236 | WB: 1:500 |
| Antibody | Elastin (rabbit monoclonal, clone EPR20603) | Abcam | ab213720 | Tissue staining 1:100 |
| Antibody | Fibronectin (rabbit polyclonal) | Abcam | ab2413 RRID:AB_2262874 | WB: 1:1000 IF: 1:500 |
| Antibody | Fibronectin (mouse monoclonal, clone IST-9) | Abcam | ab6328 RRID:AB_305428 | IF: 1:480 |
| Antibody | Fibronectin (rabbit monoclonal, clone F14) | Abcam | ab45688 RRID:AB_732380 | WB: 1:500 Tissue staining: 1:250 |
| Antibody | β1 integrin activating (12 G10) | Published; *Parsons et al., 2008* | | Functional assay: 5 µg/ml |
| Antibody | β1 integrin inhibiting (mouse monoclonal, clone 4B4LDC9LDH8 (4B4)) | Beckman Coulter | 41116015 | Functional assay: 5 µg/ml |
| Antibody | β1 integrin (mouse monoclonal, clone MEM-101A) | Santa Cruz Biotechnology | sc-51649 RRID:AB 629022 | Flow Cyt: 1:250 WB: ≈1:1000 |
| Antibody | Vinculin (mouse monoclonal, clone hVIN-1) | Sigma | V9264 RRID:AB_10603627 | IF: 1:400-1:500 WB: 1:1000 |
| Antibody | GAPDH (mouse) | Abcam | ab139416 | WB: 1:250 |
| Antibody | β-actin (mouse monoclonal, clone AC-74) | Sigma | A2228 RRID:AB_476697 | WB: 1:10000 |
| Antibody | α5 integrin (mouse monoclonal, clone P1D6) | Abcam | ab78614 RRID:AB_1603217 | Flow Cyt: 1:250 |
| Antibody | Myo10 (rabbit polyclonal) | Sigma | HPA024223 RRID:AB_1854248 | IF: 1:250 |
| Antibody | Galectin-3BP/MAC-2BP (goat polyclonal) | R&D Systems | AF2226 RRID:AB_2137065 | WB: 1:1000 |
| Antibody | EDIL3 (mouse monoclonal, clone #670421) | R&D Systems | MAB6046 RRID:AB_10993573 | WB: 1:250 |
| Antibody | TGFBI (goat polyclonal) | Sigma | SAB2501486 RRID:AB_10964448 | WB: 1:170 |
| Antibody | Mouse IgG (mouse polyclona) | Sigma | PP54 RRID:AB_97851 | Functional assay: 5 µg/ml or 50 µg/ml |
| Antibody | Anti-collagen Type VI (mouse monoclonal, clone 3 C4) | Sigma | MAB1944 RRID:AB_2083113 | Functional assay: 50 µg/ml |
| Antibody | Anti-collagen Type VI, rabbit polyclonal | Abcam | ab6588 RRID:AB_305585 | WB: 1:1000 |
| Antibody | Anti-collagen Type VI, rabbit monoclonal | Abcam | ab182744 RRID:AB_2847919 | Dot blot: 1:200 |
| Antibody | ARPC2/p34-Arc, rabbit polyclonal | Millipore | #07–227 RRID:AB_11212539 | IF: 1:500 WB: 1:2000 |
| Antibody | Cortactin, rabbit monoclonal, clone EP1922Y | Abcam | ab81208 RRID:AB_1640383 | IF: 1:500 |
| Antibody | GAPDH, mouse monoclonal | Abcam | ab139416 | WB: 1:250 |
| Antibody | Alix, mouse monoclonal, clone 3 A9 | ThermoFisher Scientific | MA1-83977 RRID:AB_2162469 | Dot blot: 1:500 |
| Antibody | AlexaFluor 488 anti-rabbit IgG, donkey polyclonal | Invitrogen | A21206 RRID:AB_2535792 | IF: 1:200 |

*Continued on next page*

*Continued*

| Reagent type (species) or resource | Designation | Source or reference | Identifiers | Additional information |
|---|---|---|---|---|
| Antibody | AlexaFluor 488 anti-rabbit IgG, goat polyclonal | Invitrogen | A11008 RRID:AB_143165 | IF: 1:200 |
| Antibody | AlexaFluor 488 anti-mouse IgG, donkey polyclonal | Invitrogen | A21202 RRID:AB_141607 | IF: 1:200 |
| Antibody | AlexaFluor 546 anti-rabbit IgG, donkey polyclonal | Invitrogen | A10040 RRID:AB_2534016 | IF: 1:200 |
| Antibody | AlexaFluor 546 anti-mouse IgG donkey polyclonal | Invitrogen | A10036 RRID:AB_11180613 | IF: 1:200 |
| Antibody | AlexaFluor 647 anti-mouse IgG, donkey polyclonal | Invitrogen | A31571 RRID:AB_162542 | IF: 1:200 |
| Antibody | AlexaFluor 647 anti-rabbit IgG, donkey polyclonal | Invitrogen | A31573 RRID:AB_2536183 | IF: 1:200 |
| Antibody | Anti-mouse Alexa Fluor 488- conjugated IgG, donkey polyclonal | Abcam | ab150109 RRID:AB_2571721 | Tissue staining 1:500 |
| Antibody | Anti-rabbit Alexa Fluor 555-conjugated IgG, donkey polyclonal | Abcam | ab150062 RRID:AB_2801638 | Tissue staining 1:500 |
| Antibody | Anti-mouse IgG IRDye@680 LT, donkey polyclonal | Li-COR | 926–68022 RRID:AB_10715072 | Dot-blot: 1:10000 |
| Antibody | Anti-rabbit IgG IRDye@800 CW, donkey polyclonal | Li-COR | 926–32213 RRID:AB_621848 | Dot-blot: 1:10000 |
| Antibody | Anti-mouse IgG IRDye 680RD, goat polyclonal | Li-COR | 925–68070 RRID:AB_2651128 | WB: 1:10000 |
| Antibody | Anti-rabbit IgG IRDye@800 CW, goat polyclonal | Li-COR | 926–32211 RRID:AB_621843 | WB: 1:10000 |
| Antibody | Horseradish peroxidase-conjugated anti-rabbit IgG, goat polyclonal | Cell Signaling Technology | 7074 RRID:AB_2099233 | WB: 1:200 |
| Antibody | Horseradish peroxidase-conjugated anti-mouse IgG&IgM, goat polyclonal | Sigma-Aldrich | AP130P RRID:AB_91266 | WB: 1:500 |
| Antibody | Horseradish peroxidase-conjugated anti-mouse IgG (sheep) | GE Healthcare | NA931V RRID:AB_772210 | WB: 1:10000 |
| Antibody | Horseradish peroxidase-conjugated anti-rabbit IgG (donkey) | GE Healthcare | NA934V | WB: 1:10000 |
| Peptide, recombinant protein | Fibronectin | Cell Guidance Systems | AP-23 | Matrix coating |
| Peptide, recombinant protein | Collagen I | Gibco | #A1048301 | Matrix coating |
| Peptide, recombinant protein | Laminin | Roche | 11243217001 | Matrix coating |
| Peptide, recombinant protein | Matrigel | Corning | #356237 | Matrix for 3D assay |
| Peptide, recombinant protein | Phalloidin-rhodamin | ThermoFisher Scientific | R415 | Actin staining IF: 1:200 |
| Chemical compound, drug | 3-O-Methyl-Sphingomyelin (SMPD3 inhibitor) | Enzo Life Technologies | BML-SL225-0005 | Blocks sEV secretion, 10 µM |
| Chemical compound, drug | FAM14 (FAK inhibitor) | Abcam | ab146739 | Blocks FAK pathway, 0.9 µM |
| Chemical compound, drug | PP2 (Src inhibitor) | Life Technologies | PHZ1223 | Blocks Src pathway, 4 µM |

*Continued on next page*

*Continued*

| Reagent type (species) or resource | Designation | Source or reference | Identifiers | Additional information |
|---|---|---|---|---|
| Chemical compound, drug | CK666 (Arp2/3 inhibitor) | Abcam | ab141231 | Blocks Arp2/3 complex, 100 μM |
| Chemical compound, drug | SMIFH2 (Formin inhibitor) | Sigma | S4826 | Blocks formin pathway, 5 μM |
| Cell line (Homo- sapiens) | Human aortic vascular smooth muscle cells (VSMCs) | In-house (King's College London) | N/A | Isolated and characterized |
| Sequence-based reagent | Control siRNA pool | Dharmacon | D-001810-10-05 | Negative control |
| Sequence-based reagent | Collagen VI siRNA (COL6A3, Human), SMARTPool | Horizon | L-003646-00-0005 | Knockdown experiments |
| Recombinant DNA reagent | CD63-GFP | Dr. Aviva Tolkovsky; *Bampton et al., 2005* | | sEV marker (fluorescent) |
| Recombinant DNA reagent | CD63-pHluorin | Published; *Verweij et al., 2018* | | sEV marker (pH sensitive) |
| Recombinant DNA reagent | CD63-RFP | Published; *Leifer et al., 2004* | | sEV marker (fluorescent) |
| Recombinant DNA reagent | Paxillin-RFP | Published; *Parsons et al., 2008* | | FA live imaging |
| Recombinant DNA reagent | ARPC2–GFP DNA vector | Published; *Abella et al., 2016* | | Actin nucleator |
| Recombinant DNA reagent | F-tractin-RFP | Dr. Thomas S. Randall | N/A | F-actin marker |
| Recombinant DNA reagent | Myo10-GFP | Addgene | RRID:Addgene_135403 | Filopodia marker |
| Software, algorithm | ACEA xCELLigence | ACEA | N/A | Cell adhesion assay |
| Software, algorithm | Fiji (ImageJ) | NIH | https://imagej.net/ RRID:SCR_002285 | Image analysis |
| Software, algorithm | ExoView | NanoView Biosciences | N/A | EV analysis |

## Materials

Proteins were fibronectin (Cell Guidance Systems, AP-23), collagen I (Gibco, #A1048301), laminin (Roche, 11243217001), Matrigel (Corning, #356237), Phalloidin-rhodamin (ThermoFisherScientific, R415). Peptides were Gly-Arg-Gly-Asp-Ser-Pro (GRGDSP, Merck, SCP0157) and scramble control Gly-Arg-Ala-Asp-Ser-Pro (GRADSP, Merck, SCP0156). All chemical inhibitors were diluted in DMSO and were: 3-O-Methyl-Sphingomyelin (SMPD3 inhibitor, Enzo Life technologies, BML-SL225-0005), FAM14 inhibitor (FAK inhibitor, Abcam, ab146739), PP2 (Src inhibitor, Life technologies, PHZ1223), CK666 (Arp2/3 inhibitor, Abcam ab141231), SMIFH2 (Formin inhibitor, Sigma, S4826). Control siRNA pool (ON-TARGETplus siRNA, Dharmacon, D-001810-10-05), collagen VI siRNA ON-TARGETplus siRNA (*COL6A3*, Human), SMARTPool, Horizon, L-003646-00-0005. Antibodies were CD9 (SA35-08 clone, NBP2-67310, Novus Biologicals), CD63 (BD Pharmingen, 556019), CD81 (BD Pharmingen, 555676, B-11, SantaCruz, sc-166029 and M38 clone, NBP1-44861, Novus Biologicals), Syntenin-1 (Abcam, ab133267), Syndecan-4 (Abcam, ab24511), α-Actinin-4 (Abcam, ab108198), fibronectin (Abcam, ab2413, ab6328 [IST-9] (3D matrix staining) and F14 clone, ab45688 (clinical samples analysis)), β1 activating (12G10) antibody was previously described (*Parsons et al., 2008*), 4B4 integrin inhibiting antibody (Beckman Coulter, 41116015), vinculin (Sigma, V9264), α5 integrin (P1D6, Abcam, ab78614), Myo10 (Sigma, HPA024223), galectin-3BP/MAC-2BP (R&D systems, AF2226), EDIL3 antibody (R&D systems, MAB6046), TGFBI (Sigma, SAB2501486), IgG mouse (Sigma PP54), Anti-collagen Type VI antibody, clone 3C4 (Sigma, MAB1944) (functional blocking), Anti-collagen Type VI antibody (Abcam, AB6588) (western blot), Anti-collagen Type VI antibody (Abcam, AB182744) (dot blot), p34-Arc/ARPC2 antibody (Millipore, #07–227), Cortactin (Abcam, ab81208 [EP1922Y]), GAPDH (ab139416, Abcam), Alix (Clone 3A9, ThermoFisher Scientific, MA1-83977), EEA1 (BD Pharmingen, 610456), Elastin (Abcam, ab213720), β1 integrin (Santa Cruz Biotechnology, sc-51649), β-actin (Sigma, A2228), Donkey Anti-mouse Secondary IRDye@680 LT (Li-COR, 926–68022) and Donkey

Anti-Rabbit Secondary IRDye@800 CW (Li-COR, 926–32213), AlexaFluor 488 anti-rabbit IgG (Invitrogen, A21206), AlexaFluor 488 anti-rabbit IgG (Invitrogen, A11008), AlexaFluor 488 anti-mouse IgG (Invitrogen, A21202), AlexaFluor 546 anti-rabbit IgG (Invitrogen, A10040), AlexaFluor 546 anti-mouse IgG (Invitrogen, A10036), AlexaFluor 647 anti-mouse IgG (Invitrogen, A31571), AlexaFluor 647 anti-rabbit IgG (Invitrogen, A31573), anti-mouse Alexa Fluor 488-conjugated IgG (Abcam, ab150109), anti-rabbit Alexa Fluor 555-conjugated IgG (Abcam, ab150062), anti-mouse IgG IRDye 680RD (Li-COR, 925–68070), anti-rabbit IgG IRDye@800 CW, goat polyclonal (Li-COR, 926–32211), horseradish peroxidase-conjugated anti-rabbit IgG (Cell Signaling Technology, 7074), horseradish peroxidase-conjugated anti-mouse IgG&IgM (Sigma-Aldrich, AP130P), horseradish peroxidase-conjugated anti-mouse IgG (GE Healthcare, NA931V), horseradish peroxidase-conjugated anti-rabbit IgG (GE Healthcare, NA934V), DNA plasmids were: CD63-GFP was kindly provided by Dr Aviva Tolkovsky (*Bampton et al., 2005*), CD63-pHluorin was previously described (*Verweij et al., 2018*), CD63-RFP was previously published (*Leifer et al., 2004*), Paxillin-RFP was previously described (*Parsons et al., 2008*), ARPC2–GFP DNA vector was a kind gift from Professor Michael Way laboratory (*Abella et al., 2016*), F-tractin-RFP was kindly provided by Dr Thomas S. Randall (King's College London, UK), Myo10-GFP (Addgene Plasmid#135403) was previously described (*Berg and Cheney, 2002*).

## Cell culture

Cell isolates used in this study were primary human aortic vascular smooth muscle cells (VSMCs) isolated in house from explants of human aortic tissues from healthy transplant donors (35-year-old female (04–35F-11A), 20-year-old male (05–20M-8A), 22-year-old male (05–22M-18A), 38-year-old female (03–38F-9A) and 33-year-old female (05–33F-5A)) as previously described (*Reynolds et al., 2004*). All experiments were performed using at least 3 biological replicates unless stated. Human materials were handled in compliance with the Human Tissue Act (2004, UK) and with ethical approval from the local research ethics committee (LREC 97/084). VSMCs were verified using morphology, immunofluorescence for smooth muscle contractile markers (α-smooth muscle actin, transgelin (SM22α), calponin and smooth muscle myosin heavy chain) and proteomics analysis as described (*Kapustin et al., 2015*; *Reynolds et al., 2004*). Cells were mycoplasma-free on monthly PCR-based testing. VSMCs were cultured in M199 medium (Sigma) supplemented with 20% fetal calf serum, 100 U/ml penicillin, 100 µg/ml streptomycin and 2 mmol/L L-glutamine (Gibco) and used between passages 4 and 12.

## VSMC adhesion

24-well plate (Corning Costar) was incubated with 5 µg/ml fibronectin in PBS overnight at +4°C. Next, sEVs (10 µg/ml) diluted in PBS were added to the wells and incubated overnight at +4°C. Wells were washed with PBS, blocked with PBS-1% BSA for 30 min at 37°C and washed three times with PBS again. VSMCs were incubated in serum-free media, M199 supplemented with 0.5% BSA, 100 U/ml penicillin and 100 µg/ml streptomycin overnight and removed by brief trypsin treatment. Next, 20,000 cells resuspended in M199 media supplemented with 20% sEV-free FBS were added to each well and incubated for 30 min at +37°C. Unattached cells were washed away with PBS and attached cells were fixed with 3.7% PFA for 15 min at 37°C. Cells were washed with $H_2O$ three times and stained with 0.1% crystal violet for 30 min at 10% $CH_3COOH$ for 5 min at room temperature. Samples were transferred to a 96-well plate and absorbance was measured at 570 nm using the spectrophotometer (Tecan GENios Pro).

## iCELLigence adhesion protocol

E-Plate L8 (Acea) was coated with FN (5 µg/ml) in PBS at 4°C overnight. Next, E-Plate L8 was gently washed once with cold PBS followed by the incubation with EV or sEV (10 µg/ml) at 4°C overnight. The following day, E-Plate L8 was incubated with 0.1% BSA in PBS for 30 min at 37°C to block non-specific binding sites and washed with PBS before adding 250 µl of M199 media supplemented with 20% sEV-free FBS. VSMCs were serum-deprived by incubation in M199 media supplemented with 0.5% BSA overnight, were passaged with trypsin and resuspended in media M199 supplemented with 20% sEV-free FBS to a final concentration 80,000 cells/ml. Cell aliquot (250 µl) was added to E8-plate L8 which was then transferred to iCELLigence device for the adhesion assay. Cell adhesion was measured at 37°C with time intervals of 20 s for 1 hr.

## Focal adhesion turnover assay

iBIDI 35 mm dishes were incubated with FN (5 µg/ml) in PBS overnight at +4°C and treated with sEVs (10 µg/ml) diluted in PBS overnight at +4°C. 500,000 cells were transfected with Paxillin-RFP/Myo10-GFP by using electroporation (see below) and plated on iBIDI 35 mm dishes coated with FN and sEVs in Dulbecco's Modified Eagle Medium (DMEM) supplemented with 20% sEV-free FBS, 100 U/ml penicillin, and 100 µg/ml streptomycin. Next, cells were transferred to DMEM supplemented with 20% sEV-free FBS, 10 mM HEPES, 100 U/ml penicillin, and 100 µg/ml streptomycin and images were acquired (Nikon Spinning Disk) every 1 min for 30 min at 37°C. FA images were extracted from timelapses and FA turnover was quantified using Mathematica by producing the adhesion map as previously described (*Holt et al., 2008*).

## Confocal spinning disk microscopy and live cell imaging

VSMC were transfected using electroporation (see below) and plated onto FN-coated 35 mm iBIDI glass bottom dish and incubated for 48 hr. Then cells were transferred to DMEM supplemented with 20% sEV-free FBS, 10 mM HEPES, 100 U/ml penicillin, and 100 µg/ml streptomycin and images were acquired every 1 min for 20 min at 37°C using a Nikon Ti-E (inverted) microscope equipped with a Yokogawa spinning disk and a Neo 5.5 sCMOS camera (Andor) and 60 x or 100 x/1.40 NA Plan Apo $\lambda$ oil objectives (Nikon) were used. Images were acquired using NIS Elements AR 4.2 software. Cells were maintained at 37°C, 5% $CO_2$ throughout the experiment via a $CO_2$ chamber and a temperature-regulated Perspex box which housed the microscope stage and turret.

## iSIM

Slides were imaged and super-resolved images collected using a Visitech-iSIM module coupled to a Nikon Ti-E microscope using a Nikon 100x1.49 NA TIRF oil immersion lens. Blue fluorescence was excited with a 405 nm laser and emission filtered through a 460/50 filter. Green fluorescence was excited with a 488 nm laser and emission filtered through a 525/50 filter. Red fluorescence was excited with a 561 nm laser and emission filtered through a 630/30 filter. Far Red fluorescence was excited with a 640 nm laser and emission filtered through a 710/60 filter. Images were collected at focal planes spaced apart by 0.08 µm. Data was deconvolved using a Richardson-Lucy algorithm utilising 20 iterations using the Nikon deconvolution software module (Nikon Elements). 65 stack images were taken with Z step 0.08 µm. ImageJ (153t) or NIS Elements (5.21.00, built1483, 64bit) software was used for image analysis and assembly.

## Cell electroporation

iBidi dishes were coated with FN (5 µg/ml) in PBS for 1 hour at 37°C or overnight at 4°C. VSMCs were grown to 70% confluence, washed twice with EBSS and detached by trypsin treatment for 5 min at +37°C. Cells (500,000) were mixed with plasmids (2.5 µg each) in electroporation buffer (100 µl, Lonza) and transfected using either Nucleofector II (program U25) or Nuclefector III (program CM-137).

## Isolation of apoptotic bodies, extracellular vesicles and small extracellular vesicles

Flasks (T150) were incubated with PBS or FN (5 µg/ml) in PBS overnight at +4°C and VSMCs ($\approx 10^6$ cells) were plated and incubated for 16 hr at +37°C. Next, cells were washed with EBSS three times and incubated with DMEM supplemented with 0.1% BSA, 100 U/ml penicillin, and 100 µg/ml streptomycin. Conditioned media was collected and centrifuged at 2500 rpm (Thermo Fisher Scientific Heraeus Multifuge 3SR +centrifuge, rotor Sorvall 75006445) for 5 min to remove apoptotic bodies (AB, 1.2 K pellet). Supernatant was transferred to centrifugation tube (Nalgene Oak Ridge High-Speed Polycarbonate Centrifuge Tubes, Thermo Fisher Scientific, 3138–0050) and centrifuged 10,000 x$g$ for 30 min. 10 K pellet (EVs) was collected, washed in PBS once and kept at –80°C until further analysis. 10 K supernatant was transferred to ultracentrifugation tubes (Polycarbonate tubes for ultracentrifugation, BeckmanCoulter, 355647) and centrifuged at 35,000 rpm (100,000x$g$) for 40 min at 4°C (Fixed-angle rotor, Beckman Coulter Optima Max Unltracentrifuge). 100 K pellet (sEVs) was washed with PBS twice and re-suspended in PBS. EV and sEV pellets were kept at –80°C for the proteomic analysis and freshly isolated sEVs were used for all other assays.

## EV and sEV proteomic analysis

EV and sEV samples were submitted to the CEMS Proteomics Facility in the James Black Centre, King's College London for mass spectrometric analysis. The data were processed by Proteome Discover software for protein identification and quantification. Scaffold 5 proteome software was utilised to visualise differential protein expression. Individual protein intensities were acquired and transformed to log2 scale before Hierarchical clustering analysis based on Euclidean distance and k-means processing. Differential genes defined by multiple groups comparison were applied to perform the gene ontology functional assay through using the DAVID Gene Ontology website (https://david.ncifcrf.gov/). Selected genes which showed abundance in each condition were applied to generate a Venn diagram through the Interacti website (http://www.interactivenn.net/).

## Cell lysis

Cells were washed with PBS and removed by cell-scraper in 1 ml of PBS. The cells were pelleted by centrifugation at 1000x*g* for 5 min and cell pellets were kept at –80°C until further analysis. To prepare cell lysates, pellets were incubated with lysis buffer (0.1 M TrisHCl (pH 8.1), 150 mM NaCl, 1% Triton X-100 and protease inhibitors cocktail (Sigma, 1:100)) for 15 min on ice. Cell lysates were subjected to ultrasound (Branson Sonifier 150), centrifuged at 16,363x*g* for 15 min (Eppendorf) at 4°C and supernatants were collected and analysed by western blotting.

## CD63-bead capturing assay

CD63-bead capturing assay was conducted as previously described (*Kapustin et al., 2015*) with some modifications. In brief, CD63 antibody (35 µg) was immobilised on $1 \times 10^8$ 4 µm aldehyde-sulfate beads (Invitrogen) in 30 mM MES buffer (pH 6.0) overnight at room temperature and spun down by centrifugation (3000x*g*, 5 min). Next, beads were washed three times with PBS containing 4% BSA and kept in PBS containing 0.1% Glycine and 0.1% $NaN_3$ at +4°C. VSMCs were plated onto 24-well plates (10,000 cells per well) and incubated in the complete media overnight. Next, cells were washed three times with EBSS and incubated in M199 supplemented with 2.5% sEV-free FBS in the presence or absence of inhibitors for 2 hr. Then, conditioned media was replaced once to the fresh aliquot of 2.5% sEV-free FBS in the presence or absence of inhibitors and cells were incubated for 18–24 hr. VSMC conditioned media was collected and centrifuged and centrifuged at 2500x*g* for 5 min. VSMCs were detached from the plate by trypsin treatment and viable cells were quantified using NC3000 (Chemo-Metec A/S). The conditioned media supernatants were mixed with 1 µl of anti-CD63-coated beads and incubated on a shaker overnight at +4°C. Beads were washed with PBS-2%BSA and incubated with anti-CD81-PE antibody (1:50 in PBS containing 2% BSA) for 1 hr at room temperature. Next, beads were washed with PBS-2%BSA and PBS and analysed by flow cytometry (BD Accuri C6). sEV secretion (fold change) was calculated as ratio of Arbitrary Units (fluorescence units x percentage of positive beads and normalised to the number of viable VSMCs) in the treatment and control conditions.

## sEV labelling with Alexa Fluor 568 C5 Maleimide

sEVs (10 µg/ml) were incubated with 200 µg/ml Alexa Fluor 568 C5 Maleimide (Thermo Fisher Scientific, #A20341) in PBS for 1 hr at room temperature. An excessive dye was removed by using Exosome Spin Columns (MW 3000, Thermo Fisher Scientific, #4484449) according to the manufacturer protocol.

## Fetuin-A and Fibronectin labelling and uptake studies

Bovine fetuin-A (Sigma) and Fibronectin were labelled using an Alexa488 (A10235) and Alex568 (A10238) labelling kits in accordance with the manufacturer's protocol (Invitrogen). VSMCs were serum-starved for 16 hr and then incubated with Alexa488-labelled fetuin-A (10 µg/ml) and Alexa555-labelled fibronectin (10 µg/ml) from 30 to 180 min at 37°C.

## VSMC and beads-coupled sEVs flow cytometry

VSMC were removed from the plate by brief trypsin treatment, washed with M199 media supplemented with 20% FBS and resuspended in HBSS supplemented with 5% FBS. Cells were kept on ice throughout the protocol. Next, cells (200,000) were incubated with primary antibody for 30 min on ice, washed with HBSS-5%FBS and incubated with secondary fluorescently-labelled antibody for

30 min on ice. Then cells were washed twice with HBSS-5%FBS and once with PBS and analysed by flow cytometry (BD FACScalibur, BD Bioscience).

Flow cytometry analysis of beads-coupled sEVs was conducted as described before (*Kapustin et al., 2011*). In brief, sEVs (10 µg) isolated by differential centrifugation were coupled to 4 µm surfactant-free aldehyde/sulfate latex beads (Invitrogen) and were incubated with primary antibody and fluorescently labelled secondary antibody and analysed by flow cytometry on BD FACScalibur (BD Bioscience). Data were analysed using the Cell Quest Software (BD). Cells and beads stained with isotype-control antibodies were used as a negative control in all experiments.

## Nanoparticle tracking analysis

sEVs were diluted 1:150 and analysed using LM-10 using the light scattering mode of the NanoSight LM10 with sCMOS camera (NanoSight Ltd, Amesbury, United Kingdom). 5 frames (30 s each) were captured for each sample with camera level 10 and background detection level 11. Captured video was analysed using NTA software (NTA 3.2 Dev Build 3.2.16).

## Western blotting and dot blotting

For western blotting, cell lysates or apoptotic bodies, EVs and sEVs in Laemmli loading buffer were separated by 10% SDS-PAGE. Next, separated proteins were transferred to ECL immobilon-P membranes (Millipore) using semi-dry transfer (Bio-Rad). Membranes were incubated in the blocking buffer (TBST supplemented with 5% milk and 0.05% tween-20) for 1 hr at room temperature, then incubated with primary antibody overnight at +4°C. Next, membranes were washed with blocking buffer and incubated with secondary HRP-conjugated antibody and washed again in PBS supplemented with 0.05% tween-20 and PBS. Protein bands were detected using ECL plus (Pierce ECL Western Blotting Substrate, #32109). Alternately, membranes were incubated with the secondary fluorescent antibody (Alexa fluor antibody) and washed again with TBST or PBST. Blots were detected using Odyssey Licor.

Dot blot was conducted as previously described (*Kapustin et al., 2021*). In brief, an aliquot of isolated sEVs (0.5 µg) was absorbed onto nitrocellulose membrane (Odyssey P/N 926–31090) for 30 min at room temperature and air dried. Next, membranes were blocked with PBS-3% BSA in PBS and incubated with primary antibodies (anti-CD63 1:500, anti-COLVI 1:200, anti-Alix, 1:500) in the absence or presence of 0.2% (v/v) Tween-20 for 16 hr at the cold room (+4C) room temperature. Membranes were washed three times in PBS-3% BSA, and were incubated with Donkey Anti-Rabbit Secondary IRDye@800 CW (Li-COR, 926–32213) and Donkey Anti-Mouse Secondary IRDye@680 LT (Li-COR, 926–68022) (1:10,000), washed with PBS-0.2% tween20 and PBS and visualised and quantified using Odyssey CLx Infrared Imaging System.

## Generation of 3D extracellular matrix

Generation of 3D matrices was conducted as previously described (*Beacham et al., 2007*) with some modifications. In brief, plates or glass coverslides were covered with 0.2% gelatin in PBS for 1 hr at 37°C. Wells were washed with PBS and fixed with 1% glutaraldehyde in PBS for 30 min at room temperature. Plates were washed with PBS and incubated with 1 M ethanolamine for 30 min at room temperature, then washed with PBS again. VSMCs ($5 \times 10^5$ per well) were plated and cultured for 24 hr. Then the medium was replaced with medium containing 50 µg/ml of ascorbic acid and cells were incubated for 9 days. The medium was replaced every 48 hr. To extract the matrix, cells were rinsed with PBS and incubated with pre-warmed extraction buffer (20 mM $NH_4OH$ containing 0.5% Triton X-100) for 3 min until intact cells were not seen. Next, equal volume of PBS was added to extraction buffer and plates were incubated for 24 hr at +4°C. Plates were washed with PBS twice and kept with PBS containing 100 U/ml penicillin and 100 µg/ml streptomycin up to 2 weeks at +4°C.

## sEVs isolation from 3D extracellular matrix

VSMCs were cultured on dishes pre-coated with 0.1% gelatin as described above, with minor modifications. Briefly, cells were maintained in M199 supplemented with sEV-free FBS, 1% penicillin-streptomycin, and 50 µg/ml ascorbic acid. Cultures were maintained for 10 days, with media changes and collections performed every 2–3 days. At the end of the culture period, the medium was gently removed and collected as the 'Media' fraction. Conditioned media was clarified by centrifugation at

2500 × *g* for 15 min to remove cellular debris; the resulting supernatant was retained for ExoView analysis and small EV (sEV) isolation. The cell monolayer was then rinsed with phosphate-buffered saline (PBS) and incubated with extraction buffer (0.5 M NaCl, 20 mM Tris, pH 7.5) for 4 hr at room temperature as previously described (*Didangelos et al., 2010*). This NaCl-extracted ECM fraction, termed the 'loosely-bound sEV fraction', was collected and centrifuged at 2500 × *g* for 15 min. The supernatant was retained for ExoView analysis and sEV isolation. Following NaCl extraction, cells were treated with 0.1% sodium dodecyl sulfate and 25 mM ethylenediaminetetraacetic acid (EDTA), incubating at 37°C for 10 min to lyse any remaining cellular material. To extract the remaining ECM-associated fraction, the matrix was incubated with 4 M guanidine hydrochloride (GuHCl) in 50 mM sodium acetate buffer (pH 5.8) for 48 hr at room temperature. The GuHCl fraction, referred to as 'strongly-bound sEV fraction' was collected and centrifuged at 2500 × *g* for 15 min at 4°C; the supernatant was retained for ExoView analysis. sEVs from all fractions were isolated by ultracentrifugation as described above.

## ExoView analysis of ECM-derived sEV fractions

ECM-derived EV fractions (media, loosely bound and strongly trapped sEVs) were prepared as detailed above. For ExoView analysis, each fraction was diluted 1:2 in the manufacturer-supplied incubation solution. Leprechaun Exosome Human Tetraspanin chips (Unchained Labs, REF 251–1044) were used according to the manufacturer's protocol.

Briefly, 35–50 µl of diluted sample (1:2-1:10) was applied to each chip well. Chips were incubated overnight at room temperature in a sealed humidity chamber to allow EV capture onto surface-immobilised antibodies. The following day, chips were washed per the supplied protocol to remove unbound material.

Secondary immunostaining was performed using fluorescently labelled antibodies provided in the Leprechaun Exosome Human Tetraspanin Kit (CD9, CD63, and CD81, Unchained Labs, LOT UL038460001F). After antibody incubation, chips were washed thoroughly as recommended and air-dried briefly.

Chips were scanned on an ExoView R200 platform (Unchained Labs). Imaging and data analysis were performed using ExoView Analyzer software, with spot intensity and particle counts determined for each tetraspanin (CD9, CD63, CD81) according to the manufacturer's guidelines. Data were exported for downstream quantitative and statistical analysis.

## Immunocytochemistry

VSMCs (10,000 cells per well) were plated onto the coverslips and incubated for 48 hr. Next, cells were treated with inhibitors in M199 supplemented with 2.5% sEV-free FBS and fixed using 3.7% paraformaldehyde for 15 min at +37°C. Cells were then washed with PBS, permeabilised with PBS-0.1% triton X-100 for 5 min at room temperature and washed with PBS again. Cells were blocked with PBS-3% BSA for 1 hr at room temperature and incubated with primary antibodies overnight at +4°C. Following washing three times with PBS-3% BSA, cells were incubated with secondary fluorescently labelled antibodies in the dark for 1 hr at room temperature, stained with DAPI for 5 min and mounted onto 75-Superdex slides using gelvatol mounting media and analysed by spinning disk confocal microscopy as above (Nikon). Primary antibodies were used in the following dilutions: CD63 1:500, Cortactin 1:500, Arp2C 1:500, Myo10 1:250, Vinculin 1:500, fibronectin 1:500, CD81 1:500, Rhodamine phalloidin 1:200 and secondary fluorescently labelled antibodies 1:200.

## Focal adhesion turnover/Individual traction force

PDMS pillar (500 nm diameter, 1.3 µm height, 1 µm centre-to-centre) substrates were prepared as described previously (*Pandey et al., 2018*). Briefly, Cd $Eq3: T_{tilt}(\nu) = a\frac{(1+\nu)}{2\pi}\left\{2\left(1-\nu\right) + \left(1 - \frac{1}{4(1-\nu)}\right)\right\}$ SeS/ZnS alloyed quantum dots (490 nm, Sigma) were spun on the master for 30 s at 10,000 rpm with a 150i spin processor (SPS), before the addition of PDMS. PDMS (Sylgard 184, Dow Corning) was mixed thoroughly with its curing agent (10:1), degassed, poured over the silicon master, placed upside-down on a plasma-treated coverslip-dish (Mattek), or coverslip four-well dishes (Ibidi) and cured at 80°C for 2 hr. The mould was then removed and the pillars were incubated with fibronectin for 1 hr at 37°C, after which pillars were incubated with sEVs (10 µg/ml in PBS) at 4°C overnight.

VSMC previously transfected with Paxillin-RFP by using electroporation as above were plated on the Pillars and imaged on a Nikon Eclipse Ti-E epifluorescent microscope with a 100x1.4NA objective, Nikon DS-Qi2 Camera and a Solent Scientific chamber with temperature and $CO_2$ control. For the calculation of pillar displacements, a perfect grid was assumed and deviations from the grid were calculated using reference pillars outside the cell. The traction stress was calculated for a 10 µm wide region at the cell edge that was enriched in paxillin staining, taking into account all pillar displacements above a ~20 nm noise level, which was calculated from pillars outside the cells.

The stiffness of the pillars was calculated as described by *Ghibaudo et al., 2008* but taking into account substrate warping as described by *Schoen et al., 2010* (*Equations 1–5*):

$$k_{bend} = \frac{3}{64}\pi E \frac{D^4}{H^3}; \tag{1}$$

$$corr = \frac{\frac{16}{3}(\frac{L}{D})^3}{(\frac{16}{3}(\frac{L}{D})^3) + \frac{7+6vL}{3}\frac{}{D} + 8T_{tilt}(v)(\frac{L}{D})^2}; \tag{2}$$

$$T_{tilt}(\nu) = a\frac{(1+\nu)}{2\pi}\left\{2(1-\nu) + \left(1 - \frac{1}{4(1-\nu)}\right)\right\} \tag{3}$$

$$k = k_{bend} * corr; \tag{4}$$

$$E_{Eff} = \frac{9k}{4\pi a}; \tag{5}$$

## Protein concentration

Protein concentration was determined by DC protein assay (Bio-Rad).

## Focal adhesion TIRF imaging

µ-Slides (iBidi) were coated with FN (5 µg/ml) in PBS at 4°C overnight. Next, µ-Slides were gently washed once with cold PBS following incubation with EV or sEV at 4°C overnight. The following day, µ-Slides were incubated with 0.1% BSA in PBS for 30 min at 37°C to block non-specific binding sites and washed with PBS three times, and 250 µl of M199 media supplemented with 20% sEV-free FBS was added to each well. VSMCs were incubated in M199 media supplemented with 0.5% BSA overnight, detached with trypsin and resuspended to 80,000 cells/ml in M199 supplemented with 20% FBS. Cells (250 µl) were added to µ-Slides and were incubated for 30 min at 37°C. VSMC were fixed with 4% PFA for 10 min at 37°C, washed with PBS three times and were permeabilised with PBS supplemented with 0.2% Triton X-100 for 10 min at 37°C. Next, VSMC were washed three times with PBS for 5 min and were incubated with 3% BSA in PBS for 1 hr at room temperature. VSMC were incubated with anti-vinculin antibody (1:400) in blocking buffer at 4°C overnight. Next, VSMC were washed with PBS for 5 min and incubated with secondary antibody (1:500, AlexaFluor488) diluted in blocking buffer for 1 hr at room temperature. Cells were incubated with CellMaskRed (Thermo Fisher Scientific) and DAPI (0.1 mg/ml) for 5 min, washed and analysed by TIRF microscopy.

TIRF microscopic images were collected on a Nikon Ti2 upright microscope with a TIRF illuminator (Nikon) attached using a 100x oil immersion (1.49 NA) objective lens. In each fluorescent channel, the sample was excited with a laser beam at an angle indicated as follows (488 nm at 62.3°, 561 nm at 62.5°, and 640 nm at 65.5°). All emission light was filtered through a quad channel filter set (Chroma 89000) in the upper filter turret and a secondary bandpass emission filter in the lower turret position was used as follows for each channel (488 nm excited: filter BP 525/50 - Chroma; 561 nm excited: filter BP 595/50 - Chroma; 640 nm excited: filter BP 700/75 - Chroma). Light was then passed to a Hamamatsu Orca-Flash 4v3.0 sCMOS camera and exposures for channels were set as follows (488 nm: 100ms, 561 nm: 100ms, 640 nm: 50ms) with 16bit channel images collected.

Acquisition was all controlled through NIS-Elements software (Nikon). Analysis of TIRF images was completed using NIS-Elements General Analysis Module. A general description of the process follows: Cell Area measurements were determined by use of an intensity threshold on the HCS CellMask Deep Red channel at an intensity of 100 AU (grey levels or arbitrary units) above the background to mask the cell. The Cell area, whole cell Vinculin and Far-Red intensity measurements were calculated from

this binary region. In addition, an inversion of the cell mask binary was generated to have a means to measure distance to edge of the cell.

Vinculin-derived foci were masked using an intensity-based threshold (Above 10000 AU) and a size minimum for objects was set to 0.5 µm² to eliminate smaller non-foci objects from the binary mask. The foci mask was then limited to the cell mask regions to allow interpretation of the foci in the cell area. Minimum distance to the outside of the cell for each foci was calculated by measurement of the nearest distance to the inverted non-cell mask. The average distance was then recorded for each image. For each foci intensity measurements were taken for the Vinculin and FarRed channel and the Mean intensity of the foci was recorded.

## 2D VSMC migration time-lapse assay

24-well plates were incubated with FN (5 µg/ml) in PBS overnight at +4°C and treated with sEVs (10 µg/ml) diluted in PBS overnight at +4°C. VSMC (8000 cells per well) were plated on the plate in M199 supplemented with 20% sEV-free FBS and incubated for 16 hr. Cells were washed twice with DMEM supplemented with 2.5% sEV-free FBS and incubated in this media for 2hrs +/-3'-OMS. Media was replaced for a fresh aliquot and cells were imaged using Opera Phenix High Content Screening System (PerkinElmer) every 6 min 20 s for 8 hr in a transmitted light channel. Quantification of cellular velocity and directionality was performed in Harmony 4.9 (Perkin Elmer) and the following filters were applied to the raw data: 'Number of Timepoints' ≥ 5, 'First Timepoint' <5.

## 3D VSMC invasion time lapse assay

### Acquisition

Migration assay was completed using IBidi Chemotaxis µ-slides (Ibidi µ-slides #80326). VSMC were deprived overnight in M199 supplemented with 0.5% BSA, counted and mixed with Matrigel supplemented with/without sEV (5 µg total in 100 µl volume) to a final cell concentration of ca. 3x10⁶ cells/ml and stained with Draq5 according to the manufacturer protocol. Slides were left for 30 min at 37°C for gelation and chemoattractant-free medium (65 µl of M199 supplemented with 0.5% BSA) was filled through Filling Port E. Next, the empty reservoir was filled by injecting chemoattractant medium (65 µl of M199 supplemented with 20% FBS sEV-free FBS) through Filling Port C. Three slides were loaded and cells were imaged with the Opera Phenix (Perkin Elmer) high content imaging platform (10x Objective NA 0.3). Images were collected for digital phase contrast and a fluorescence channel for Draq5. Fluorescence was obtained by exciting with a 640 nm laser and emission light was filtered with a 700/75 bandpass filter to a sCMOS camera. Images were recorded every 10 min for 12 hr in several locations for each slide.

### Migration tracking and analysis of tracking

Analysis of cell motion was completed on Draq5 images using Harmony Software (Perkin Elmer) with measurements of cell area, speed and direction obtained for each cell/track and timepoint. Cells were not selected for tracking if they were in proximity to other cells at a distance of 35 µm. The measurements and object data were exported and further analysis was performed. Data for each track included total length (timepoints), speed (µm/s), Straightness, accumulated distance (µm), lateral displacement (µm), and vertical displacement (µm).

### Data analysis

This data was then analysed using an in-house script using Python library Pandas (https://github.com/Alex-biochem/eLife-VOR-RA-2023-90375, copy archived at *Kapustin, 2025*). Tracks were discarded if they were less than 3 time-points in length. Means and Standard Deviations were calculated for track length (timepoints), Speed (µm/s), Track Straightness and accumulated distance. For each track, the Parallel FMI (forward motion index) was calculated by taking a ratio of vertical displacement to accumulated distance. The means and standard deviations for these were calculated and reported for each condition similarly. Track straightness is calculated as the ratio of total displacement over total track length. Accumulated distance is the total length of the track from the first to last time point of the track.

## Clinical samples

### Patients enrollment

The study was approved by the Local Ethical committee of the Research Institute for Complex Issues of Cardiovascular Diseases (Kemerovo, Russia, protocol number 20200212, dates of approval: 12 February, 2020), and a written informed consent was provided by all study participants after receiving a full explanation of the study. The investigation was carried out in accordance with the Good Clinical Practice and the Declaration of Helsinki. Criteria of inclusion were: (1) performance of carotid endarterectomy due to chronic brain ischaemia or ischaemic stroke; (2) a signed written informed consent to be enrolled. A criterion of exclusion was incomplete investigation regardless of the reason; in this case, we enrolled another subject with similar age, gender, and clinicopathological features who met the inclusion criteria. Cerebrovascular disease (chronic brain ischaemia and ischaemic stroke) as well as comorbid conditions (arterial hypertension, chronic heart failure, chronic obstructive pulmonary disease, asthma, chronic kidney disease, diabetes mellitus, overweight, and obesity) were diagnosed and treated according to the respective guidelines of European Society of Cardiology, Global Initiative for Chronic Obstructive Lung Disease, Global Initiative for Asthma, Kidney Disease: Improving Global Outcomes, American Diabetes Association, and European Association for the Study of Obesity. eGFR was calculated according to the Chronic Kidney Disease Epidemiology Collaboration (CKD-EPI) equation. Extracranial artery stenosis was assessed using the color duplex screening (Vivid 7 Dimension Ultrasound System, General Electric Healthcare). Data on age, gender, smoking status, and pharmacological anamnesis were collected at the time of admission. In total, we enrolled 20 patients. The detailed characteristics of the study samples are presented in *Supplementary file 1*.

### Sample collection and preparation

Carotid atherosclerotic plaques (n=14) and adjacent intact arterial segments (n=14) were pairwise excised during the carotid endarterectomy and divided into 3 segments each. The first segment was snap-frozen in the optimal cutting temperature compound (Tissue-Tek, 4583, Sakura) using a liquid nitrogen and was then sectioned on a cryostat (Microm HM525, 387779, Thermo Scientific). To ensure the proper immunofluorescence examination, we prepared 8 sections (7 μm thickness), evenly distributed across the entire carotid artery segment, per slide. The second and third segments were homogenised in TRIzol Reagent (15596018, Thermo Fisher Scientific) for RNA extraction or in T-PER Tissue Protein Extraction Reagent (78510, Thermo Fisher Scientific) supplied with Halt protease and phosphatase inhibitor cocktail (78444, Thermo Fisher Scientific) for the total protein extraction according to the manufacturer's protocols. Quantification and quality control of the isolated RNA was performed employing Qubit 4 fluorometer (Q33238, Thermo Fisher Scientific), Qubit RNA BR assay kit (Q10210, Thermo Fisher Scientific), Qubit RNA IQ assay kit (Q33222, Thermo Fisher Scientific), Qubit RNA IQ standards for calibration (Q33235, Thermo Fisher Scientific) and Qubit assay tubes (Q32856, Thermo Fisher Scientific) according to the manufacturer's protocols. Quantification of total protein was conducted using BCA Protein Assay Kit (23227, Thermo Fisher Scientific) and Multiskan Sky microplate spectrophotometer (51119700DP, Thermo Fisher Scientific) in accordance with the manufacturer's protocol.

### Immunofluorescence examination

Upon the sectioning, vascular tissues were dried at room temperature for 30 min, fixed and permeabilised in ice-cold acetone for 10 min, incubated in 1% bovine serum albumin (Cat. No. A2153, Sigma-Aldrich) for 1 hr to block non-specific protein binding, stained with unconjugated mouse anti-human CD81 (M38 clone, NBP1-44861, 1:100, Novus Biologicals) and rabbit anti-human fibronectin (F14 clone, ab45688, 1:250, Abcam) primary antibodies and incubated at 4°C for 16 hr. Sections were further treated with pre-adsorbed donkey anti-mouse Alexa Fluor 488-conjugated (ab150109, 1:500, Abcam) and donkey anti-rabbit Alexa Fluor 555-conjugated secondary antibodies (ab150062, 1:500, Abcam) and incubated for 1 hr at room temperature. Between all steps, washing was performed thrice with PBS (pH 7.4, P4417, Sigma-Aldrich). Nuclei were counterstained with 4',6-diamidino-2-phenylindole (DAPI) for 30 min at room temperature (10 μg/ml, D9542, Sigma-Aldrich). Coverslips were mounted with ProLong Gold Antifade (P36934, Thermo Fisher Scientific). Sections were examined by confocal laser scanning microscopy (LSM 700, Carl Zeiss). Colocalisation analysis (n=12 images) was performed using the respective ImageJ (National Institutes of Health) plugins (Colocalisation Threshold and

Coloc2). To evaluate the colocalisation, we calculated Pearson's r above threshold (zero-zero pixels), thresholded Mander's split colocalisation coefficient (the proportion of signal in each channel that colocalises with the other channel) for both (red and green) channels, percent volume colocalised with each channel, and intensity volume above threshold colocalised with each channel in both neointima and media.

## Reverse transcription-quantitative polymerase chain reaction (RT-qPCR)

Reverse transcription was carried out utilising High Capacity cDNA Reverse Transcription Kit (4368814, Thermo Fisher Scientific). Gene expression was measured by RT-qPCR using the customised primers (500 nmol/L each, Evrogen, Moscow, Russian Federation, *Supplementary file 5*), cDNA (20 ng) and PowerUp SYBR Green Master Mix (A25778, Thermo Fisher Scientific) according to the manufacturer's protocol for Tm ≥60°C (fast cycling mode). Technical replicates (n=3 per each sample collected from one vascular segment) were performed in all RT-qPCR experiments. The reaction was considered successful if its efficiency was 90–105% and $R^2$ was ≥0.98. Quantification of the *CD9*, *CD63*, *CD81*, *COL6A3*, *EDIL3*, *CSPG4*, *TGFBI*, *FN1*, and *MYADM* mRNA levels in carotid atherosclerotic plaques (n=5) and adjacent intact arterial segments (n=5) was performed by using the $2^{-\Delta\Delta Ct}$ method. Relative transcript levels were expressed as a value relative to the average of three housekeeping genes (*ACTB*, *GAPDH*, *B2M*).

## Western blotting

Equal amounts of protein lysate (15 µg per sample) of carotid atherosclerotic plaques (n=12), adjacent intact arterial segments (n=12), and plaque-derived sEVs (n=6) were mixed with NuPAGE lithium dodecyl sulfate sample buffer (NP0007, Thermo Fisher Scientific) at a 4:1 ratio and NuPAGE sample reducing agent (NP0009, Thermo Fisher Scientific) at a 10:1 ratio, denatured at 99°C for 5 min, and then loaded on a 1.5 mm NuPAGE 4–12% Bis-Tris protein gel (NP0335BOX, Thermo Fisher Scientific). The 1:1 mixture of Novex Sharp pre-stained protein standard (LC5800, Thermo Fisher Scientific) and MagicMark XP Western protein standard (LC5602, Thermo Fisher Scientific) was loaded as a molecular weight marker. Proteins were separated by the sodium dodecyl sulphate-polyacrylamide gel electrophoresis (SDS-PAGE) at 150 V for 2 hr using NuPAGE 2-(N-morpholino)ethanesulfonic acid SDS running buffer (NP0002, Thermo Fisher Scientific), NuPAGE Antioxidant (NP0005, Thermo Fisher Scientific), and XCell SureLock Mini-Cell vertical mini-protein gel electrophoresis system (EI0001, Thermo Fisher Scientific). Protein transfer was performed using polyvinylidene difluoride (PVDF) transfer stacks (IB24001, Thermo Fisher Scientific) and iBlot 2 Gel Transfer Device (IB21001, Thermo Fisher Scientific) according to the manufacturer's protocols using a standard transfer mode for 30–150 kDa proteins (P0 – 20 V for 1 min, 23 V for 4 min, and 25 V for 2 min). PVDF membranes were then incubated in iBind Flex Solution (SLF2020, Thermo Fisher Scientific) for 1 hr to prevent non-specific binding.

Blots were probed with rabbit primary antibodies to fibronectin or GAPDH (loading control). Horseradish peroxidase-conjugated goat anti-rabbit (7074, Cell Signaling Technology) or goat anti-mouse (AP130P, Sigma-Aldrich) secondary antibodies were used at 1:200 and 1:1000 dilution, respectively. Incubation with the antibodies was performed using iBind Flex Solution Kit (SLF2020, Thermo Fisher Scientific), iBind Flex Cards (SLF2010, Thermo Fisher Scientific) and iBind Flex Western Device (SLF2000, Thermo Fisher Scientific) during 3 hr according to the manufacturer's protocols. Chemiluminescent detection was performed using SuperSignal West Pico PLUS chemiluminescent substrate (34580, Thermo Fisher Scientific) and C-DiGit blot scanner (3600–00, LI-COR Biosciences) in a high-sensitivity mode (12 min scanning). Densitometry was performed using the ImageJ software (National Institutes of Health) using the standard algorithm (consecutive selection and plotting of the lanes with the measurement of the peak area) and subsequent adjustment to the loading control (GAPDH).

## Shotgun proteomics and proteomics data analysis

Carotid atherosclerotic plaques and adjacent intact arterial segments were excised pairwise during carotid endarterectomy. Tissue segments were homogenised in T-PER Tissue Protein Extraction Reagent (Thermo Fisher Scientific), supplemented with Halt protease and phosphatase inhibitor cocktail (Thermo Fisher Scientific), for total protein extraction according to the manufacturer's protocols. Following acetone (Supelco) precipitation to remove T-PER buffer, protein pellets were resuspended in 8 M urea (Sigma-Aldrich) in 50 mM ammonium bicarbonate (Sigma-Aldrich). Protein concentration

was measured using a Qubit 4 fluorometer and QuDye Protein Quantification Kit (Lumiprobe), per manufacturer's instructions. Each protein sample (10 µg) was incubated with 5 mmol/L dithiothreitol (Sigma-Aldrich) for 1 hr at 37°C, followed by 15 mmol/L iodoacetamide (Sigma-Aldrich) for 30 min in the dark at room temperature. Samples were diluted with 7 volumes of 50 mM ammonium bicarbonate and digested for 16 hr at 37°C using trypsin gold (Promega) at a 1:50 trypsin:protein ratio (200 ng per sample). Peptides were dried using a Labconco CentriVap Centrifugal Concentrator (Labconco), dissolved in 0.1% formic acid (Sigma-Aldrich), and desalted with C18 spin tips according to the manufacturer's instructions. After desalting, peptides were dried again and re-dissolved in 0.1% formic acid for LC-MS/MS analysis. Each sample was analysed in technical duplicate. Approximately 500 ng of peptides were subjected to shotgun proteomics using UHPLC-MS/MS with ion mobility on a TimsToF Pro mass spectrometer (Bruker Daltonics) coupled to a nanoElute UHPLC system (Bruker Daltonics). Separation was performed in two-column mode using an Acclaim PepMap 5 mm Trap Cartridge and Bruker Ten separation column (C18 ReproSil AQ, 100 mm × 0.75 mm, 1.9 µm, 120 Å; Bruker Daltonics) at a flow rate of 500 nL/min and 50°C. Mobile phase A was water/0.1% formic acid; mobile phase B was acetonitrile/0.1% formic acid. The UHPLC gradient was set as: 2–30% phase B over 16 min, 30–38% phase B over 5 min, and 38–95% phase B over 3 min, followed by 6 min washing at 95% phase B. Electrospray ionisation was achieved using a CaptiveSpray ion source with a capillary voltage of 1600 V, nitrogen flow of 3 L/min, and source temperature of 180°C. Mass spectrometry acquisition was performed in data-independent acquisition mode with trapped ion mobility spectrometry (TIMS, in PASEF mode, diaPASEF). The diaPASEF method included five cycles with three or two mobility windows each, totaling 13 mass width windows (50 Da width) spanning an m/z range of 457.81–1046.80 and ion mobility range (1 /K0) of 0.85–1.18 Vs/cm². Collision energy was programmed as a function of ion mobility, increasing linearly from 20 eV at 1 /K0 = 0.85 Vs/cm² to 59 eV at 1 /K0 = 1.3 Vs/cm². Protein identification was performed using DIA-NN software (v1.8) (*Demichev et al., 2020*) with the human SwissProt protein database (https://www.uniprot.org; accessed 14 December 2021; taxonomy: Human [9606]; uploaded 2 March 2021). A library-free search was carried out with default settings and the generated library comprised 20,338 proteins and 8,454,641 precursors. Search parameters were set to parent and fragment mass error tolerance of 10 ppm, *protein and peptide FDR* <1%, and allowance for up to two missed cleavage sites. Carbamidomethylation of cysteine was set as a fixed modification.

Further analysis was performed in **R** (version 3.6.1; R Core Team, 2019). Proteins with missing values in more than two samples were excluded, and remaining missing values were imputed using k-nearest neighbours via the *impute* package. Data was log-transformed and quantile-normalised prior to differential expression analysis with the *limma* package (*Ritchie et al., 2015*). Sample clustering was performed using sparse partial least squares discriminant analysis in the *MixOmics* package (*Rohart et al., 2017*). Data visualisation utilised the *ggplot2* (*Wickham, 2011*) and *EnhancedVolcano* (*Ubuntu, 2018*) packages.

## Statistics

Data were analysed using GraphPad Prizm (version 8.4.3). CD63-beads assay, NanoView, adhesion assay, FA quantification were analysed by one-way ANOVA test. Two group CD63-bead assays were analysed by non-paired T-test. 2D migration multiple comparison data and VSMC invasion multiple comparison data were analysed using non-parametric Kruskal-Wallis test. Two group invasion data were analysed using a non-parametric Kolmogorov-Smirnov test. Clinical data for two groups was analysed by non-paired T-test and multiple comparison was analysed by Mann-Whitney U-test. All analysis was conducted using PRISM software (GraphPad, San Diego, CA). Values of p<0.05 were considered statistically significant.

## Acknowledgements

ANK and CS were supported by BHF-PG/17/37/33023. TI was supported by BHF PG/20/6/34835 and FS/14/30/30917. REC was supported by National Institutes of General Medical Sciences grant R01GM134531. AK, MS and LB were supported by the Ministry of Science and Higher Education of the Russian Federation-Complex Program of Basic Research under the Siberian Branch of the Russian Academy of Sciences within the Basic Research Topic of Research Institute for Complex Issues of Cardiovascular Diseases № 0419-2021-001. We are thankful to Dr Adam A Walters for the flow

cytometry consultancy and help. The funders were not involved in study design, data collection and interpretation, or the decision to submit the work for publication.

## Additional information

### Competing interests

Alexander N Kapustin: Currently an employee and shareholder of AstraZeneca. The other authors declare that no competing interests exist.

### Funding

| Funder | Grant reference number | Author |
|---|---|---|
| British Heart Foundation | BHF-PG/17/37/33023 | Catherine M Shanahan |
| British Heart Foundation | PG/20/6/34835 | Thomas Iskratsch |
| British Heart Foundation | FS/14/30/30917 | Thomas Iskratsch |
| National Institute of General Medical Sciences | R01GM134531 | Richard E Cheney |
| Ministry of Science and Higher Education of the Russian Federation | 0419-2021-001 | Anton Kutikhin |

The funders had no role in study design, data collection and interpretation, or the decision to submit the work for publication.

### Author contributions

Alexander N Kapustin, Conceptualization, Data curation, Funding acquisition, Investigation, Visualization, Methodology, Writing – original draft, Project administration, Writing – review and editing; Sofia Serena Tsakali, Data curation, Formal analysis, Validation, Investigation, Visualization, Methodology, Writing – review and editing; Meredith Whitehead, Data curation, Formal analysis, Investigation, Visualization, Methodology, Writing – review and editing; George Chennell, Data curation, Software, Formal analysis, Investigation, Methodology, Writing – review and editing; Meng-Ying Wu, Data curation, Software, Formal analysis, Writing – review and editing; Chris Molenaar, Data curation, Software, Formal analysis, Methodology, Writing – review and editing; Anton Kutikhin, Resources, Data curation, Formal analysis, Funding acquisition, Validation, Investigation, Visualization, Methodology, Project administration, Writing – review and editing; Yimeng Chen, Sadia Ahmad, Investigation, Visualization, Writing – review and editing; Leo Bogdanov, Maxim Sinitsky, Arseniy Lobov, Data curation, Investigation, Methodology, Writing – review and editing; Kseniya Rubina, Data curation, Investigation, Writing – review and editing; Aled Clayton, Conceptualization, Resources, Investigation, Writing – review and editing; Frederik J Verweij, Dirk Michiel Pegtel, Resources, Methodology, Writing – review and editing; Simona Zingaro, Formal analysis, Investigation, Writing – review and editing; Bozhana Zainullina, Investigation, Methodology, Writing – review and editing; Dylan Owen, Thomas Iskratsch, Resources, Data curation, Formal analysis, Investigation, Methodology, Writing – review and editing; Maddy Parsons, Resources, Formal analysis, Methodology, Writing – review and editing; Richard E Cheney, Martin James Humphries, Conceptualization, Resources, Methodology, Writing – review and editing; Derek T Warren, Resources, Formal analysis, Investigation, Methodology, Writing – review and editing; Mark Holt, Data curation, Software, Formal analysis, Investigation, Visualization, Methodology, Writing – review and editing; Catherine M Shanahan, Conceptualization, Resources, Formal analysis, Supervision, Funding acquisition, Methodology, Project administration, Writing – review and editing

### Author ORCIDs

Alexander N Kapustin ⓘ https://orcid.org/0000-0003-3552-830X
Sofia Serena Tsakali ⓘ https://orcid.org/0009-0005-7564-6724
Meredith Whitehead ⓘ https://orcid.org/0000-0002-2211-9673
George Chennell ⓘ https://orcid.org/0000-0001-5859-361X

Anton Kutikhin https://orcid.org/0000-0001-8679-4857
Sadia Ahmad https://orcid.org/0000-0002-6206-6764
Leo Bogdanov https://orcid.org/0000-0003-4124-2316
Maxim Sinitsky https://orcid.org/0000-0002-4824-2418
Kseniya Rubina https://orcid.org/0000-0002-7166-7406
Aled Clayton https://orcid.org/0000-0002-3087-9226
Arseniy Lobov https://orcid.org/0000-0002-0930-1171
Dylan Owen https://orcid.org/0000-0002-5284-2782
Maddy Parsons https://orcid.org/0000-0002-2021-8379
Richard E Cheney https://orcid.org/0000-0001-6565-7888
Derek T Warren https://orcid.org/0000-0003-0346-7450
Martin James Humphries https://orcid.org/0000-0002-4331-6967
Thomas Iskratsch https://orcid.org/0000-0002-3738-7830
Mark Holt https://orcid.org/0000-0001-7775-8539
Catherine M Shanahan https://orcid.org/0000-0002-8352-8171

### Ethics

Human subjects: Yes, this work involved the use of human tissue samples. No direct research involving living human subjects was performed. All human tissue was obtained from donors with appropriate ethical approval and in accordance with relevant regulations. Tissue procurement was conducted under the Human Tissue Act (2004, UK) and with approval from the local research ethics committee. No patient-identifying information was used or accessed.For clinical samples: The study was approved by the Local Ethical committee of the Research Institute for Complex Issues of Cardio-vascular Diseases (Kemerovo, Russia, protocol number 20200212, dates of approval: 12 February 2020), and a written informed consent was provided by all study participants after receiving a full explanation of the study. The investigation was carried out in accordance with the Good Clinical Practice and the Declaration of Helsinki. Criteria of inclusion were: (1) performance of carotid endarterectomy due to chronic brain ischemia or ischemic stroke; (2) a signed written informed consent to be enrolled.

Reviewer #1 (Public review): https://doi.org/10.7554/eLife.90375.3.sa1
Reviewer #2 (Public review): https://doi.org/10.7554/eLife.90375.3.sa2
Author response https://doi.org/10.7554/eLife.90375.3.sa3

---

## Additional files

### Supplementary files

Supplementary file 1. Anonymised patient's clinical data.

Supplementary file 2. Quantitative proteomics results for carotid atherosclerotic plaques and adjacent intact vascular segments excised during carotid endarterectomy.

Supplementary file 3. Differentially expressed proteins in carotid atherosclerotic plaques and adjacent intact vascular segments identified by quantitative proteomics.

Supplementary file 4. Proteins uniquely identified in intact vascular segments by proteomics.

Supplementary file 5. Proteins uniquely identified in carotid atherosclerotic plaques by proteomics.

Supplementary file 6. Unique and common proteins in vascular smooth muscle cell-derived EVs and sEVs.

Supplementary file 7. Complete mass spectrometry data for extracellular vesicle (EV) and small extracellular vesicle (sEV) proteomes.

Supplementary file 8. Protein functional enrichment analysis in extracellular vesicles (EV) and small extracellular vesicles (sEV).

MDAR checklist

### Data availability

Source numerical data for all graphs, as well as the original raw western blotting and microscopy data, have been deposited and are publicly available at: https://osf.io/tykcq/.

The following dataset was generated:

| Author(s) | Year | Dataset title | Dataset URL | Database and Identifier |
|-----------|------|---------------|-------------|-------------------------|
| Kapustin A | 2025 | Matrix-associated extracellular vesicles modulate smooth muscle cell adhesion and directionality by presenting collagen VI | https://doi.org/10.17605/OSF.IO/TYKCQ | Open Science Framework, 10.17605/OSF.IO/TYKCQ |

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
