## [Editor Report · eLife Assessment]

This paper explores the role of extracellular vesicles in providing extracellular matrix signals for migration of vascular smooth muscle cells. The evidence, based on cell culture experiments and supporting imaging of human samples, is mostly **convincing**. The paper will be **valuable** for researchers investigating cell migration during vessel repair and atherogenesis.

---

## [Referee Report · Reviewer #1 (Public review)]

In this revised submission from Kapustin et al., the authors have made significant changes to the manuscript. Namely, the authors have addressed several of the major issues with the original submission, providing a more concrete link between fibronectin and the secretion of extracellular vesicles. Additionally, the authors have moderated some of the conclusions to better suit the rigor of the experimental results and limitations of their approach. Generally, the findings convey an interesting cell autonomous pathway in which smooth muscle cells sense fibronectin, which canonically is a proinflammatory substrate with activating properties in many tissues. Fibronectin-mediated integrin signaling stimulates secretion of small extracellular vesicles containing collagen VI which is deposited into the surrounding extracellular matrix. Collagen VI itself gleaned from extracellular vesicle secretion seems to further alter smooth muscle cell morphodynamics. For this later finding, much of the mechanism behind collagen VI vesicle loading and secretion has yet to be worked out. The authors provide evidence of extracellular vesicles containing collagen VI trapped in fibronectin in atherosclerotic plaques providing a nice validation of their in vitro findings in a diseased human cohort. Some limitations do still exist in the manuscript in its current form such as the assessment of the vesicle origins, contents and their association with the actin cytoskeleton; however, the rigor and execution are much improved from the preceding version. Overall, the pathobiology underlying vascular smooth muscle remodeling in disease states is a critical area of research that warrants further exploration.

---

## [Referee Report · Reviewer #2 (Public review)]

The findings in the current manuscript are interesting and valuable contributions to the fields of vascular biology and extracellular vesicle-related mechanisms. They suggest a potential role for smooth muscle cell-derived extracellular vesicles in presenting Type VI collagen to cells to orchestrate their migration, with proposed relevance to aberrant smooth muscle cell movements in the progression of atherosclerotic lesions. A wide range of assays are utilized to test various aspects of this working model, with the resulting data being largely solid and supporting several of the interpretations articulated by the authors. The revised manuscript has adequately addressed key weaknesses.

The authors present data suggesting a working model in which vascular smooth muscle cells (vSMCs) are stimulated by fibronectin (FN) to generate small extracellular vesicles (sEVs) that harbor Type VI Collagen (collagen VI). These collagen VI-associated sEVs are suggested to accumulate in the extracellular matrix (ECM) and influence cell migration and adhesion dynamics, potentially contributing to disease progression in atherosclerosis. Majors strengths of this manuscript include robust imaging data and the inclusion of human-derived samples in their analysis. The authors also make a reasonable attempt to provide data to support the potential existence of these mechanistic connections, though some minor questions remain regarding data interpretation. The authors largely achieved their aims of finding evidence consistent with their interpretations, and they have presented logical support for their conclusions while acknowledging important limitations and caveats to their current study. This work will likely have a sustained impact on the field of sEV biology and potential intersections with vascular biology, including their methodology e.g., imaging approaches. As biologists continue to explore the role of sEVs in physiological and pathological processes, this work raises an interesting aspect that must be considered more broadly, and that is, what is the role of sEVs that are ECM-associated and not necessarily internalized by recipient cells? Are there discrete mechanisms that govern their role in maintaining and/or disrupting normal physiological processes? This manuscript makes an attempt to address these unresolved yet critical questions.

---

## [Author Response]

The following is the authors’ response to the original reviews.

**Reviewer #1 (Public Review):**
SummaryIn this investigation Kapustin et al. demonstrate that vascular smooth muscle cells (VSMCs) exposed to the extracellular matrix fibronectin stimulates the release of small extracellular vesicles (sEVs). The authors provide experimental evidence that stimulation of the actin cytoskeleton boosts sEV secretion and posit that sEVs harbor both fibronectin and collagen IV protein themselves which also, in turn, alter cell migration parameters. It is well established that fibronectin is associated with increased cell migration and adherence; therefore, this association with VSMCs is not novel.

The reviewer is correct that FN has been associated with migration and adherence in previous studies. However we have extended these observations to show that the extracellular fibronectin matrix stimulates small extracellular vesicle (sEVs) secretion by modulating the actin cytoskeleton. We also showed that sEVs are trapped in the extracellular matrix and that by presenting collagen VI induce early focal adhesion formation, reduce excessive cellular spreading and guide cell invasion directionality though a 3D matrix. Hence, sEVs mediate cell-matrix cross talk and change cell behaviour in the context of fibronectin matrix. This is critically important for vasculature where regulated VSMC invasion is essential for repair with its deregulation leading to pathology.

The authors purport that sEV are largely born of filopodia origin; however, this data is not well executed and seems generally at odds with the presented data.

Our experimental data showed that CD63 MVs are associated with filopodia in fixed and live cells (Fig 2E, 2F and Video S1) and that inhibition of filopodia formation using the formin inhibitor, SMIFH2 reduced sEV secretion on FN (Fig 2B). However, we agree with the reviewer that further studies are required to connect sEV secretion to filopodia. To address this we have provided further data analysis but also toned down our conclusions regarding this point: . Changes include:

(1) Title: Matrix-associated extracellular vesicles modulate smooth muscle cell adhesion and directionality by presenting collagen VI.

(2) Results, section title: 2. FN-induced sEV secretion is modulated by Arp2/3 and formin-dependent actin cytoskeleton remodelling

(3) Results, page 6 Line 27-44 and conclusion page 7, Ln 3 “Interestingly, CD63+ MVBs can be observed in filopodia-like structures suggesting that sEV secretion can also occur spatially via cellular protrusion-like filopodia but more studies are needed to confirm this hypothesis.”

(4) Discussion, page 12, line 19. “Curiously we observed CD63+ MVB transport toward the filopodia tips as well as inhibition of sEV-secretion with filopodia formation inhibitors suggesting that sEV secretion can be directly linked to filopodia but further studies are needed to define the contribution of this pathway to the overall sEV secretion by cells.”

Similarly, the effect of sEVs on parameters of cell migration has almost no magnitude of effect, making mechanism exploration somewhat nebulous.

VSMC are mesenchymal-type cells with a low migration rate and we agree that the changes in the motility are not of great magnitude even for the positive controls suggesting that this is a complex, multifactorial process for VSMCs. In our experiments we collected data from >5000 individual cells to measure the average speed and found that fibronectin matrix on its own increased VSMC speed from ~0.61 um/min to ~0.68 μm/min (~12% raise) which was statistically significant (Fig 5A). Addition of a sEV inhibitor caused a modest but significant decrease in cellular speed. Interestingly, addition of ECM-associated sEVs did not influence cell speed in 2D or 3D assays. However in a 3D model we observed a 22% change in cell directionality (Fig 5G) and a 235% change in cell alignment index (FMI, Fig 5H) which we believe is very strong evidence that VSMC-derived sEVs are involved in a regulation of VSMC invasion directionality. These data are also in agreement with sEV effects in tumour cells (Sung et al., 2015) though this previous study did not identify the factor driving the directionality and we think our Collagen VI data extends significantly these previous observations.

Results, page 9: “Hence, ECM-associated sEVs have modest influence on VSMC speed but influence VSMC invasion directionality.”.

Lastly, the proposed mechanism of VSMCs responding to, and depositing, ECM proteins via sEVs was not rigorously executed; again, making the conclusions challenging for the reader to interpret.

We appreciate the reviewer’s comment regarding the mechanistic aspects of VSMCs responding to and depositing ECM proteins via sEVs. In our revised manuscript, we have expanded the data demonstrating that sEVs can be retained within the extracellular matrix (see Figs 3A, 3B, S3A, S3B). Additionally, we show that collagen VI is present on the surface of sEVs, where it may modulate cell adhesion and influence the directionality of cell invasion (Fig 7E). Our results further indicate that both fibronectin (FN) and collagen VI can be recycled through multivesicular bodies (see Figs S3C, S3D, S3E–S3G). However, we acknowledge that the precise mechanisms governing the selective loading of ECM proteins onto sEVs, as well as the specific contributions of sEVs to overall ECM organization, remain to be fully elucidated and warrant further investigation. Based on our current evidence, we propose that collagen VI–loaded sEVs act primarily in a signaling capacity by modulating focal adhesion formation but are not directly involved in ECM structural remodeling.

Results, page 7: To quantify ECM-trapped sEVs we applied a modified protocol for the sequential extraction of extracellular proteins using salt buffer (0.5M NaCl) to release sEVs which are loosely-attached to ECM via ionic interactions, followed by 4M guanidine HCl buffer (GuHCl) treatment to solubilize strongly-bound sEVs (Fig S3A) [42]. We quantified total sEV and characterised the sEV tetraspanin profile in conditioned media, and the 0.5M NaCl and GuHCl fractions using ExoView. The total particle count showed that EVs are both loosely bound and strongly trapped within the ECM. sEV tetraspanin profiling showed differences between these 3 EV populations. While there was close similarity between the conditioned media and the 0.5M NaCl fraction with high abundance of CD63+/CD81+ sEVs as well as CD63+/CD81+/CD9+ in both fractions (Fig S3A). In contrast, the GuHCl fraction was particularly enriched with CD63+ and CD63+/CD81+ sEVs with very low abundance of CD9+ EVs (Fig S3A). The abundance of CD63+/CD81+ sEVs was confirmed independently by a CD63+ bead capture assay in the media and loosely bound fractions (Fig S3B).

Results, page 7: We previously found that the serum protein prothrombin binds to the sEV surface both in the media and MVB lumen showing it is recycled in sEVs and catalyses thrombogenesis being on the sEV surface43. So we investigated whether FN can also be associated with sEV surface where it can be directly involved in sEV-cell cross-talk43. We treated serum-deprived primary human aortic VSMCs with FN-Alexa568 and found that it was endocytosed and subsequently delivered to early and late endosomes together with fetuin A, another abundant serum protein that is a recycled sEV cargo and elevated in plaques (Figs S3C and S3D). CD63 visualisation with a different fluorophore (Alexa488) confirmed FN colocalization with CD63+ MVBs (Fig S3E). Next, we stained non-serum deprived VSMC cultured in normal growth media (RPMI supplemented with 20% FBS) with an anti-FN antibody and observed colocalization of CD63 and serum-derived FN. Co-localisation was reducd likely due to competitive bulk protein uptake by non-deprived cells (Fig S3F). Notably, when we compared FN distribution in sparsely growing VSMCs versus confluent cells we found that FN intracellular spots, as well as colocalization with CD63, completely disappeared in the confluent state (Fig S3F and S3G). This correlated with nearly complete loss of CD63+/CD81+ sEV secretion by the confluent cells indicating that confluence abrogates intracellular FN trafficking as well as sEV secretion by VSMCs (Fig S3H). Finally, FN could be co-purified with sEVs from VSMC conditioned media (Fig S3I) and detected on the surface of sEVs by flow cytometry confirming its loading and secretion via sEVs (Fig 3C).

Results: page 10 Collagen VI was the most abundant protein in VSMC-derived sEVs (Fig 7B, Table S7) and was previously implicated in the interaction with the proteoglycan NG2[53] and suppression of cell spreading on FN[54]. To confirm the presence of collagen VI in ECM-associated sEVs we analysed sEVs extracted from the 3D matrix using 0.5M NaCl treatment and showed that both collagen VI and FN are present (Fig 7D). Next, we analysed the distribution of collagen VI using dot-blot. Alix staining was bright only upon permeabilization of sEV indicating that it is preferentially a luminal protein (Fig 7E). On the contrary, CD63 staining was similar in both conditions showing that it is surface protein (Fig 7E). Interestingly, collagen VI staining revealed that 40% of the protein is located on the outside surface with 60% in the sEV lumen (Fig 7E).

Discussion page 12. “In fact, we observed that an extensive secretion of sEVs effectively ceased protrusion activity; also VSMCs acquired a rounded morphology when “hovering” over the FN matrix decorated with sEVs (data not shown). Hence, it will be interesting in future studies to investigate whether sEVs can stimulate Rho activity by presenting adhesion modulators—particularly collagen VI—on their surface, thereby guiding cell directionality during invasion..”

Discussion, page 14 “In summary, cooperative activation of integrin signalling and F-actin cytoskeleton pathways results in the secretion of sEVs which associate with the ECM and play a signalling role by controling FA formation and cell-ECM crosstalk. Further studies are needed to test these mechanisms across various cell types and ECM matrices.

StrengthsThe authors provide a comprehensive battery of cytoskeletal experiments to test how fibronectin and sEVs impact both sEV release and vascular smooth muscle cell migratory activation.

We appreciate this comment reflecting our efforts to apply a range of orthogonal methods to show the role of the integrin/actin cytoskeleton in ECM-stimulated sEV secretion.

WeaknessesUnfortunately, this article suffers from many weaknesses. First, the rigor of the experimental approach is low, which calls into question the merit of the conclusions. In this vein, there is a lack of proper controls or inclusion of experiments addressing alternative explanations for the phenotype or lack thereof.

We acknowledge this comment and agree that there was not sufficient evidence to conclude that sEV secretion occurs via filopodia despite the microscopy/inhibitory data so this claim has now been excluded from the study. However we believe that our experimental data does clearly show that FN stimulates the secretion of collagenVI-loaded sEVs which are trapped by the ECM and have the capacity to modulate VSMC adhesion and invasion directionality. To support this, we have now extended the dataset in the revised version:

(1) In addition to the use of inhibitors and live cell analysis we have added quantitative data confirming that a large proportion of CD63+ endosomes are associated with F-actin/cortactin tails and this colocalization is increased upon the inhibition of sEV secretion with 3-OMS (Fig 2D, Fig S2B).

(2) We developed a method to extract ECM-associated sEVs and quantified/characterized these using ExoView Assays further confirming significant sEV entrapment by the ECM (Figs 3B, S3A, S3B).

(3) We extended the controls to confirm FN delivery to CD63+ endosomes and showed that FN recycling is stopped upon reaching cell confluence (Figs S3F, S3G and Fig S3H).

(4) We included more intensive characterisation of human atherosclerotic plaque morphology (H&E, Masson’s trichrome staining, Orcein, elastin fibers staining) to confirm predominant accumulation of sEV in the neointima (Figs S4A, S4B and S4C). We also excluded an endothelial origin for the CD81+ sEVs (Fig 4G).

(5) We included individual cellular tracks to the 2D migration analysis to confirm the statistical significance and concluded that ECM-associated sEVs regulate cell invasion directionality but not the cell speed (Figs 5A and 5B).

(6) We showed surface localisation of collagen VI on sEVs confirming that it can activate signalling pathways leading to early FA formation on the FN matrix (Figs 7D and 7E).

(7) We included alternative explanations for some of our data in the discussion.

**Reviewer #2 (Public Review):**
Extracellular vesicles have recently gained significant attention across a wide variety of fields, and they have therefore been implicated in numerous physiological and pathophysiological processes. When such a discovery and an explosion of interest occur in science, there is often much excitement and hope for answers to mechanisms that have remained elusive and poorly understood. Unfortunately, there is an equal amount of hype and overstatement that may also be put forth in the name of "impact", but this temptation must be avoided so that scientists and the broader public are not misled by overreaching interpretations and statements that lack rigorous and fully convincing evidence.

Thank you for your comment and we agree that investigating sEVs is particularly challenging due to the their heterogeneity and nano-size, as well as complex biogenesis mechanisms. ECM-associated sEVs is a very new direction for the EV field but one that is particularly relevant to the vasculature where cells must invade through a thick ECM and where the accumulation of ECM-bound EVs is a unique and documented phenomenon. To further strengthen out conclusions we have included new data to support our statements but also excluded statements re: filopodia as the origin of sEVs, that are out of scope of our study and need to be investigated further.

The study presented by Kapustin et al. is certainly intriguing and timely, and it offers an interesting working hypothesis for the fields of extracellular vesicles and vascular biology to consider. The authors do a reasonable job at detecting these small extracellular vesicles, though some aspects of data presentation are missing such as full Western blots with accompanying size markers for the viewer to more fully appreciate that data and comparisons being made (see Figures 1 and 7).

We agree with the reviewer and have now included molecular weight markers (Fig 1F, 7C, 7D, S3I, S4E) and provided all original western blot scans (uncropped and unedited) to the eLife editor.

Much of the imaging data from cell-based experiments is strong and conducted with many cutting-edge tools and approaches. That said, the static images and the dynamic imaging fall short of being fully convincing that the small extracellular vesicles found in the neighboring extracellular matrix are indeed being deposited there via the smooth muscle cell filopodia. Many of the lines of evidence presented suggest that this could occur, but alternative hypotheses also exist that were not fully ruled out, such as the ECM-deposited vesicles were secreted more from the soma and/or the lamellipodia that are also emitted and retracted from the cells. In particular, the authors show very nice dynamic imaging (Supplementary Figure S2A and Supplemental Video S1) that is interpreted as "extracellular vesicles being released from the cell" and these are seen as "bursts" of fluorescent signal; however, none of these appear to occur in filopodia as they appear within the cell proper (a "burst" of signal vs. a more intense "streak" of signal), which would be a stronger and more consistent observation predicted by the working model proposed by the authors.

Our live and fixed cell microscope data as well as inhibitor analysis showed that sEV secretion can be associated with the filopodia. However we agree with the reviewer that the data generated using pHluoron GFP marker clearly indicate that the majority of sEVs are secreted from the cell soma toward the ECM:

To reflect this, we have added further changes:

(1) Title: Matrix-associated extracellular vesicles modulate smooth muscle cell adhesion and directionality by presenting collagen VI.

(2) Results, section title: 2. FN-induced sEV secretion is modulated by Arp2/3 and formin-dependent actin cytoskeleton remodelling

(3) Results, page 6 Line 27-36 “Formins and the Arp2/3 complex play a crucial role in the formation of filopodia, a cellular protrusion required for sensing the extracellular environment and cell-ECM interactions36. To test whether MVBs can be delivered to filopodia, we stained VSMCs for Myosin-10 (Myo10)37. We observed no difference between total filopodia number per cell on plastic or FN matrices (n=18±8 and n=14±3, respectively) however the presence of endogenous CD63+ MVBs along the Myo10-positive filopodia were observed in both conditions (Fig 2E, arrows). Filopodia have been implicated in sEV capture and delivery to endocytosis “hot-spots”38, so next we examined the directionality of CD63+ MVB movement in filopodia by overexpressing Myo10-GFP and CD63-RFP in live VSMCs. Importantly, we observed anterograde MVB transport toward the filopodia tip (Fig 2F and Supplementary Video S2) indicative of MVB secretion”.

(4) Results, page 6, Ln 37-44 “We also attempted to visualise sEV release in filopodia using CD63-pHluorin where fluorescence is only observed upon the fusion of MVBs with the plasma membrane39. Using total internal reflection fluorescence microscopy (TIRF) we observed the typical “burst”-like appearance of sEV secretion at the cell-ECM interface in full agreement with an earlier report showing MVB recruitment to invadopodia-like structures in tumor cells18 (Fig S2B and Supplementary Video S1). Although we also observed an intense CD63-pHluorin staining along filopodia-like structures we were not able to detect typical “burst”-like events to confirm sEV secretion in filopodia. (Fig S2C and Supplemental Video S1)”.

(5) Results, page 7 Ln 3 “Interestingly, CD63+ MVBs can be observed in filopodia-like structures suggesting that sEV secretion can also occur spatially via cellular protrusion-like filopodia but more studies are needed to confirm this hypothesis.”

(6) Discussion, page 12, line 19. “Curiously we observed CD63+ MVB transport toward the filopodia tips as well as inhibition of sEV-secretion with filopodia formation inhibitors suggesting that sEV secretion can be directly linked to filopodia but further studies are needed to define the contribution of this pathway to the overall sEV secretion by cells.”

Imaging of related human samples is certainly a strength of the paper, and the authors are commended for attempting to connect the findings from their cell culture experiments to an important clinical scenario. However, the marker selected for marking extracellular vesicles is CD81, which has been described as present on the endothelium of atherosclerotic plaques with a proposed role in the recruitment of monocytes into diseased arteries (Rohlena et al. Cardiovasc Res 2009). More data should address this potentially confounding interpretation of the signals presented in images within Figure 4.

We thank the reviewer for this insightful comment that the sEV marker CD81 can originate from endothelial cells in agreement with Rohlena et al., 2009. To address this we investigated the spatial overlap between CD81 and the endothelial marker, CD31. We observed very strong CD81 staining in the intact endothelial cell (intima) layer and occasional CD31 positive cells in the neointima. Importantly, quantification of colocalization confirmed that 80% of CD81 in the neointima does not overlap with CD31 excluding an endothelial origin of these sEVs. (Fig 4G). Moreover, we included complete morphological characterisation of the atherosclerotic plaques confirming that CD81 sEVs were primarily observed in the neointima where VSMCs constitute the cellular majority (Fig S4A, S4B, S4C and S4D).

On a conceptual level, the idea that the small extracellular vesicles contain Type VI Collagen, and this element of their cargo is modulating smooth muscle cell migration, is an intriguing aspect of the authors' working model. Nevertheless, the evidence supporting this potential mechanism does not quite fit together as presented. It is not entirely clear how the collagen VI within the vesicles is somehow accessed by the smooth muscle cell filopodia during migration. Are the vesicles lysed open once on the extracellular matrix? If so, what is the proposed mechanism for that to occur? If not, how are the adhesion molecules on the smooth muscle cell surface engaging the collagen VI fibers that are contained within the vesicles? This aspect of the model does not quite fit together with the proposed mechanism and may be an interesting speculative interpretation, warranting further investigation, but it should not be considered a strong conclusion with sufficient convincing data supporting this idea.

We thank the reviewer for their insightful comments regarding the mechanism by which collagen VI associated with sEVs could modulate smooth muscle cell adhesion and migration. To clarify, our new data suggest that collagen VI is predominantly present on the surface of the sEVs, as evidenced by Fig 7E. This surface localization strongly implies that collagen VI can be directly accessed by cell surface adhesion receptors, without the need for vesicle lysis or opening. While we cannot entirely rule out all alternative mechanisms, we consider vesicle rupture or lysis within the extracellular matrix to be a highly unlikely route for collagen VI exposure, given the known stability of sEVs under physiological conditions. We have added these points to clarify:

(1) Results, page 10, Ln 45 “To confirm the presence of collagen VI in ECM-associated sEVs we analysed sEVs extracted from the 3D matrix using 0.5M NaCl treatment and showed that both collagen VI and FN are present (Fig 7D). Next, we analysed the distribution of collagen VI using dot-blot. Alix staining was bright only upon permeabilization of sEV indicating that it is preferentially a luminal protein (Fig 7E). On the contrary, CD63 staining was similar in both conditions showing that it is surface protein (Fig 7E). Interestingly, collagen VI staining revealed that 40% of the protein is located on the outside surface with 60% in the sEV lumen (Fig 7E).”

(2) Discussion, page 13, Ln 2 “Hence, it will be interesting in future studies to investigate whether sEVs can stimulate Rho activity by presenting adhesion modulators—particularly collagen VI—on their surface, thereby guiding cell directionality during invasion..”

(3) Discussion, page 14, Ln 30: In addition to collagen VI the unique adhesion cluster in VSMC-derived sEVS also includes EGF-like repeat and discoidin I-like domain-containing protein (EDIL3), transforming growth factor-beta-induced protein ig-h3 (TGFBI) and the lectin galactoside-binding soluble 3 binding protein (LGALS3BP) and these proteins are also directly implicated in activation of integrin signalling and cellular invasiveness85-87. Although we found that collagen VI plays the key role in sEV-induced early formation of FAs in VSMCs, it is tempting to speculate that the high sEV efficacy in stimulating FA formation is driven by cooperative action of this unique adhesion complex on the sEVs surface and targeting this novel sEV-dependent mechanism of VSMC invasion may open-up new therapeutic opportunities to modulate atherosclerotic plaque development or even to prevent undesired VSMC motility in restenosis. .

(4) Abstract Figure

On a technical level, some of the statistical analysis is not readily understood from the data presented. It is very much appreciated that the authors show many of the graphs with technical and biological replicate values in addition to the means and standard deviations (though this is not clearly stated in all figure legends). However, in figures such as Figure 5, there are bars shown and indicated to be different by statistical comparison (see panel B in Figure 5). It is not clear how the values for Group 1 (no FN, no 3-OMS, no sEV) are statistically different (denoted by three asterisks but no p value provided in the legend) than Group 3 (no FN, 3-OMS added, no sEV), when their means and standard deviations appear almost identical. If this is an oversight, this needs to be corrected. If this is truly the outcome, further explanation is warranted. A higher level of transparency in such instances would certainly go a long way in helping address the current crisis of mistrust within the scientific community and at the interface with society at-large.

We thank the reviewer for their careful reading and important comments on the statistical analysis. We acknowledge that the technical and biological replicate data were not clearly reported in all figure legends and that the statistical approach for Figures 5A and 5B required clarification. In response, we have made several changes for greater transparency and rigor:

First, we have now explicitly included the numbers of biological replicates (N) and technical replicates (n) in all relevant figure legends for Figures 1–7. In addition, the number of individual cell tracks is now annotated for the migration/invasion analyses, along with the mean values for each dataset.

Upon review, we found that the original statistical analyses for Figures 5A and 5B were conducted using pooled averaged data. To address this, we have repeated the statistical tests using pooled individual cell track data, applying the Kruskal–Wallis test with Dunn’s multiple comparison correction. This more stringent approach revealed revised p-values, which are now indicated in Figures 5A and 5B.

With these corrections, we reconfirm our major findings: In the 2D model, fibronectin (FN) coating promotes VSMC velocity, while inhibition of sEV secretion with 3-OMS leads to reduced cell speed (Fig. 5A). Addition of sEVs to the ECM had no effect on VSMC speed at baseline but did rescue cell speed and distance in the presence of 3-OMS, consistent with EVs acting primarily on invasion directionality rather than speed in both 2D and 3D models (Fig. 5A, 5D). Furthermore, sEVs continue to significantly impact VSMC invasion directionality (Figs. 5G, 5H), in agreement with previous reports in tumor cells (Sung et al., 2015).

In summary, we have implemented the following revisions:

(1) Figures 5A and 5B: Individual cell track data are now shown, and statistical analyses have been repeated using the Kruskal–Wallis test with Dunn’s multiple comparisons.

(2) Figure legends and results sections: Numbers of biological and technical replicates, as well as individual data points, are now clearly stated.

Results, page 9, line 14: The text has been updated to clarify the statistical approach and major findings as described above.

We hope that these changes address the reviewer’s concerns and improve the transparency and reproducibility of our data presentation

**Reviewer #1 (Recommendations For The Authors):**

We are very thankful for the comprehensive review and comments which helped to improve our data.

Figure 1.The authors clearly show that FN stimulation (immobilized or cell-derived) promotes sEV secretion via canonical integrin pathways. FN is a promigratory substrate, hence its extensive use as a cell adhesion aid; thus one could assume that simply plating on FN induces a pro-migratory phenotype (later data supports this notion). Does the addition of growth factors also increase sEV release? An endogenous function of FN is siloing of various GFs during clot formation. Also, FAK and SRC networks intersect with canonical RTK signaling in terms of promoting Rac1, CDC42 and other migration mediators. The reason I believe this is important is because the data could be interpreted in two ways: (1) FN induces pro-migration signaling and then sEVs are released, or visa versa, FN induces sEV release and migration is initiated. GF supplementation in the absence of FN would clarify this relationship.

We thank the reviewer for this insightful comment regarding the possible role of growth factors (GFs) and the mechanistic relationship between FN stimulation, sEV secretion, and cell migration. We agree that FN is a well-established promoter of cell migration, and it is important to distinguish whether FN directly induces a pro-migratory phenotype or does so via sEV-mediated signaling.

Our data show that FN stimulation markedly increases VSMC motility, as reflected by enhanced cell speed (Fig. 5A), an increased number of focal adhesions (Fig. 6E), and facilitated centripetal movement of FAs (Fig. 6F). Interestingly, ECM-associated sEVs appear to play a complementary but distinct role: they do not significantly affect cell migration speed (Fig. 5A) but instead guide cell invasion directionality (Figs. 5G, 5H), reduce the number of FAs per cell (Fig. 6E), and promote early peripheral FA formation (Fig. 6F). In light of these findings, we have updated our graphical abstract to reflect the unique cross-talk mediated by sEVs between VSMCs and the ECM.

Regarding the influence of growth factors, we acknowledge that FN can bind and present different GFs, which could also contribute to changes in sEV secretion. Although our inhibition studies and integrin-blocking antibody results support a primary role for β1 integrin activation and actin assembly in triggering sEV secretion, we cannot entirely exclude the possibility that FN-bound growth factors play a role in this process. We have now incorporated this point into the discussion to address the reviewer’s suggestion.

Discussion, page 14 , Ln 7 “Although our small inhibitors and integrin modulating antibody data clearly indicate that β1 activation triggers sEV secretion via activation of actin assembly we cannot fully rule out that FN may also be modulating growth factor activity which in turn contributes to sEV secretion by VSMCs^23^. Excessive collagen and elastin matrix breakdown in atheroma has been tightly linked to acute coronary events hence it will be interesting to study the possible link between sEV secretion and plaque stability as sEV-dependent invasion is also likely to influence the necessary ECM degradation induced by invading cells^96^

Figure 2.• The authors provide no evidence (or references) that SMIFH2 or CK666 halts filopodia extensions.

Thank you for this important note. We have included the corresponding references:

Results, page 5: “So next we tested the contribution of Arp2/3 and formins by using the small molecule inhibitors, CK666 and SMIFH2, respectively31, 32”.

• Is there an increase in filopodia density when plated on FN vs plastic? Similarly, if there are more filopodia present is that associated with more sEV? Please provide evidence in this regard.

We agree that connecting the number of filopodia with the secretion level of sEVs may be an important clue if sEV secretion can be driven by FN-induced filopodia formation. However, Myosin10 staining to quantify filopodia (Fig 2E) showed no difference between VSMCs plated on plastic versus FN matrix. Therefore, we agree with the reviewer that the filopodia contribution to sEV secretion needs to be investigated further. This idea is reflected in the following comments:

(1) Results, page 6, Ln 29 “We observed no difference between total filopodia number per cell on plastic or FN matrices (n=18±8 and n=14±3, respectively) however the presence of endogenous CD63+ MVBs along the Myo10-positive filopodia were observed in both conditions (Fig 2E, arrows).

(2) Results, page 6, Ln 37 “We also attempted to visualise sEV release in filopodia using CD63-pHluorin where fluorescence is only observed upon the fusion of MVBs with the plasma membrane39. Using total internal reflection fluorescence microscopy (TIRF) we observed the typical “burst”-like appearance of sEV secretion at the cell-ECM interface in full agreement with an earlier report showing MVB recruitment to invadopodia-like structures in tumor cells18 (Fig S2B and Supplementary Video S1). Although we also observed an intense CD63-pHluorin staining along filopodia-like structures we were not able to detect typical “burst”-like events to confirm sEV secretion in filopodia. (Fig S2C and Supplemental Video S1)..”

(3) Discussion, page 12, Ln 15 : “Focal complexes either disassemble or mature into the elongated centripetally located FAs48. In turn, these mature FAs anchor the ECM to actin stress fibres and the traction force generated by actomyosin-mediated contractility pulls the FAs rearward and the cell body forward12, 13. Here we report that β1 integrin activation triggers sEV release followed by sEV entrapment by the ECM. Curiously we observed CD63+ MVB transport toward the filopodia tips as well as inhibition of sEV-secretion with filopodia formation inhibitors suggesting that sEV secretion can be directly linked to filopodia but further studies are needed to define the contribution of this pathway to the overall sEV secretion by cells..”

As hinted above, this data could be interpreted in the light of generally inhibiting cell migration to blunt sEV shedding. Does cell confluence affect sEV release? If cells are cultured to 100% confluency this would limit filopodia formation regardless of ECM type. If sEV secretion remains elevated on FN in this culture condition it would suggest a lack of dependency on filopodia.

We thank the reviewer for this thoughtful suggestion regarding the influence of cell confluence on sEV release and filopodia formation. To directly address this hypothesis, we performed additional experiments comparing VSMCs cultured at low and high confluency. As described in the revised Results (page 7, line 39), we found that high cellular confluency reduced FN recycling, as indicated by the marked decrease in intracellular FN-positive spots and loss of colocalization with CD63 (Figs S3F, S3G). Importantly, this was accompanied by a significant reduction in CD63+/CD81+ sEV secretion by confluent cells (Fig S3H). These results suggest that VSMC confluence, which suppresses filopodia formation, nearly abolishes both intracellular FN trafficking and sEV secretion, even in the presence of FN. Thus, under our experimental conditions, sEV secretion by VSMCs appears to be closely linked to dynamic cell–matrix interactions and is dramatically reduced when these processes are limited by confluence:

(1) Results, page 7, Ln 39 : “Notably, when we compared FN distribution in sparsely growing VSMCs versus confluent cells we found that FN intracellular spots, as well as colocalization with CD63, completely disappeared in the confluent state (Fig S3F and S3G). This correlated with nearly complete loss of CD63+/CD81+ sEV secretion by the confluent cells indicating that confluence abrogates intracellular FN trafficking as well as sEV secretion by VSMCs (Fig S3H)..

• Inhibition of branched actin polymerization has been shown to reduce both exocytic and endocytic activity. Thus, it is hard to interpret the results of Fig. 2B than anything more than a generalized effect of losing actin.

We thank the reviewer for this important point regarding the broad cellular functions of branched actin polymerization, and agree that generalized actin loss can influence both exocytic and endocytic pathways. To address this, we performed additional experiments and analyses to better define the relationship between branched actin structures and sEV-related processes in VSMCs.

As described in the revised Results (page 6), we overexpressed ARPC2-GFP (an Arp2/3 subunit) together with F-tractin-RFP in VSMCs and carried out live-cell imaging. This approach revealed that Arp2/3 and F-actin organize into lamellipodial scaffolds at the cell cortex, as expected (Fig. S2A; Supplementary Video S2). Additionally, and more unexpectedly, we observed numerous Arp2/3– and F-actin–positive dynamic spots within the VSMC cytoplasm. These structures resemble actin comet tails seen in other systems, previously implicated in endosomal propulsion (Fig. S2A, arrow; Supplementary Video S2).

Quantitative analysis confirmed that a substantial fraction of these dynamic F-actin/cortactin spots colocalized with CD63+ endosomes (Fig. 2D), and that these structures are indeed branched actin tails based on cortactin immunostaining. Furthermore, inhibition of SMPD3 (with 3-OMS) induced enlarged cortactin/F-actin/CD63+ complexes, morphologically similar to invadopodia (Fig. 2D, arrowheads), supporting a functional link between actin branching and MVB dynamics.

To quantify the association, we calculated Manders’ colocalization coefficients for F-actin tails and CD63+ endosomal structures in fixed VSMCs, observing that ~50% of F-actin tails were associated with ~13% of endosomes. Upon 3-OMS treatment, this overlap increased further (Fig. S2B).

Finally, using live-cell imaging (Fig 2C; Supplementary Video S4), we directly observed CD63+ MVBs being propelled through the cytoplasm by Arp2/3-driven actin tails, suggesting a mechanistic role for branched actin assembly in MVB intracellular transport, rather than a generalized effect of actin disruption alone.

We believe these combined data reinforce a more specific mechanistic role for Arp2/3-mediated branched actin in MVB/endosome transport and, consequently, in sEV secretion in VSMCs—over and above an indirect effect of global actin loss. We hope these additional experiments and quantitative analyses address the reviewer’s concern and clarify the functional relevance of branched actin structures to sEV trafficking:

(1) Results, page 6, Ln 3 “As regulators of branched actin assembly, the Arp2/3 complex and cortactin are thought to contribute to sEV secretion in tumour cells by mediating MVB intracellular transport and plasma membrane docking[28, 33]. Therefore, we overexpressed the Arp2/3 subunit, ARPC2-GFP and the F-actin marker, F-tractin-RFP in VSMCs and performed live-cell imaging. As expected, Arp2/3 and F-actin bundles formed a distinct lamellipodia scaffold in the cellular cortex (Fig S2A and Supplementary Video S2). Unexpectedly, we also observed numerous Arp2/3/F-actin positive spots moving through the VSMC cytoplasm that resembled previously described endosome actin tails observed in Xenopus eggs[33] and parasite infected cells where actin comet tails propel parasites via filopodia to neighbouring cells[34, 35] (Fig S2A, arrow, and Supplementary Video S2). Analysis of the intracellular distribution of Arp2/3 and CD63-positive endosomes in VSMCs showed CD63-MVB propulsion by the F-actin tail in live cells (Fig 2C and Supplementary Video S4).”

(2) Results, New data Fig 2D, page 6, Ln 14. “we observed numerous F-actin spots in fixed VSMCs that were positive both for F-actin and cortactin indicating that these are branched-actin tails (Fig 2D). Moreover, cortactin/F-actin spots colocalised with CD63+ endosomes and addition of the SMPD3 inhibitor, 3-OMS, induced the appearance of enlarged doughnut-like cortactin/F-actin/CD63 complexes resembling invadopodia-like structures similar to those observed in tumour cells (Fig 2D, arrowheads)[18].”

(3) Results, New data Fig S2B, page 6, Ln 19 “To quantify CD63 overlap with the actin tail-like structures, we extracted round-shaped actin structures and calculated the thresholded Manders colocalization coefficient (Fig S2B). We observed overlap between F-actin tails and CD63 as well as close proximity of these markers in fixed VSMCs (Fig S2B). Approximately 50% of the F-actin tails were associated with 13% of all endosomes (tM1=0.44±0.23 and tM2 = 0.13±0.06, respectively, N=3). Addition of 3-OMS enhanced this overlap further (tM1=0.75±0.18 and tM2=0.25±0.09) suggesting that Arp2/3-driven branched F-actin tails are involved in CD63+ MVB intracellular transport in VSMCs”

• In video 1 the author states (lines 8-9; pg6) "intense CD63 staining along filopodia" Although, there is some fluorescence (not strong) in these structures, there was no visible exocytic activity. This data is more suggestive that sEVs (marked by CD63) are not associated with filopodia. The following conclusion statement the authors make is overreaching given this result.

We thank the reviewer for this careful observation and agree that the previous conclusion regarding sEV release from filopodia was overstated. In response, we have revised both the Results and Discussion sections to more accurately reflect the data..

(1) Results, page 6, Ln37 “We also attempted to visualise sEV release in filopodia using CD63-pHluorin where fluorescence is only observed upon the fusion of MVBs with the plasma membrane39. Using total internal reflection fluorescence microscopy (TIRF) we observed the typical “burst”-like appearance of sEV secretion at the cell-ECM interface in full agreement with an earlier report showing MVB recruitment to invadopodia-like structures in tumor cells18 (Fig S2B and Supplementary Video S1). Although we also observed an intense CD63-pHluorin staining along filopodia-like structures we were not able to detect typical “burst”-like events to confirm sEV secretion in filopodia. (Fig S2C and Supplemental Video S1)..”

(2) Discussion, page 12, Ln19 “Curiously we observed CD63+ MVB transport toward the filopodia tips as well as inhibition of sEV-secretion with filopodia formation inhibitors suggesting that sEV secretion can be directly linked to filopodia but further studies are needed to define the contribution of this pathway to the overall sEV secretion by cells.”.

• Fig 2D and video 2 are wholly unconvincing with regard to sEV secretion sites. The authors could use their CD63-pHluroin construct to count exocytic events in the filopodia vs the whole cell. Given the movie, I have a suspicion this would not be significant. The authors could also perform staining CD63 in non-permeabilized cells to capture and count exocytic events at the plasma membrane as well as their location between groups.

We thank the reviewer for these constructive suggestions and their critical assessment of our current data regarding the sites of sEV secretion. We agree that our CD63-pHluorin approach clearly indicates sEV secretion events in the soma at the cell–ECM interface, while we did not observe comparable events in filopodia. Accordingly, we have clarified these points in the revised manuscript.

(1) Results, page 6, Ln37 “We also attempted to visualise sEV release in filopodia using CD63-pHluorin where fluorescence is only observed upon the fusion of MVBs with the plasma membrane39. Using total internal reflection fluorescence microscopy (TIRF) we observed the typical “burst”-like appearance of sEV secretion at the cell-ECM interface in full agreement with an earlier report showing MVB recruitment to invadopodia-like structures in tumor cells18 (Fig S2B and Supplementary Video S1). Although we also observed an intense CD63-pHluorin staining along filopodia-like structures we were not able to detect typical “burst”-like events to confirm sEV secretion in filopodia. (Fig S2C and Supplemental Video S1)..”

(2) Discussion, page 12, Ln19 “Curiously we observed CD63+ MVB transport toward the filopodia tips as well as inhibition of sEV-secretion with filopodia formation inhibitors suggesting that sEV secretion can be directly linked to filopodia but further studies are needed to define the contribution of this pathway to the overall sEV secretion by cells.”.

• Fig. 2E and video 4. Again, the conclusions drawn from this data are very strained. First, no co-localization quantification is presented on the proportion of CD63 vesicles with actin. Once again, the movie, if anything convinces the reader that 95-99% of all CD63 vesicles are not associated with actin; therefore, this is an unlikely mechanism of transport.

We thank the reviewer for this valuable comment and for highlighting the need for quantitative co-localization analysis. In response, we developed a method to systematically quantify F-actin and CD63 co-localization in fixed VSMCs, as now presented in new Figures 2D and S2B. We acknowledge that the majority of CD63+ endosomes are not associated with F-actin, consistent with the reviewer’s interpretation. However, our quantitative data now show that a specific subpopulation of MVBs appears to utilize this actin-based mechanism for transport. We believe this addresses the concern and more accurately reflects the prevalence and significance of the mechanism described.

(1) Results, page 6 , Ln 19. “To quantify CD63 overlap with the actin tail-like structures, we extracted round-shaped actin structures and calculated the thresholded Manders colocalization coefficient (Fig S2B). We observed overlap between F-actin tails and CD63 as well as close proximity of these markers in fixed VSMCs (Fig S2B). Approximately 50% of the F-actin tails were associated with 13% of all endosomes (tM1=0.44±0.23 and tM2 = 0.13±0.06, respectively, N=3). Addition of 3-OMS enhanced this overlap further (tM1=0.75+/-0.18 and tM2=0.25+/-0.09) suggesting that Arp2/3-driven branched F-actin tails are involved in CD63+ MVB intracellular transport in VSMCs.”

• Are there perturbations that increase filopodia numbers? A gain of function experiment would be valuable here.

We thank the reviewer for this important suggestion regarding the potential value of gain-of-function experiments to clarify filopodia’s contribution to sEV release. In agreement with the reviewer’s scepticism, we have removed statements linking filopodia to sEV release from both the title and abstract to avoid overinterpretation. At present, our understanding of filopodia biology and the lack of robust tools to selectively and substantially increase filopodia numbers in VSMCs prevent us from directly addressing this question through gain-of-function assays. We acknowledge that future studies using established methods—such as overexpression of filopodia-inducing proteins (e.g., mDia2 or fascin)—could provide insight into whether an increased number of filopodia affects sEV release. However, such experiments are beyond the scope of the current manuscript. We have made the following changes to clarify these points:

(1) Results, page 6, Ln37 “We also attempted to visualise sEV release in filopodia using CD63-pHluorin where fluorescence is only observed upon the fusion of MVBs with the plasma membrane39. Using total internal reflection fluorescence microscopy (TIRF) we observed the typical “burst”-like appearance of sEV secretion at the cell-ECM interface in full agreement with an earlier report showing MVB recruitment to invadopodia-like structures in tumor cells18 (Fig S2B and Supplementary Video S1). Although we also observed an intense CD63-pHluorin staining along filopodia-like structures we were not able to detect typical “burst”-like events to confirm sEV secretion in filopodia. (Fig S2C and Supplemental Video S1)..”

(2) Discussion, page 12, Ln19 “Curiously we observed CD63+ MVB transport toward the filopodia tips as well as inhibition of sEV-secretion with filopodia formation inhibitors suggesting that sEV secretion can be directly linked to filopodia but further studies are needed to define the contribution of this pathway to the overall sEV secretion by cells.”.

Figure 3• Fig 3A. The CD63 staining is strongly associated with the entire plasma membrane. How are the authors distinguishing between normal membrane shedding and bona fida sEVs based on this staining alone (?)- this is insufficient as all membrane structures are seemingly positive. Additionally, there are very few sEVs in scrutinizing the provided images. For the "sEV secretion, fold change" graphs in previous figures, could the authors provide absolute values, or an indication of what these values are in absolute terms?

We thank the reviewer for raising this important point regarding the specificity of CD63 staining and the need to distinguish bona fide sEVs from membrane fragments or general membrane shedding. We agree that CD63 staining alone at the plasma membrane or in the extracellular matrix is not sufficient to unequivocally identify sEVs. To address this, we employed several complementary approaches to rigorously characterize ECM-associated sEVs:

First, using high-resolution iSIM imaging, we confirmed the association of CD63-positive particles specifically with the FN-rich matrix, and demonstrated that SMPD3 knockdown significantly reduced the number of CD63+ particles in the matrix (Fig. 3B; revised from Fig. S3A).

Second, by incubating FN matrices with purified and fluorescently labeled sEVs, we directly observed efficient entrapment of these labeled sEVs within the matrices (Fig. 3E), confirming that sEVs can interact with and be retained by the ECM.

Third, we developed and applied a sequential extraction protocol using mild salt buffer (0.5M NaCl) and strong denaturant (4M guanidine HCl) to selectively extract ECM-associated sEVs based on the strength of their association (see new Figs. S3A and S3B). Extracted vesicles were then characterized by ExoView analysis, which demonstrated a tetraspanin profile (CD63+/CD81+/CD9+) closely matching that of sEVs from conditioned media, providing evidence that these particles are true sEVs and not merely membrane debris. We also found that the more weakly bound (NaCl-extracted) fraction closely resembles media-derived sEVs, while the strongly bound (GuHCl-extracted) fraction is more enriched in CD63+ and CD63+/CD81+ sEVs but contains very few CD9+ vesicles, further supporting distinct extracellular vesicle subpopulations within the ECM.

In addition, the abundance of CD63+/CD81+ sEVs in both media and ECM-derived fractions was independently validated by CD63 bead-capture assay (Fig. S3B).

We hope these clarifications and the expanded data set address the reviewer’s concerns about sEV identification and quantification in the extracellular matrix:

(1) Results, page 7, Ln 16. To quantify ECM-trapped sEVs we applied a modified protocol for the sequential extraction of extracellular proteins using salt buffer (0.5M NaCl) to release sEVs which are loosely-attached to ECM via ionic interactions, followed by 4M guanidine HCl buffer (GuHCl) treatment to solubilize strongly-bound sEVs (Fig S3A) 42. We quantified total sEV and characterised the sEV tetraspanin profile in conditioned media, and the 0.5M NaCl and GuHCl fractions using ExoView. The total particle count showed that EVs are both loosely bound and strongly trapped within the ECM. sEV tetraspanin profiling showed differences between these 3 EV populations. While there was close similarity between the conditioned media and the 0.5M NaCl fraction with high abundance of CD63+/CD81+ sEVs as well as CD63+/CD81+/CD9+ in both fractions (Fig S3A). In contrast, the GuHCl fraction was particularly enriched with CD63+ and CD63+/CD81+ sEVs with very low abundance of CD9+ EVs (Fig S3A). The abundance of CD63+/CD81+ sEVs was confirmed independently by a CD63+ bead capture assay in the media and loosely bound fractions (Fig S3B).

• A control of fig 3b would be helpful to parse out random uptake of extracellular debris verses targeted sEV internalization. It would be helpful if the authors added particles of similar size to that of the sEVs to test whether these structures are endocytosed/micropinocytosed at similar levels.

We thank the reviewer for this useful suggestion regarding the need for better controls to distinguish specific sEV uptake from nonspecific internalization of extracellular debris or similarly sized particles. As a comparison, in our study we analyzed the uptake of both sEVs and serum proteins such as fibronectin and fetuin-A (Figs S3C and S3D), and observed similar patterns of intracellular trafficking. However, we acknowledge that inert nanoparticles or beads of a similar size to sEVs could serve as potential controls to assess nonspecific micropinocytosis or endocytosis.

It is important to note, however, that the uptake of sEVs is strongly influenced by their surface protein composition and the so-called “protein corona.” Recent work from Prof. Khuloud T. Al-Jamal’s group underscores that exosome uptake mechanisms may be highly specific (Liam-Or et al., 2024), and studies from Mattias Belting’s lab have also shown the importance of heparan sulfate proteoglycans in exosome endocytosis (Cerezo-Magana et al., 2021). As a result, uptake comparisons with inert particles or beads may not fully recapitulate the specificity of sEV internalization, and distinct nanoparticle classes may rely on different uptake pathways.

Figure 4• Fig. 4E,F,G. How are the authors determining the neointima and media compartments without ancillary staining for basement membrane or endothelial markers? Anatomic specific markers need to be incorporated here for the reader to evaluate the specificity of the FN and CD81 staining. It is also hard to understand the severity of the atherosclerotic lesion without a companion H&E cross section.

We thank the reviewer for highlighting the need for more rigorous characterization of atherosclerotic lesion architecture and anatomical compartments in our study. In response, we have incorporated additional histological analyses and now provide ancillary staining and companion images to enable clear identification of the neointima and medial compartments, as well as to assess lesion severity (see new Figs S4A–S4D):

(1)Results, page 8, Ln 28. . “To test if FN associates with sEV markers in atherosclerosis, we investigated the spatial association of FN with sEV markers using the sEV-specific marker CD81. Staining of atherosclerotic plaques with haematoxylin and eosin revealed well-defined regions with the neointima as well as tunica media layers formed by phenotypically transitioned or contractile VSMCs, respectively (Fig S4A). Masson's trichrome staining of atherosclerotic plaques showed abundant haemorrhages in the neointima, and sporadic haemorrhages in the tunica media (Fig S4B). Staining of atherosclerotic plaques with orcein indicated weak connective tissue staining in the atheroma with a confluent extracellular lipid core, and strong specific staining at the tunica media containing elastic fibres which correlated well with the intact elastin fibrils in the tunica media (Figs S4C and S4D). Using this clear morphological demarcation, we found that FN accumulated both in the neointima and the tunica media where it was significantly colocalised with the sEV marker, CD81 (Fig. 4D, 4E and 4F). Notably CD81 and FN colocalization was particularly prominent in cell-free, matrix-rich plaque regions (Figs. 4E and 4F).”

• Figs s4c, S4d- proper controls are not provided. Again, a non-FN internalization control as well as a 4oC cold block negative control is required to interpret this data.

We thank the reviewer for this valuable suggestion. To enhance the rigor of our internalization assays, we have now included several additional controls using alternative treatments, fluorophore combinations, and internalization conditions:

a) We performed FN-Alexa568 uptake assays, followed by immunostaining for CD63 with a distinct fluorophore (Alexa488), to confirm the colocalization of internalized FN with CD63+ endosomal compartments in VSMCs (new Fig. S3E).

b) We also stained VSMCs, cultured under normal growth conditions, with an anti-FN antibody to visualize intracellular serum-derived FN and again observed colocalization with CD63 (new Figs. S3F and S3G). Notably, in cells grown to confluence, we observed a complete loss of intracellular FN staining and FN/CD63 colocalization, suggesting that FN recycling is prominent in sparse, motile cells, but not in confluent populations.

These additional controls strengthen our conclusions regarding FN internalization pathways and the conditions under which FN trafficking to the endosomal system occurs:

(1) Results, page 7, Ln 31 We treated serum-deprived primary human aortic VSMCs with FN-Alexa568 and found that it was endocytosed and subsequently delivered to early and late endosomes together with fetuin A, another abundant serum protein that is a recycled sEV cargo and elevated in plaques (Figs S3C and S3D). CD63 visualisation with a different fluorophore (Alexa488) confirmed FN colocalization with CD63+ MVBs (Fig S3E). Next, we stained non-serum deprived VSMC cultured in normal growth media (M199 supplemented with 20% FBS) with an anti-FN antibody and observed colocalization of CD63 and serum-derived FN. Co-localisation was reduced likely due to competitive bulk protein uptake by non-deprived cells (Fig S3F). Notably, when we compared FN distribution in sparsely growing VSMCs versus confluent cells we found that FN intracellular spots, as well as colocalization with CD63, completely disappeared in the confluent state (Fig S3F and S3G)..

• Can the authors please provide live and fixed imaging of FN and CD63-mediate filopodial secretion to amply support their conclusions.

We have observed CD63 MVBs in both fixed (Fig 2E) and live VSMCs (Fig 2F) yet we agree that further studies are required to establish the contribution of filopodia to sEV secretion. Therefore, we have added the following changes:

(1) Results, page 6, Ln37 “We also attempted to visualise sEV release in filopodia using CD63-pHluorin where fluorescence is only observed upon the fusion of MVBs with the plasma membrane39. Using total internal reflection fluorescence microscopy (TIRF) we observed the typical “burst”-like appearance of sEV secretion at the cell-ECM interface in full agreement with an earlier report showing MVB recruitment to invadopodia-like structures in tumor cells18 (Fig S2B and Supplementary Video S1). Although we also observed an intense CD63-pHluorin staining along filopodia-like structures we were not able to detect typical “burst”-like events to confirm sEV secretion in filopodia. (Fig S2C and Supplemental Video S1)..”

(2) Discussion, page 12, Ln19 “Curiously we observed CD63+ MVB transport toward the filopodia tips as well as inhibition of sEV-secretion with filopodia formation inhibitors suggesting that sEV secretion can be directly linked to filopodia but further studies are needed to define the contribution of this pathway to the overall sEV secretion by cells.”.

Figure 5• Fig. 5A,B. The authors claim that sEV supplementation enhances VSMC migration speed and distance. The provided graphs show only a marginal increase in speed with sEV addition (A) but, concerningly, there is a four-star significant difference between the FN condition compared with FN+sEV (B) while the means appear the same. How are these conditions statistically different? The statistics seem off for these comparisons.

We thank the reviewer for highlighting concerns regarding the statistical analysis in Figures 5A and 5B. In response, we have carefully re-examined our data and statistical approach to ensure accuracy and transparency.

First, we have now included all individual cell migration tracks in the data representation for these figures. The statistical tests were repeated using the Kruskal–Wallis test with Dunn’s multiple comparison correction across all groups. This more stringent analysis confirmed our key findings: fibronectin (FN) stimulates VSMC migration speed, while inhibition of sEV secretion (with 3-OMS) reduces cellular speed (Fig. 5A). Addition of exogenous ECM-associated sEVs modestly restored cell speed in the presence of 3-OMS, but had no effect on baseline migration speed in 2D or 3D models (Figs. 5A, 5D).

Regarding the four-star significance observed in the original Fig. 5B, the previous result reflected an analysis based on pooled group averages, which may have overstated marginal differences. The revised analysis, based on individual cell tracks, does not support a substantial difference between FN and FN+sEV groups. The revised p-values and comparisons are now provided directly on the figures and described in the figure legends. We also clearly report the numbers of biological replicates, technical replicates, and individual data points for every condition.

Further, the modest effect of ECM-associated sEVs on speed is consistent with our observation that sEVs influence invasion directionality rather than baseline migration velocity, in agreement with previous findings in tumor models (Sung et al., 2015).

The manuscript has been revised accordingly, with updates in:

(1) Figures 5A and 5B: Individual cell track data are now shown, and statistical analyses have been repeated using the Kruskal–Wallis test with Dunn’s multiple comparisons.

(2) Figure legends and results sections: Numbers of biological and technical replicates, as well as individual data points, are now clearly stated.

(3) Results, page 9, line 14: “FN as a cargo in sEVs promotes FA formation in tumour cells and increases cell speed14, 15. As we found that FN is loaded into VSMC-derived sEVs we hypothesized that ECM-entrapped sEVs can enhance cell migration by increasing cell adhesion and FA formation in the context of a FN-rich ECM. Therefore, we tested the effect of sEV deposition onto the FN matrix on VSMC migration in 2D and 3D models. We found that FN coating promoted VSMC velocity and inhibition of bulk sEV secretion with 3-OMS reduced VSMC speed in a 2D single-cell migration model (Figs. 5A, 5B) in agreement with previous studies using tumour cells14, 15. However, addition of sEVs to the ECM had no effect on VSMC speed at baseline but rescued cell speed and distance in the presence of the sEV secretion inhibitor, 3-OMS suggesting the EVs are not primarily regulating cell speed (Figs 5A and 5B).”

(4) Results, page 9, Ln 29 “Hence, ECM-associated sEVs have modest influence on VSMC speed but influence VSMC invasion directionality.”.

We hope that these changes address the reviewer’s concerns and improve the transparency and reproducibility of our data presentation

• Fig d-h. Generally, the magnitude of the difference between the presented conditions are biologically insignificant. Several of the graphs show a four-star difference with means that appear equivalent with overlapping error bars. Do the authors conclude that a 0.1%, or less, effect between groups is biologically meaningful?

We thank the reviewer for drawing attention to the apparent mismatch between statistical significance and biological relevance in Figures 5d–h. In response, we have reanalyzed the data using individual cell tracks and more stringent non-parametric statistical tests, as described above. This reanalysis confirmed that the magnitude of differences in migration speed and related parameters between the groups is minimal and not biologically meaningful. Thus, we no longer claim that sEVs significantly affect VSMC migration speed under these conditions in either 2D or 3D assays. Our revised manuscript now accurately reflects this finding in both the Results and Discussion sections, and the updated figures and legends clarify the true extent of any differences observed.

Figure 6• Generally, the author's logic for looking into adhesion, focal adhesion and traction forces is hard to follow. If there are sEV-mediated migration differences, then there would inexorably be focal adhesion alterations. However, the data indicates few differences brought on by sEVs, which speaks to the lack of migration differences presented in Fig. 5. Overall, the sEV migration phenotype has so little of an effect, to then search for a mechanism seems destine to not turn up anything significant.

We thank the reviewer for highlighting the importance of connecting the observed phenotypic effects of sEVs to the investigation of adhesion and focal adhesion mechanisms. While our revised analysis confirms that sEVs have little to no effect on VSMC migration speed or distance in 2D and 3D models, we did observe a robust effect of sEVs on the directionality of cell invasion (Figs. 5G and 5H). This prompted us to look more closely at pathways involved in cell guidance rather than bulk cell motility.

Our proteomic comparison between larger EVs (10K fraction) and sEVs (100K fraction) revealed a unique adhesion complex present specifically on the sEVs—comprising collagen VI, TGFBI, LGALS3BP, and EDIL3 (Figs. 7A–C)—each of which has previously been implicated in integrin signaling, cell adhesion, or invasion. Functional blocking and knockdown studies further identified collagen VI as a key mediator in the regulation of cell adhesion and invasion directionality influenced by sEVs (Figs. 7F and 7I).

In response to this mechanistic insight, we have modified the graphical abstract and discussion to clarify our approach:

We now explicitly state that our focus has shifted from analyzing baseline migration speed to mechanisms guiding invasion directionality, in line with our key phenotypic findings.We highlight that the unique adhesion cluster identified on sEVs—including collagen VI and its cooperative partners—provides a strong mechanistic rationale for examining focal adhesion dynamics and ECM interactions, even in the absence of changes in migration velocity.Discussion excerpts (pages 13–14) have been updated to reflect this rationale and to summarize the potential significance of these findings for vascular biology and disease.

We hope this clarifies the logic underlying our approach and justifies the mechanistic studies performed in this context:

(1) Discussion, page 13, Ln 2 “Hence, it will be interesting in future studies to investigate whether sEVs can stimulate Rho activity by presenting adhesion modulators—particularly collagen VI—on their surface, thereby guiding cell directionality during invasion.”

(2) Discussion, page 13, Ln 30 “In addition to collagen VI the unique adhesion cluster in VSMC-derived sEVS also includes EGF-like repeat and discoidin I-like domain-containing protein (EDIL3), transforming growth factor-beta-induced protein ig-h3 (TGFBI) and the lectin galactoside-binding soluble 3 binding protein (LGALS3BP) and these proteins are also directly implicated in activation of integrin signalling and cellular invasiveness85-87. Although we found that collagen VI plays the key role in sEV-induced early formation of FAs in VSMCs, it is tempting to speculate that the high sEV efficacy in stimulating FA formation is driven by cooperative action of this unique adhesion complex on the sEVs surface and targeting this novel sEV-dependent mechanism of VSMC invasion may open-up new therapeutic opportunities to modulate atherosclerotic plaque development or even to prevent undesired VSMC motility in restenosis”. .

(3) Discussion, page 14, Ln 14 “In summary, cooperative activation of integrin signalling and F-actin cytoskeleton pathways results in the secretion of sEVs which associate with the ECM and play a signalling role by controlling FA formation and cell-ECM crosstalk. Further studies are needed to test these mechanisms across various cell types and ECM matrices. ”.

Figure 7• The authors need to provide additional evidence Col IV is harbored in sEVs and not a contaminant of sEV isolation as VSMCs secrete a copious amount of this in culture. For instance, IHC of isolated sEVs stained for CD63 and Col IV as well as single cell staining of the same sort.

We thank the reviewer for this important comment regarding the specificity of collagen VI detection in sEVs. To ensure that collagen VI is associated with bona fide sEVs—rather than being a contaminant resulting from high extracellular abundance—we performed a comparative analysis of vesicles isolated from the same conditioned media. Both proteomic mass spectrometry and western blotting revealed that collagen VI was exclusively present in the small EV (100K pellet) fraction and not in the larger EVs (10K pellet), as shown in Figs. 7B and 7C. Collagen VI was further identified in sEVs extracted from the ECM using our salt/guanidine protocol (new Fig. 7D).

**Reviewer #2 (Recommendations For The Authors):**
The authors have presented a nice collection of data with strong approaches to address their hypotheses. Nevertheless, an additional section within the Discussion would be welcome in addressing the potential limitations and important caveats to be considered alongside their study. These caveats and limitations could be reshaped by additional data supporting the ideas that: (1) small extracellular vesicles can be directly observed during their secretion from filopodia, (2) CD81 labeling in tissue can be interpreted clearly as extracellular vesicles and not the cell surface of other cell types (co-staining with an endothelial cell marker such as PECAM-1 perhaps), and (3) collagen VI within the vesicles is somehow accessed by adhesion molecules on the cell surface of migrating cells.

We thank the reviewer for these important suggestions and we have now added further studies and modified our conclusions to reflect the data more accurately:

(1) Results. Page 6, Ln37 “We also attempted to visualise sEV release in filopodia using CD63-pHluorin where fluorescence is only observed upon the fusion of MVBs with the plasma membrane39. Using total internal reflection fluorescence microscopy (TIRF) we observed the typical “burst”-like appearance of sEV secretion at the cell-ECM interface in full agreement with an earlier report showing MVB recruitment to invadopodia-like structures in tumor cells18 (Fig S2B and Supplementary Video S1). Although we also observed an intense CD63-pHluorin staining along filopodia-like structures we were not able to detect typical “burst”-like events to confirm sEV secretion in filopodia. (Fig S2C and Supplemental Video S1)”..

(2) Discussion, page 12, Ln18: “Here we report that β1 integrin activation triggers sEV release followed by sEV entrapment by the ECM. Curiously we observed CD63+ MVB transport toward the filopodia tips as well as inhibition of sEV-secretion with filopodia formation inhibitors suggesting that sEV secretion can be directly linked to filopodia but further studies are needed to define the contribution of this pathway to the overall sEV secretion by cells”..

We quantified the colocalization of CD81 and CD31 to exclude the endothelial cell origin of sEVs and extended the characterisation of the atherosclerotic matrix as well as highlighting any limitations to interpretation ie re CD81 ECM localisation:

(1) Results, page 8, Ln 43 “An enhanced expression of CD81 by endothelial cells in early atheroma has been previously reported so to study the contribution of CD81+ sEVs derived from endothelial cells we investigated the localisation of CD31 and CD8145. In agreement with a previous study, we found that the majority of CD31 colocalises with CD81 (Thresholded Mander's split colocalization coefficient 0.54±0.11, N=6) indicating that endothelial cells express CD81 (Fig 4G)45. However, only a minor fraction of total CD81 colocalised with CD31 (Thresholded Mander's split colocalization coefficient 0.24±0.06, N=6) confirming that the majority of CD81 in the neointima is originating from the most abundant VSMCs..

(2) Results, page 8, Ln 28: “To test if FN associates with sEV markers in atherosclerosis, we investigated the spatial association of FN with sEV markers using the sEV-specific marker CD81. Staining of atherosclerotic plaques with haematoxylin and eosin revealed well-defined regions with the neointima as well as tunica media layers formed by phenotypically transitioned or contractile VSMCs, respectively (Fig S4A). Masson's trichrome staining of atherosclerotic plaques showed abundant haemorrhages in the neointima, and sporadic haemorrhages in the tunica media (Fig S4B). Staining of atherosclerotic plaques with orcein indicated weak connective tissue staining in the atheroma with a confluent extracellular lipid core, and strong specific staining at the tunica media containing elastic fibres which correlated well with the intact elastin fibrils in the tunica media (Figs S4C and S4D). Using this clear morphological demarcation, we found that FN accumulated both in the neointima and the tunica media where it was significantly colocalised with the sEV marker, CD81 (Fig. 4D, 4E and 4F). Notably CD81 and FN colocalization was particularly prominent in cell-free, matrix-rich plaque regions (Figs. 4E and 4F). .”

We showed that collagen VI is presented on the surface of sEVs:

(1) Results, page 10, Ln43: “Collagen VI was the most abundant protein in VSMC-derived sEVs (Fig 7B, Table S7) and was previously implicated in the interaction with the proteoglycan NG253 and suppression of cell spreading on FN54. To confirm the presence of collagen VI in ECM-associated sEVs we analysed sEVs extracted from the 3D matrix using 0.5M NaCl treatment and showed that both collagen VI and FN are present (Fig 7D). Next, we analysed the distribution of collagen VI using dot-blot. Alix staining was bright only upon permeabilization of sEV indicating that it is preferentially a luminal protein (Fig 7E). On the contrary, CD63 staining was similar in both conditions showing that it is surface protein (Fig 7E). Interestingly, collagen VI staining revealed that 40% of the protein is located on the outside surface with 60% in the sEV lumen (Fig 7E)